# Provable Scaling Laws of Feature Emergence from Learning Dynamics of Grokking

**Yuandong Tian**
Meta Superintelligence Labs (FAIR) & Independent Researcher
`yuandong.tian@gmail.com`

## Abstract

While the phenomenon of grokking, i.e., delayed generalization, has been studied extensively, it remains an open problem whether there is a mathematical framework that characterizes what kind of features will emerge, how and in which conditions it happens, and is still closely connected with the gradient dynamics of the training, for complex structured inputs. We propose a novel framework, named $\text{Li}_2$, that captures three key stages for the grokking behavior of 2-layer nonlinear networks: (I) **L**azy learning, (II) **i**ndependent feature learning and (III) **i**nteractive feature learning. At the lazy learning stage, top layer overfits to random hidden representation and the model appears to memorize, and at the same time, the *backpropagated gradient $G_F$* from the top layer carries information about the target label, with a specific structure that enables each hidden node to learn their representation *independently*. Moreover, the independent dynamics follows exactly the *gradient ascent* of an energy function $\mathcal{E}$, and its local maxima are precisely the emerging features. We study whether these local-optima induced features are generalizable, their representation power, and how they change on sample size, in group arithmetic tasks. When hidden nodes start to interact in the later stage of learning, we provably show how $G_F$ changes to focus on missing features that need to be learned. Our study sheds lights on roles played by key hyperparameters such as weight decay, learning rate and sample sizes in grokking, leads to provable scaling laws of feature emergence, memorization and generalization, and reveals the underlying cause why recent optimizers such as Muon can be effective, from the first principles of gradient dynamics. Our analysis can be extended to multilayer architectures. The code is available[1].

## 1 Introduction

While modern deep models such as Transformers have achieved impressive empirical performance, it remains a mystery how such models acquire the knowledge during the training process. There have been ongoing arguments on whether the models can truly generalize beyond what it is trained on, or just memorize the dataset and performs poorly in out-of-distribution (OOD) data (Wang et al., 2024b; Chu et al., 2025; Mirzadeh et al., 2024).

Modeling the memorization and generalization behaviors have been a goal of many works. One such behavior, know as *grokking* (Power et al., 2022; Doshi et al., 2024; Nanda et al., 2023; Wang et al., 2024a; Varma et al., 2023; Liu et al., 2023; Thilak et al., 2022), shows that the model initially overfits to the training set, and then suddenly generalizes to unseen test samples after continuous training. Many explanation exists, e.g., effective theory (Liu et al., 2022; Clauw et al., 2024), efficiency of memorization and generalization circuits (Varma et al., 2023), Bayesian interpretation with weight decay as prior (Millidge, 2022), etc. Most works focus on a direct explanation of its empirical behaviors, or leveraging property of very wide networks (Barak et al., 2022; Mohamadi et al., 2024; Rubin et al., 2024), but few explores the details of the grokking learning procedure by studying the gradient dynamics on the weights.

In this work, we propose a mathematical framework $\text{Li}_2$ that divides the grokking dynamics for 2-layer nonlinear networks into three major stages (Fig. 1). *Stage I: **L**azy Learning*: when training

---

[1] `https://github.com/yuandong-tian/understanding/tree/main/ssl/real-dataset/cogo`

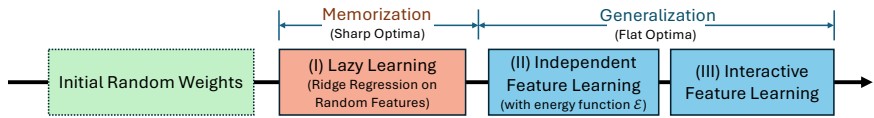

Figure 1: Three stages of feature learning. In Stage I (*Lazy learning*), model first learns to overfit on random features provided by the bottom layer. For generalizable targets, the backpropagated gradients to the feature layer $G_F \propto \tilde{Y}\tilde{Y}^\top \tilde{F}$ becomes structured and signals task structures, triggering Stage II (*independent feature learning*). Different hidden nodes independently learn to extract multiple task-related features. In Stage III (*interactive feature learning*), these features interact and consolidate via backpropagated signal.

begins, the top (output) layer learns first with random features from the hidden layer, and at the same time, the backpropagated gradient $G_F$ to the hidden layer starts to carry useful signal about the target label (Proposition 1). *Stage II: Independent feature learning*: $G_F$ drives the learning of hidden representations. In this stage, the backpropagated gradient of $j$-th neuron (node) only depends on its own activation, triggering independent feature learning for each node. *Stage III: Interactive feature learning*: When weights in the hidden layer get updated and are no longer independent, interactions across nodes adjust the learned feature to minimize the loss.

Each stage is studied in details. In *Stage I*, we show that the signal carried by the backpropagated gradient $G_F$ either appears at the initial stage (Proposition 1) which depends on the top layer initialization, or the final stage (Lemma 1) which depends on the weight decay $\eta$. In *Stage II*, we find that the dynamics that governs independent feature learning exactly follows the *gradient ascent* of an energy function $\mathcal{E}$ (Thm. 1), which corresponds to nonlinear canonical-correlation analysis (CCA) between the input $X$ and target $Y$. In group arithmetic tasks, we completely characterize the local maxima structure of $\mathcal{E}$ (Thm. 2), each representing one learned feature. We further show that the amount of training samples determine whether these features/local optima remain stable (Thm. 4), whether they are generalizable or not, and thus determine the scaling laws of generalization and memorization. In *Stage III*, we provably show that there is a push for diversity between hidden nodes with similar activations (Thm. 6), when a subset of features are learned, backpropagated gradient $G_F$ changes to push the model to focus on missing features that need to be learned (Thm. 7), and optimizers like Muon (Jordan et al., 2024) further improves the diversity (Thm. 8). Experiments on graph arithmetic tasks support our claims, e.g., the proved scaling laws about the generalization/memorization boundary (Thm. 4) fits well with the experiments (Fig. 5).

**Comparison with existing grokking frameworks.** Our framework provides a theoretical foundation from first principles (i.e., gradient dynamics) that explains the empirical hypothesis Varma et al. (2023) that "*generalization circuits $\mathcal{C}_{gen}$ is more efficient but learn slower than memorization circuits $\mathcal{C}_{mem}$*". Specifically, we show that the data distribution determines the optimization landscape, which in turn governs which local optima the weights converge into, which lead to the behavior of memorization or generalization. We also show that the initial memorization, or lazy learning (Stage I), has to happen before feature learning (Stage II-III), since the former provides meaningful backpropagated gradient $G_F$ for the latter to start developing. In comparison, (Nanda et al., 2023) also provides a three stage framework of grokking, but mostly from empirical observations. Using our framework, we provide a systematic framework to explain when and why the grokking happens that are consistent with many empirical observations (Sec. E).

Our study goes beyond Neural Tangent Kernel (NTK) (Du et al., 2018; Jacot et al., 2018) and mean field regime (Rubin et al., 2024), in which the hidden weights do not update substantially from initializations. Our framework demonstrates concretely how features emerge from gradient dynamics and multiple key factors that affect the procedure.

**Summary**. Our framework `Li₂` splits the network training process into three stages (**L**azy, **i**ndependent, **i**nteractive), and provides insights into grokking dynamics and feature learning.

- *Explanation of Grokking behavior*. At the beginning of grokking, the lazy learning corresponds to memorization, in which the top layer finds a (temporary) solution to explain the target with the random features. Only after that, the backpropagated gradient $G_F$ becomes meaningful, and causes the hidden layer to learn generalizable, emerging features. Grokking is accelerated with regularization.

- *Emerging features* are the local maxima of an energy function $\mathcal{E}$ that governs the independent stage. These features are more efficient in label prediction than simple memorization.

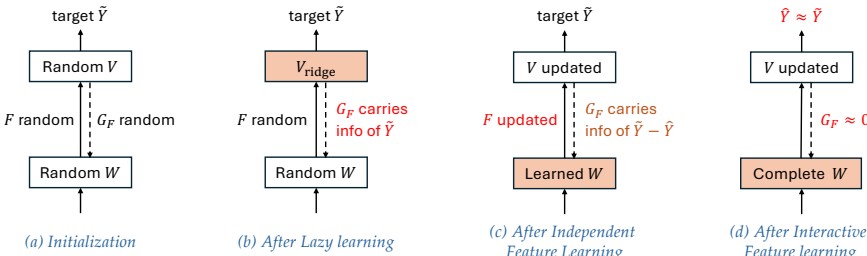

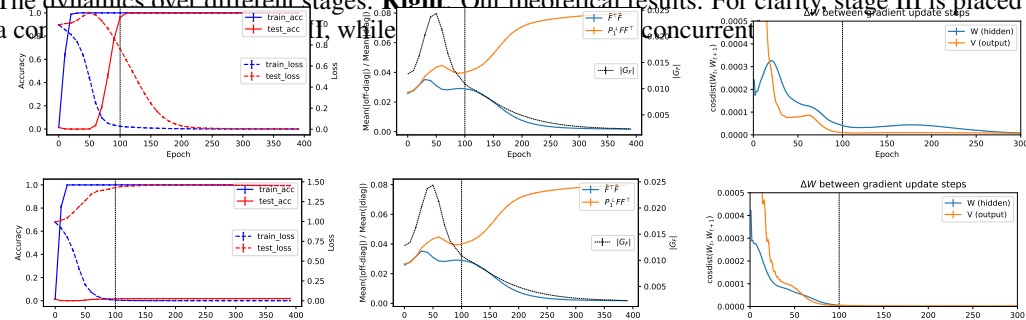

Figure 2: **Left:** The loss curves of training and test sets and their gap (generalization gap). **Middle:** The dynamics over different stages. **Right:** Our theoretical results. For clarity, stage III is placed as a concurrent II, while concurrent

Figure 3: Grokking dynamics on modular addition task with $M = 71$, $K = 2048$, $n = 2016$ (40% training out of $71^2$ samples) with and without weight decay. *Top*: $\eta = 0.0002$ and grokking happens. *Bottom*: $\eta = 0$ and no grokking happens. Weight decay leads to larger $|G_F|$ around epoch 100 and induces grokking behavior. The weights difference $\Delta W$ between consecutive weights at time $t$ and $t + 1$, measured by cosine distance, shows two-stage behaviors: first there is huge update on the output weight $V$, then large update on the hidden weight $W$. Throughout the training, $\tilde{F}^\top \tilde{F}$ and $P_1^\perp F F^\top$ remains diagonal with up to 8% error, validating our analysis (independent feature learning, Sec. 4). Experiments averaged over 15 seeds.

- *Data* governs the landscape of $\mathcal{E}$. Sufficient training data maintain the shape of those generalizable local maxima, while insufficient data lead to non-generalizable local maxima.

- *Scaling Laws of Feature Emergence, Generalization and Memorization* can be derived by inspecting how the landscape changes with the data distribution.

## 2 PROBLEM FORMULATION

We consider a 2-layer network $\hat{Y} = \sigma(XW)V$ and $\ell_2$ loss function on $n$ samples:

$$\min_{V,W} \frac{1}{2} \| P_1^\perp (Y - \hat{Y}) \|_F^2 = \min_{V,W} \frac{1}{2} \| P_1^\perp (Y - \sigma(XW)V) \|_F^2 \tag{1}$$

where $P_1^\perp := I - \mathbf{1}\mathbf{1}^\top / n$ is the zero-mean projection matrix along the sample dimension, $Y \in \mathbb{R}^{n \times M}$ is a label matrix (each row is a one-hot vector), $X = [\mathbf{x}_1, \mathbf{x}_2, \ldots, \mathbf{x}_n]^\top \in \mathbb{R}^{n \times d}$ is the data matrix, $V \in \mathbb{R}^{K \times M}$ and $W \in \mathbb{R}^{d \times K}$ are the weight matrices of the last layer and hidden layer, respectively. $\sigma$ is the nonlinear activation function.

In the following, we show that grokking is a consequence of "leaked" backpropagated gradient $G_F$.

## 3 STAGE I: LAZY LEARNING (OVERFITTING)

Let $F = \sigma(XW)$ be the activation of the hidden layer and $\tilde{F} = P_1^\perp F$ be the zero-mean version of it. Similarly define $\tilde{Y} = P_1^\perp Y$. We first write down the backpropagated gradient $G_F$ sent to the hidden layer:

$$G_F = -\frac{\partial J}{\partial F} = P_1^\perp (Y - FV)V^\top \tag{2}$$

### 3.1 ANALYSIS OF THE BACKPROPAGATED GRADIENT $G_F$

At the beginning of the training, both $W$ and $V$ are initialized with independent zero-mean random variables. Therefore, the backpropagated gradient $G_F$ is pure random noise. Over time, the hidden

activation $F$ is mostly unchanged, and only the output layer $V$ learns. In this case, $F$ can be treated as fixed during this stage of learning, and we can prove the following properties of $G_F$ (Sec. H):

**Proposition 1.** *If $\tilde{F}$ is fixed and is full column rank, entries of $V(0)$ is initialized from normal distribution $N(0, \alpha^2)$ with $0 < \alpha \ll 1$, then $\|G_F(0)\|_F = O(\epsilon\sqrt{KM})$ and the backpropagated gradient $G_F$ is dominated by the term $\tilde{Y}\tilde{Y}^\top F$ at initial time stamps:*

$$G_F(t) = t\tilde{Y}\tilde{Y}^\top \tilde{F} + O(\alpha) + O(\alpha t) + O(t^2) \tag{3}$$

*and converges exponentially to the following fixed point when $V = V_{ridge} = (\tilde{F}^\top \tilde{F} + \eta I)^{-1}\tilde{F}^\top \tilde{Y}$:*

$$G_F(+\infty) = \eta(\tilde{F}\tilde{F}^\top + \eta I)^{-1}\tilde{Y}\tilde{Y}^\top \tilde{F}(\tilde{F}^\top \tilde{F} + \eta I)^{-1} \tag{4}$$

$G_F$ **at initial phase**. The proposition suggests that for small top layer initialization (measured by $\alpha$), $\|G_F\|$ will first increase from $O(\alpha)$ to $O(1)$ and then converge exponentially to $O(\eta)$. Fig. 3 shows that this is indeed the case for $\|G_F\|$, regardless whether grokking happens or not.

$G_F$ **at later phase**. The structure of $G_F(+\infty)$ is revealed by the following lemma:

**Lemma 1** (Structure of backpropagated gradient $G_F$). *Assume that (1) entries of $W$ follow standard normal distribution $N(0, 1)$, (2) $\|\mathbf{x}_i\|_2 = const$, (3) $\|\mathbf{x}_i^\top \mathbf{x}_{i'} - \rho\|_2 \le \epsilon$ for all $i \ne i'$ and (4) large width $K$, then both $\tilde{F}^\top \tilde{F}$ and $\tilde{F}\tilde{F}^\top$ becomes a multiple of identity and Eqn. 4 becomes:*

$$G_F(+\infty) = \frac{\eta}{(Kc_1 + \eta)(nc_2 + \eta)}\tilde{Y}\tilde{Y}^\top \tilde{F} + O(K^{-1}\epsilon) \tag{5}$$

*where $c_1, c_2 > 0$ are constants related to nonlinearity. When $\eta$ is small, we have $G_F \propto \eta\tilde{Y}\tilde{Y}^\top \tilde{F}$. Note that the input features and/or weights can be scaled and what changes is $c_1$ and $c_2$.*

Interestingly, in both cases, we see that $G_F$ contains a key term $\tilde{Y}\tilde{Y}^\top \tilde{F}$. As we will see, it plays a critical role in feature learning. From Eqn. 5, it is clear that if $K \to +\infty$, then $G_F(+\infty) \to 0$ and there is no feature learning (i.e., NTK regime). Here we study the case when $K$ is large (so that Eqn. 5 is valid) but not too large so that feature learning happens.

### 3.2    Zero-init Accelerates the Feature Learning Process

Given the analysis in Sec. 3 of Stage I, it becomes tempting to think that if we initialize the top-layer weight $V$ to zero, then at the initial stage the backpropagated gradient $G_F(t)$ carries only the signal term $\tilde{Y}\tilde{Y}^\top F$ according to Eqn. 3, and the feature learning process should be accelerated. We named this new approach as `zero-init`.

Experiments show that this is indeed the case. Fig. 15, 16 and 17 show that for $M = 41, 89, 127$, `zero-init` accelerates the feature learning process compared to normal initialization.

In multi-layer setting, the boost brought up by `zero-init` is larger (Fig. 18), in particular when data are scarce and the feature learning process becomes slow. The gap can be as large as $10\times$ (e.g., 100 epochs versus 1000 epochs) when training on MSE loss.

All the experiments are in Appendix J.3.

## 4    Stage II: Independent feature learning

### 4.1    The energy function $\mathcal{E}$

Now let us explore the feature learning process with the help of $G_F$. Let $W = [\mathbf{w}_1, \mathbf{w}_2, \dots, \mathbf{w}_K]$ where $\mathbf{w}_j \in \mathbb{R}^d$ is the weight vector of $j$-th node, and $F = [\mathbf{f}_1, \mathbf{f}_2, \dots, \mathbf{f}_K]$ where $\mathbf{f}_j = \sigma(X\mathbf{w}_j) \in \mathbb{R}^n$ is the activation of $j$-th node. For $G_F \propto \tilde{Y}\tilde{Y}^\top \tilde{F}$, as the structure shown in both initial stage (Eqn. 3) and later stage (Eqn. 5), the $j$-th column $\mathbf{g}_j$ of $G_F$ is only dependent on $j$-th node $\mathbf{w}_j$, and thus we can decouple the dynamics into $K$ independent ones, each corresponding to a single node:

$$\dot{\mathbf{w}}_j = X^\top D_j\mathbf{g}_j, \quad \mathbf{g}_j \propto \tilde{Y}\tilde{Y}^\top \sigma(X\mathbf{w}_j) \tag{6}$$

where $D_j = \text{diag}(\sigma'(X\mathbf{w}_j))$ is the diagonal gating matrix of $j$-th node. Note that $\tilde{Y}^\top F = \tilde{Y}^\top \tilde{F}$ since $P_1^\perp$ is idempotent. A critical observation here is that Eqn. 6 actually corresponds to the *gradient ascent* dynamics of the energy function $\mathcal{E}$.

**Theorem 1** (The energy function $\mathcal{E}$ for independent feature learning). *The dynamics (Eqn. 6) of independent feature learning is exactly the gradient ascent dynamics of the energy function $\mathcal{E}$ w.r.t. $\mathbf{w}_j$, a nonlinear canonical-correlation analysis (CCA) between the input $X$ and target $\tilde{Y}$:*

$$\mathcal{E}(\mathbf{w}_j) = \frac{1}{2}\|\tilde{Y}^\top \sigma(X\mathbf{w}_j)\|_2^2 \tag{7}$$

Therefore, the feature learned for each node $j$ is the one that maximizes the energy function $\mathcal{E}(\mathbf{w}_j)$. Since Eqn. 6 can be unbounded, in the following, we put an additional constraint that $\|\mathbf{w}_j\|_2 = 1$ due to weight decay regularization. Note that (Tian, 2023) also arrives at an energy function when studying feature learning in the context of contrastive loss, the resulting function is abstract and difficult to interpret its structure of its local maxima. Here the structure is much clearer, which we will explore below.

## 4.2 GROUP ARITHMETIC TASKS

To demonstrate a concrete example in solving the energy function $\mathcal{E}$, we consider *group arithmetic tasks*, i.e., for group $H$, the task is to predict $h = h_1 h_2$ given $h_1, h_2 \in H$. One example is the modular addition task $h_1 h_2 = h_1 + h_2 \mod M$, which has been extensively studied in grokking (Power et al., 2022; Gromov, 2023; Huang et al., 2024; Tian, 2025).

**Definition of group**. Here $h_1 h_2$ is the *group operation* of two elements $h_1$ and $h_2$ in the group $H$. The group operation satisfies a few properties: (1) the group operation is associative, i.e., $(h_1 h_2)h_3 = h_1(h_2 h_3)$ for all $h_1, h_2, h_3 \in H$, (2) there exists an identity element $e$ such that $he = eh = h$ for all $h \in H$, (3) for each $h \in H$, there exists an inverse $h^{-1}$ such that $hh^{-1} = h^{-1}h = e$. Note that the group operation is not commutative in general, i.e., $h_1 h_2 \neq h_2 h_1$ for some $h_1, h_2 \in H$. If the operation is commutative, then we call the group *Abelian*. Modular addition is an example of Abelian group.

**The task**. We represent the group elements by one-hot vectors: each data sample $\mathbf{x}_i \in \mathbb{R}^{2M}$ is a concatenation of two $M$-dimensional one-hot vectors $(\mathbf{e}_{h_1[i]}, \mathbf{e}_{h_2[i]})$ where $h_1[i]$ and $h_2[i]$ are the indices of the two one-hot vectors. The output is also a one-hot vector $\mathbf{y}_i = \mathbf{e}_{h_1[i]h_2[i]}$, where $1 \leq i \leq n = M^2$. Here the class number $M = |H|$ is the size of the group.

**A crash course of group representation theory**. A mapping $\rho(h) : H \mapsto \mathbb{C}^{d \times d}$ is called a *group representation* if the group operation is compatible with matrix multiplication: $\rho(h_1)\rho(h_2) = \rho(h_1 h_2)$ for any $h_1, h_2 \in H$. Let $R_h \in \mathbb{R}^{M \times M}$ be the *regular representation* of group element $h$ so that $\mathbf{e}_{h_1 h_2} = R_{h_1}\mathbf{e}_{h_2}$ for all $h_1, h_2 \in H$, and $P \in \mathbb{R}^{M \times M}$ be the group inverse operator so that $P\mathbf{e}_h = \mathbf{e}_{h^{-1}}$. Note that $P^2 = I$ and $P^\top = P^{-1} = P$.

**A tale of two kind of "representations"**. We often refer the activation vectors in neural network as "neural representations", which means that we use high-dimensional vectors to represent certain semantic concepts (e.g., word2vec (Mikolov et al., 2013)). For groups, the group element corresponds to "transformation" / "action" that acts on one object and turn it to another (i.e., rotation of a vector). Therefore, group representations often take the form of matrices (e.g., permutation / rotation matrix).

**The decomposition of group representation**. The representation theory of finite group (Fulton & Harris, 2013; Steinberg, 2009) says that the regular representation $R_h$ admits a decomposition into complex *irreducible* representations (or ***irreps***):

$$R_h = Q \left( \bigoplus_{k=0}^{\kappa(H)} \bigoplus_{r=1}^{m_k} C_k(h) \right) Q^* \tag{8}$$

where $\kappa(H)$ is the number of nontrivial irreps (i.e., not all $h$ map to identity), $C_k(h) \in \mathbb{C}^{d_k \times d_k}$ is the $k$-th irrep block of $R_h$, $Q$ is the unitary matrix (and $Q^*$ is its conjugate transpose) and $m_k$ is the multiplicity of the $k$-th irrep. This means that in the decomposition of $R_h$, there are $m_k$ copies of $d_k$-dimensional irrep, and these copies are isomorphic to each other. So the $k$-th ***irrep subspace*** $\mathcal{H}_k$ has dimension $m_k d_k$.

For regular representation $\{R_h\}$, one can prove that $m_k = d_k$ for all $k$ and thus $|H| = M = \sum_k d_k^2$. For Abelian group, all complex irreps are 1d (i.e., Fourier bases). One may also choose to do the

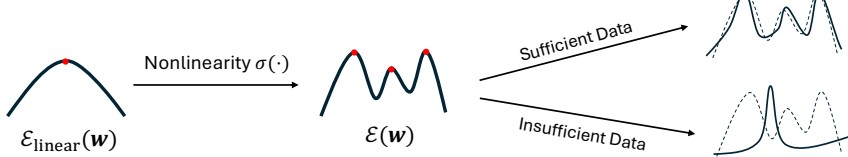

Figure 4: Change of the landscape of the energy function $\mathcal{E}$ (Thm. 1). **Left:** $\mathcal{E}$ with linear activation reduces to simple eigen-decomposition and only have one global maxima. **Middle:** With nonlinearity, the energy landscape now has multiple strict local maxima, each corresponds to a feature (Thm. 2). More importantly, these features are more efficient than memorization in target prediction (Thm. 3). **Right:** With sufficient training data, the landscape remains stable and we can recover these (generalizable) features (Thm. 4), with insufficient data, the landscape changes substantially and local maxima becomes memorization (Thm. 5).

decomposition in real domain. In this case, a pair of 1d complex irreps will become a 2d real irrep. For example, $e^{i\theta}$ and $e^{-i\theta}$ becomes a 2d matrix $[\cos(\theta), -\sin(\theta); \sin(\theta), \cos(\theta)]$.

### 4.3 LOCAL MAXIMA OF THE ENERGY FUNCTION

Now we study the local maxima of $\mathcal{E}$. With the decomposition, we can completely characterize the local maxima of the energy $\mathcal{E}$ with group inputs, even that $\mathcal{E}(\mathbf{w})$ is nonconvex.

**Theorem 2** (Local maxima of $\mathcal{E}$ for group input). *For group arithmetics tasks with $\sigma(x) = x^2$, $\mathcal{E}$ has multiple local maxima $\mathbf{w}^* = [\mathbf{u}; \pm P\mathbf{u}]$. Either it is in a real irrep of dimension $d_k$ (with $\mathcal{E}^* = M/8d_k$ and $\mathbf{u} \in \mathcal{H}_k$), or in a pair of complex irrep of dimension $d_k$ (with $\mathcal{E}^* = M/16d_k$ and $\mathbf{u} \in \mathcal{H}_k \oplus \mathcal{H}_{\bar{k}}$). These local maxima are not connected. No other local maxima exist.*

Note that our proof can be extended to more general nonlinearity $\sigma(x) = ax + bx^2$ with $b > 0$ since linear part will be cancelled out due to zero-mean operators. We can show that local maxima of $\mathcal{E}$ are flat, allowing moving around without changing $\mathcal{E}$:

**Corollary 1** (Flatness of local maxima of $\mathcal{E}$ for group input). *Local maxima of $\mathcal{E}$ for group arithmetics tasks with $|H| = M > 2$ are flat, i.e., at least one eigenvalue of its Hessian is zero.*

We can apply the above theorem to the popular modular addition task which is an Abelian group. The resulting representation is Fourier bases.

**Corollary 2** (Modular addition). *For modular addition with odd $M$, all local maxima are single frequency $\mathbf{u}_k = a_k[\cos(km\omega)]_{m=0}^{M-1} + b_k[\sin(km\omega)]_{m=0}^{M-1}$ where $\omega := 2\pi/M$ with $\mathcal{E}^* = M/16$. For even $M$, $\mathbf{u}_{M/2} \propto [(-1)^m]_{m=0}^{M-1}$ has $\mathcal{E}^* = M/8$. Different local maxima are disconnected.*

**Role played by the nonlinearity**. Note that if $\sigma(x)$ is linear, then there is no local maxima, but only a global one, which is the maximal eigenvector of $X^\top \tilde{Y} \tilde{Y}^\top X$. This corresponds to Linear Discriminative Analysis (LDA) (Balakrishnama & Ganapathiraju, 1998) that finds directions that maximally separate the class-mean vectors. For group arithmetics tasks, for each target $h = h_1 h_2$, each group element ($h_1$ and $h_2$) appears once and only once, the class-mean vectors are identical and thus LDA fails to identify any meaningful directions. With nonlinearity, the learned $\mathbf{w}$ has clear meanings.

**Meaning of the learned features**. First, the learned representation can offer a more efficient reconstruction of the target (see Thm. 3) than simple memorization of all $M^2$ pairs. Second, learned representations naturally contain useful invariance. For example, some irreps of the cyclic group of $\mathbb{Z}_{15}$ behave like its subgroup $\mathbb{Z}_3$ and $\mathbb{Z}_5$, by mapping its element $h$ to $\mathrm{div}(h, 3)$ and $\mathrm{div}(h, 5)$. If we regard $h$ to be controlled by two hidden factors, then these features lead to focusing on one factor and invariant to others. More importantly, they emerge automatically without explicit supervision.

### 4.4 REPRESENTATION POWER OF LEARNED FEATURES

With Thm. 2, we know that each node of the hidden layers will learn various representations. The question is whether they are sufficient to reconstruct the target $\tilde{Y}$ and how efficient they are.

**Theorem 3** (Target Reconstruction). *Assume (1) $\mathcal{E}$ is optimized in complex domain $\mathbb{C}$, (2) for each irrep $k$, there are $m_k^2 d_k^2$ pairs of learned weights $\mathbf{w} = [\mathbf{u}; \pm P\mathbf{u}]$ whose associated rank-1 matrices $\{\mathbf{u}\mathbf{u}^*\}$ form a complete bases for $\mathcal{H}_k$ and (3) the top layer $V$ also learns with $\eta = 0$, then $\hat{Y} = \tilde{Y}$.*

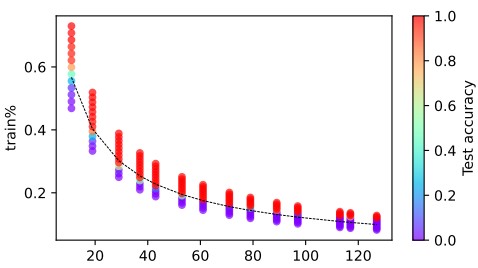 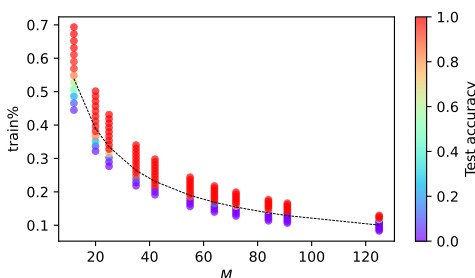

Figure 5: Generalization/memorization phase transition in modular addition tasks. When $M$ grows, the training data ratio $p = n/M^2$ required to achieve generalization decreases. This coincides with Thm. 4 which predicts $p \sim M^{-1} \log M$ (dotted line). We use learning rate 0.0005, weight decay 0.0002 and $K = 2048$. Results averaged over 20 seeds. **Left:** Simple cyclic group $\mathbb{Z}_M$ for prime $M$. **Right:** $\mathbb{Z}_M$ for composite $M$. Other groups (product and non-Abelian) are shown in Appendix J.1.

From the theorem, we know that $K = 2 \sum_{k \neq 0} m_k^2 d_k^2 \leq 2 \left[ (M - \kappa(H))^2 + \kappa(H) - 1 \right]$ suffice. In particular, for Abelian group, $\kappa(H) = M - 1$ and $K = 2M - 2$. This is much more efficient than a pure memorization solution that would require $M^2$ nodes, i.e., each node memorizes a single pair $(h_1, h_2) \in H \times H$.

**Assumptions of the theorem**. Training the model in an end-to-end manner automatically satisfies the assumption (3). If we initialize the weights randomly, then with high probability, the resulting $\mathbf{u}$ are not collinear and the assumption (2) can be satisfied. For assumption (1), interestingly, no such theorem can be stated in the real domain $\mathbb{R}$, because in $\mathbb{R}$, the subspace of orthogonal matrices, which group representations belong to, is not covered by the subspace of symmetric matrices spanned by $\{\mathbf{u}\mathbf{u}^\top\}$. In contrast, in the complex domain $\mathbb{C}$, the subspace of unitary matrices can be represented by Hermitian matrices. Does that mean that the real representation is not that useful? Not really. If we change $\mathbf{w} = [\mathbf{u}; \pm P\mathbf{u}]$ slightly to $\mathbf{w} = [\mathbf{u}; \pm P\mathbf{u}']$ in which $\mathbf{u}'$ is a small perturbation of $\mathbf{u}$, then Thm. 3 holds for real solutions. This happens in the stage III when end-to-end backpropagation refines the representation. Fig. 12 shows that using complex weights still works and may grok faster.

## 4.5 THE SCALING LAWS OF THE BOUNDARY OF MEMORIZATION AND GENERALIZATION

While Thm. 2 shows the nice structure of local maxima (and features learned), it requires training on all $n = M^2$ pairs of group elements. One may ask whether these representations can still be learned if training on a subset. The answer is yes, by checking the stability of the local maximum.

**Theorem 4** (Amount of samples to maintain local optima). *If we select $n \gtrsim d_k^2 M \log(M/\delta)$ data sample from $H \times H$ uniformly at random, then with probability at least $1 - \delta$, the empirical energy function $\hat{\mathcal{E}}$ keeps local maxima for $d_k$-dimensional irreps (Thm. 2).*

The theorem above says that we do not need all $M^2$ samples, but only $O(M \log M)$ samples suffice to learn these features, which will generalize to unseen data according to Thm. 3. Fig. 5 demonstrates that the empirical results closely match the theoretical prediction, and there is a clear phase transition around the boundary (test accuracy $0 \rightarrow 1$), where the training data ratio $p := n/M^2 = O(M^{-1} \log M)$.

**Memorization**. On the other hand, we can also construct cases when memorization is the only local maximum of $\mathcal{E}$. This happens when we only collect samples for one target $h$ but missing others, and diversity is in question.

**Theorem 5** (Memorization solution). *Let $\phi(x) := \sigma'(x)/x$ and assume $\sigma'(x) > 0$ for $x > 0$. For group arithmetic tasks, suppose we only collect sample $(g, g^{-1}h)$ for one target $h$ with probability $p_g$. Then the global optimal of $\mathcal{E}$ is a memorization solution, either (1) a focused memorization $\mathbf{w} = \frac{1}{\sqrt{2}}(\mathbf{e}_{g^*}, \mathbf{e}_{g^{*-1}h})$ for $g^* = \arg\max p_g$ if $\phi$ is nondecreasing, or (2) a spreading memorization with $\mathbf{w} = \frac{1}{2} \sum_g s_g [\mathbf{e}_g, \mathbf{e}_{g^{-1}h}]$, if $\phi$ is strictly decreasing. Here $s_g = \phi^{-1}(2\lambda/p_g)$ and $\lambda$ is determined by $\sum_g s_g^2 = 2$. No other local optima exist.*

We can verify that power activations (e.g., $\sigma(x) = x^2$) lead to focused memorization, while more practical ones (e.g., ReLU, SiLU, Tanh and Sigmoid) lead to spreading memorization. We leave it for future work whether this property leads to better results in large scale settings.

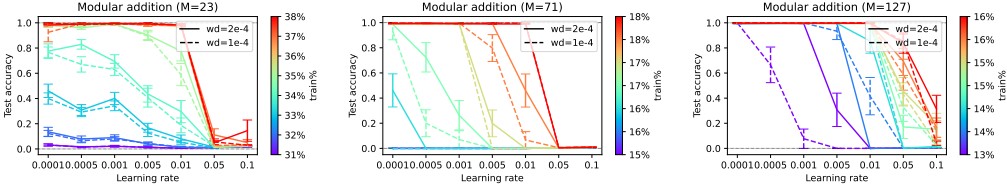

Figure 6: Phase transition from generalizable (gsol) to non-generalizable solutions (ngsol) in modular addition tasks ($M = 23, 71, 127$) with $K = 1024$. Around this critical region, small learning rate more likely lead to gsol, due to the fact that small learning rate keeps the trajectory staying within the basin towards gsol, while large learning rate converges to solutions with higher $\mathcal{E}$ (Fig. 7). Results averaged over 15 seeds.

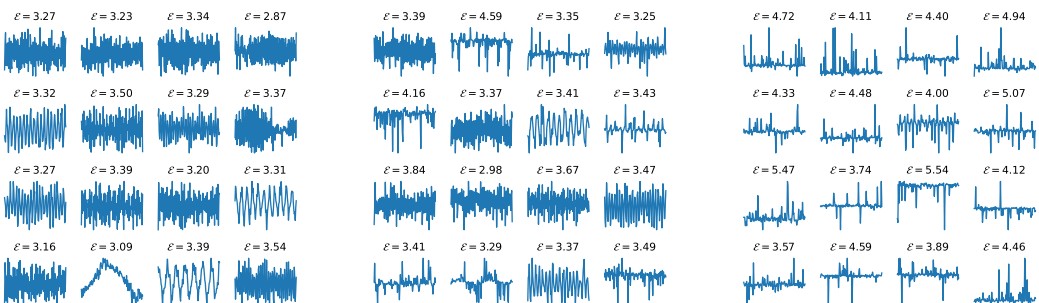

Figure 7: In small data regime of modular addition with $M = 127$ and $n = 3225$ (20% training out of $127^2$ samples), Adam optimizer with small learning rate ((0.001, left) and (0.002, middle)) leads to generalizable solutions (Fourier bases) with low $\mathcal{E}$, while with large learning rate (0.005, right), Adam found non-generalizable solutions (e.g., memorization) with much higher $\mathcal{E}$.

**Boundary of generalization and memorization (*semi-grokking* (Varma et al., 2023))**. In between the two extreme cases, local maxima of both memorization and generalization may co-exist. In this case, small learning rate keeps the optimization within the attractive basin and converges to gsol, while large learning rate leads to ngsol which has better energy $\mathcal{E}$ (Fig. 7).

Our analysis fits well with the observed empirical behaviors that there exists a critical data size (Varma et al., 2023) or more precisely a critical data ratio (Wang et al., 2024a; Abramov et al., 2025; Liu et al., 2022), above which the grokking suddenly leads to generalization. Our framework Li₂ gives a theory-backed explanation: different data distribution leads to different landscapes, which gives either generalization or memorization local maxima that the weights will fall into. It also explains the *ungrokking* phenomenon: a grokked model can move back to memorization when continues to train on a smaller dataset (Varma et al., 2023), and is consistent with (Nguyen & Reddy, 2025) that shows that task diversity is important for generalization. The empirical hypothesis (e.g., "memorization/generalization circuits") now becomes a natural consequence of changed landscape after training.

## 5 STAGE III: INTERACTIVE FEATURE LEARNING

The starting point of Stage II is to simplify the exact backpropagated gradient $G_F = P_\eta \tilde{Y} \tilde{Y}^\top \tilde{F} B$ (Eqn. 4) with $B := (\tilde{F}^\top \tilde{F} + \eta I)^{-1}$ to $G_F \propto \eta \tilde{Y} \tilde{Y}^\top F$, by two approximations: (1) $B \propto I$, and (2) $P_\eta \propto \eta I$. The two approximations are valid due to Thm. 1 when the hidden weights $W$ is randomly initialized. When training continues, $W$ evolves from random initialization and the conditions may not hold anymore. In this section we put them back and study their behaviors.

## 5.1 REPULSION OF SIMILAR FEATURES

We first study the effect of $B$, which leads to interplay of hidden nodes. Over the training, the activations of two nodes can be highly correlated and the following theorem shows that similar features leads to repulsion.

**Theorem 6** (Repulsion of similar features). *The $j$-th column of $\tilde{F}B$ is given by:*

$$[\tilde{F}B]_j = b_{jj}\tilde{\mathbf{f}}_j + \sum_{l=1}^{K} b_{jl}\tilde{\mathbf{f}}_l \tag{9}$$

*where* $\operatorname{sign}(b_{jl}) = -\operatorname{sign}(\tilde{\mathbf{f}}_j^\top P_{\eta,-jl}\tilde{\mathbf{f}}_l)$. *Here* $P_{\eta,-jl} := I - \tilde{F}_{-jl}(\tilde{F}_{-jl}^\top \tilde{F}_{-jl} + \eta I)^{-1}\tilde{F}_{-jl}^\top$ *is a projection matrix constructed from* $\tilde{F}_{-jl}$, *which is* $\tilde{F}$ *excluding the $l$-th and $j$-th columns.*

**Remark**. Intuitively, if $\tilde{\mathbf{f}}_j$ and $\tilde{\mathbf{f}}_l$ are similar, then $b_{jl}$ will be negative and the resulting $j$ and $l$ columns of $\tilde{F}B$ will be pushed away from each other and vise versa.

## 5.2 TOP-DOWN MODULATION

Over the training process, it is possible that some local optima are learned first while others learned later. When the representations are learned partially, the backpropagation offers a mechanism to focus on missing pieces, by changing the landscape of the energy function $\mathcal{E}$.

**Theorem 7** (Top-down Modulation). *For group arithmetic tasks with $\sigma(x) = x^2$, if the hidden layer learns only a subset $\mathcal{S}$ of irreps, then the backpropagated gradient $G_F \propto (\Phi_\mathcal{S} \otimes \mathbf{1}_M)(\Phi_\mathcal{S} \otimes \mathbf{1}_M)^* F$ (see proof for the definition of $\Phi_\mathcal{S}$), which yields a modified $\mathcal{E}_\mathcal{S}$ that only has local maxima on the missing irreps $k \notin \mathcal{S}$.*

## 5.3 DIVERSITY ENHANCEMENT WITH MUON

In addition to the mechanism above, certain optimizers (e.g., Muon optimizer (Jordan et al., 2024)) can also address such issue, by boosting the weight update direction that are underrepresented, enforcing diversity of nodes. While evidence (Tveit et al., 2025) and analysis exist (Shen et al., 2025) to show that Muon has advantages over other optimizers, to our best knowledge, we are the first to analyze it in the context of feature learning.

Recall that the Muon optimizer converts the gradient $G_W = U_{G_W} D V_{G_W}^\top$ (its SVD decomposition) to $G'_W = U_{G_W} V_{G_W}^\top$ and update the weight $W$ accordingly (i.e., $\dot{W} \propto G'_W$). We first show that when Muon is applied to independent feature learning on each $\mathbf{w}_j$ to make them coupled, it still gives the correct answers to the original optimization problems.

**Lemma 2** (Muon optimizes the same as gradient flow). *Muon finds ascending direction to maximize the joint energy function $\mathcal{E}_{\text{joint}}(W) = \sum_j \mathcal{E}(\mathbf{w}_j)$ and has stationary points iff the original gradient $G_W$ vanishes.*

Now we show that Muon optimizer can rebalance the gradient updates.

**Theorem 8** (Muon rebalances gradient updates). *Consider the following dynamics (Tian, 2023):*

$$\dot{\mathbf{w}} = A(\mathbf{w})\mathbf{w}, \qquad \|\mathbf{w}\|_2 \leq 1 \tag{10}$$

*where $A(\mathbf{w}) := \sum_l \lambda_l(\mathbf{w})\zeta_l\zeta_l^\top$. Assume that (1) $\{\zeta_l\}$ form orthonormal bases, (2) for $\mathbf{w} = \sum_l \alpha_l\zeta_l$, we have $\lambda_l(\mathbf{w}) = \mu_l\alpha_l$ with $\mu_l \leq 1$, and (3) $\{\alpha_l\}$ is initialized from inverse-exponential distribution with $\operatorname{CDF}(x) = \exp(-x^{-a})$ with $a > 1$. Then*

- ***Independent feature learning.*** $\Pr[\mathbf{w} \to \zeta_l] = p_l := \mu_l^a / \sum_l \mu_l^a$. *Then the expected #nodes to get all local maxima is $T_0 \geq \max\left(1/\min_l p_l, \sum_{l=1}^{L} 1/l\right)$.*

- ***Muon guiding.*** *If we use Muon optimizer to optimize $K$ nodes sequentially, then the expected #nodes to get all local maxima is $T_a = 2^{-a}T_0 + (1 - 2^{-a})L$. For large $a$, $T_a \sim L$.*

The intuition here is that once some weight vectors have "occupied" a local maximum, say $\zeta_m$, their gradients point to the same direction (before projecting onto the unit sphere $\|\mathbf{w}\|_2 = 1$), and the gradient correction of Muon will discount that component from gradients of currently optimized weight vectors, and keeping them away from $\zeta_m$. In this way, Muon pressed novel gradient directions and thus encourages exploration. Fig. 9 shows that Muon is effective with limited number of hidden nodes $K$.

Note that Eqn. 10 is closely related to $\mathcal{E}$, under the assumption of homogeneous/reversible activation, i.e., $\sigma(x) = C\sigma'(x)x$ with a constant $C$ (Zhao et al., 2024; Tian et al., 2020). In such setting, Eqn. 6 is related to the gradient dynamics with a PSD matrix $A(\mathbf{w}) = X^\top D(\mathbf{w})\tilde{Y}\tilde{Y}^\top D(\mathbf{w})X$.

## 6 EXTENSION TO DEEPER ARCHITECTURES

The above analysis and the definition of the energy function $\mathcal{E}$ can be extended to deeper architectures. Consider a multi-layer network with $L$ hidden layers, $F_l = \sigma(F_{l-1}W_l)$ with $F_0 = X$ and $\hat{Y} = F_L V$. For notation brevity, let $G_l := G_{F_l}$. Let's see how the gradient backpropagated and how the learning fits to our framework (Fig. 1).

*Stage I*. Stage I does not change since $F_L$ is still a random representation. Then when $V$ starts to learn and converges, the backpropagated gradient $G_L$ now carries meaningful information: $G_L \propto \tilde{Y}\tilde{Y}^\top F_L$ (Eqn. 5), which initiates Stage II.

*Stage II*. We assume homogeneous activation $\sigma(x) = C\sigma'(x)x$. For the next layer $L-1$, we have:

$$G_{L-1} = D_L G_L W_L^\top = D_L(\tilde{Y}\tilde{Y}^\top F_L)W_L^\top = (D_L\tilde{Y}\tilde{Y}^\top D_L)F_{L-1}(W_L W_L^\top) \qquad (11)$$

since $W_L$ is randomly initialized, we have $W_L W_L^\top \approx I$ and thus $G_{L-1} \propto D_L\tilde{Y}\tilde{Y}^\top D_L F_{L-1}$.

Doing this iteratively gives $G_l \propto \left(\tilde{D}_{l+1}\tilde{Y}\tilde{Y}^\top \tilde{D}_{l+1}\right) F_l$, where $\tilde{D}_l := \prod_{m=l}^{L} D_m$. Note that these $D$ matrices are essentially reweighing/pruning samples randomly, since right now all $\{W_l\}$ are random except for $V$. Now the lowest layer receives meaningful backpropagated gradient $G_1$ that is related to the target label, and it also exposes to input $X$. Therefore, the learning starts from there. Once layer $l$ learns decent representation, layer $l+1$ receives meaningful input $F_l$ and starts to learn, etc. When layer $l$ is learning, layer $l' > l$ do not learn since their input $F_{l'}$ remains random noise.

From this analysis, we can also see why residual connection helps. In this case, $G_{\text{res},1} = \sum_{l=1}^{L} G_l$, in which $G_L$ is definitely a much cleaner and stronger signal, compared to $G_1$ which undergoes many random reweighing and pruning of samples.

*Stage III*. Once the activation $F_l$ becomes meaningful, top-down modulation could happen (similar to Thm. 7) among nearby layers so that low-level features can be useful to support high-level representations. We leave the detailed analysis for future work.

## 7 CONCLUSION, LIMITATIONS, AND FUTURE WORK

We develop a principled mathematical framework $\texttt{Li}_2$ for the grokking dynamics in 2-layer networks. $\texttt{Li}_2$ identifies three stages, lazy learning, independent feature learning, and interactive feature learning, each marked by distinct structure of backpropagated gradient $G_F$. Weight decay enables hidden nodes to independently extract label-related features via a well-defined energy function $\mathcal{E}$, before cross interactions consolidate them. The analysis clarifies how hyperparameters like weight decay, learning rate, and sample size shape grokking, and offers first principles insight into why modern optimizers (e.g., Muon) are effective. We also extend our framework to deeper networks.

**Limitations**. While the derivation of energy $\mathcal{E}$ is applicable to any input, analysis of its local maxima relies on restrictive assumption of group structure of the input. Also our analysis does not include the transition time between consecutive learning stages. We leave them for future work.

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

## A    DISCLOSURE OF LLM USAGE

We have used SoTA LLMs extensively to brainstorm ideas to prove mathematical statements presented in the paper. Specifically, we setup research directions, provide problem setup and intuitions, proposes statements for LLM to analyze and prove, points out key issues in the generated proofs, adjust the statements accordingly and iterate. We also have done extensive experiments to verify the resulting statements. Many proofs proposed by LLMs are incorrect in subtle ways and requires substantial editing and correction. We have carefully revised all the proofs presented in the work, and take full accountability for their correctness.

## B    ETHICS STATEMENT

This work is about investigating various theoretical and empirical properties of neural networks. We do not rely on any sensitive or proprietary data, nor do we use any existing open source models that may produce harmful contents.

## C    REPRODUCIBILITY STATEMENT

All datasets used in this work can be generated synthetically. Models are pretrained from scratch with very small amount of compute. We will release code to support full Reproducibility.

## D    RELATED WORKS

**Explanation of Grokking**. Multiple explanations of grokking exist, e.g., competition of generalization and memorization circuits (Merrill et al., 2023), a shift from lazy to rich regimes Kumar et al. (2024), etc. Dynamics of grokking is analyzed in specific circumstance, e.g., for clustering data (Xu et al., 2023), linear network (Dominé et al., 2024), etc. In comparison, our work studies the full dynamics of feature emergence driven by backpropagation in group arithmetic tasks for deep nonlinear networks, and provide a systematic mathematical framework about what and how features emerge and a scaling law about when the transition between memorization and generalization happens.

**Usage of group structure**. Recent work leverages group theory to study the structure of final grokked solutions (Tian, 2025; Morwani et al., 2023; Shutman et al., 2025). None of them tackle the dynamics of grokking in the presence of the underlying structure of the data as we do.

**Scaling laws of memorization and generalization**. Previous works have identified scaling laws for memorization/generalization (Nguyen & Reddy, 2025; Wang et al., 2024a; Abramov et al., 2025; Doshi et al., 2023) without systematic theoretical explanation. Our work models such transitions as whether generalizable local optima remain stable under data sampling, and provide theoretical framework from first principles.

**Feature learning**. Previous works treats the NTK as a holistic object and study how it moves away from lazy regime, e.g., it becomes more correlated with task-relevant directions (Kumar et al., 2024; Ba et al., 2022; Damian et al., 2022), becomes adapted to the data (Rubin et al., 2025; Karp et al., 2021), etc. In contrast, our work focuses on explicit learning dynamics of individual features, their interactions, and the transition from memorization to generalization with more samples.

## E    WHEN DOES GROKKING HAPPEN?

We first want to clarify that grokking (i.e., delayed generalization) is a phenomenon/observation that may suggest multiple underlying situations. When we observe "grokking happens", it is often the case that the training gets stuck in lazy learning for a while before moving to feature learning. On the other hand, when we observe "grokking does not happen", then we may fall into one of the three very different situations: (1) training simply gets stuck in lazy training, (2) feature learning happens very early and the gap between training and test performance is always small, or (3) feature learning has converged to non-generalizable solutions due to e.g., insufficient or noisy data, and thus test performance is never improved. Note that (2) is a very good outcome while (1) and (3) are bad ones.

Therefore, it would be tricky to directly relate the hyperparameters with grokking behaviors. Instead, using $\texttt{Li}_2$ framework, we focus on relating them with the backpropagated gradient $G_F$ and the underlying feature learning process, which is more fundamental. Only when the $G_F$ become large and sends a sufficient "amount" of the key term $\tilde{Y}\tilde{Y}^\top F$ down to the hidden layers, grokking could happen. Here we categorize these factors into several categories.

**Learning rate**. (Gromov, 2023) reports that grokking happens without regularization, but with a large initial learning rate (verified by the author). This corresponds to increasing the strength of $G_F(t) \propto t\tilde{Y}\tilde{Y}^\top F$ at the initial phase of learning so that the hidden layers receives enough correct gradient signal to start the feature learning.

**Stay longer before overfitting.** (Prieto et al., 2025) uses stable softmax (linear form) rather than regular softmax (exponential form) in computing probability. This prevents the model from over-fitting to the label too quickly, and thus maintains a nonzero backpropagated gradient that can be useful for feature learning. (Kumar et al., 2024) also reports that grokking happens without regularization, using vanilla SGD optimizer. Our explanation is that it may take longer for SGD to converge to $V_{\text{ridge}}$ than Adam, and during that period, the hidden layer has already accumulated a sufficient amount of correct gradient signal.

**Weight initialization**. (Liu et al., 2023) reports that grokking happens with small initialization, regardless of the weight decay. This is straightforward from our framework, since $G_F(t) = O(\alpha) + t\tilde{Y}\tilde{Y}^\top F + O(\alpha t) + O(t^2)$ and if the weight initialization $\alpha$ is small, then $G_F(t)$ is dominated by clear signal term $t\tilde{Y}\tilde{Y}^\top \tilde{F}$, which leads to fast feature learning. If $\alpha$ is large, then $O(\alpha)$ term is large and the initial phase of $G_F$ contains too much noise, and we need to rely on the signal provided by the convergence phrase of $G_F$ controlled by the weight decay $\eta$. This is consistent with the finding by (Liu et al., 2023) that for large weight initialization, regularization is needed for grokking to happen, and small regularization leads to slow grokking transition.

**Scaling factor $\beta$ of the output**. (Kumar et al., 2024; Chizat et al., 2019) reports that scaling the output by a factor $\beta > 1$ will make the grokking faster. From $\texttt{Li}_2$ framework, this corresponds to optimizing $J_\beta(V) = \|\tilde{Y} - \beta\tilde{F}V\|_F^2 + \eta\|V\|_F^2$. Following a similar derivation as in Sec. H, in Sec. H.3 we can show that at the initial phase, the backpropagated gradient $G_F(t) = O(\alpha) + t\left[\beta\tilde{Y}\tilde{Y}^\top \tilde{F} + O(\beta^2\alpha) + O(\beta^3\alpha^2)\right] + O(t^2)$. So if $\beta > 1$ is large and the initialization $\alpha \ll 1$ then the signal term $t\beta\tilde{Y}\tilde{Y}^\top \tilde{F}$ becomes more dominant than the case of $\beta = 1$, and the grokking happens faster. However, if we have too large $\beta$ over $\alpha$, then the term $O(\beta^2\alpha)$ may cause trouble in feature learning and the final $G_F(+\infty)$ becomes small (Eqn. 69). Equivalently, same theory applies if we downscale the target label (Kumar et al., 2024).

**Weight decay $\eta$**. According to Eqn. 4, since $G_F(+\infty) \propto \eta\tilde{Y}\tilde{Y}^\top F$, it is clear that the weight decay $\eta$ becomes the *learning rate* of feature learning process. This coincides with findings in empirical works (Power et al., 2022; Clauw et al., 2024) that low regularization leads to slow grokking transition. This is also consistent with $t \sim 1/\eta$ laws to start grokking (Liu et al., 2023) or reach maximal test performance (Lewkowycz & Gur-Ari, 2020).

**Data size $n$**. Our sample analysis (Theorem. 4) shows the local maxima can be kept with sufficient number of samples ($n \gtrsim M \log M$). Intuitively, more samples lead to better shaped local maxima with less noise and thus the feature learning is faster, which is consistent with (Power et al., 2022; Nanda et al., 2023; Wang et al., 2024a; Gromov, 2023; Abramov et al., 2025).

**The number of hidden nodes $K$**. Our analysis requires that we need a decent number of hidden nodes $K$ to cover the diverse set of the local maxima of $\mathcal{E}$. On the other hand, Lemma 1 tells that very large $K$ may reduce $|G_F(+\infty)|$ and makes grokking slower. This is consistent with the finding by (Chizat et al., 2019) and NTK regime (Du et al., 2018; Jacot et al., 2018) that feature learning does not happen.

**Early stopping**. Model may overfit after generalization Montanari & Urbani (2025). Our framework shows that if data are scarce, generalization solutions have lower energy than memorization ones. If trained long enough, weights will move to solutions with larger $\mathcal{E}$.

# F    INDEPENDENT FEATURE LEARNING (SEC. 4)

**Lemma 3.** *Let $\phi_n(z) := \mathrm{He}_n(z)/\sqrt{n!}$ be the orthonormal Hermite system on $L^2(\gamma)$. If $(Z_1, Z_2)$ are standard normals with correlation $\rho$, then*

$$\mathbb{E}[\phi_n(Z_1)\,\phi_m(Z_2)] = \rho^{\,n}\,\delta_{nm} \qquad (n, m \ge 0).$$

*Proof of Lemma 3.* Use the generating function[2] $\exp(tz - \frac{t^2}{2}) = \sum_{k \ge 0} \phi_k(z)\, t^k$ for $z \sim \mathcal{N}(0,1)$. Then, for correlated normals $(Z_1, Z_2)$ with correlation $\rho$,

$$\mathbb{E}\!\left[e^{\,tZ_1 - \frac{t^2}{2}}\, e^{\,uZ_2 - \frac{u^2}{2}}\right] = \exp(\rho\,tu) = \sum_{k \ge 0} \rho^{\,k}\,(tu)^k.$$

Expanding the left-hand side by the generating functions and matching coefficients of $t^n u^m$ yields $\mathbb{E}[\phi_n(Z_1)\phi_m(Z_2)] = \rho^{\,n}\delta_{nm}$.

To show why $\mathbb{E}\!\left[e^{\,tZ_1 - \frac{t^2}{2}}\, e^{\,uZ_2 - \frac{u^2}{2}}\right] = \exp(\rho\,tu)$ is correct, decompose $(Z_1, Z_2)$ into Gaussian independent random variables $(X, Y)$:

$$Z_1 := X, \qquad Z_2 := \rho X + \sqrt{1 - \rho^2}\, Y,$$

Then we have

$$\mathbb{E}\!\left[e^{\,tZ_1 - \frac{t^2}{2}}\, e^{\,uZ_2 - \frac{u^2}{2}}\right] = \mathbb{E}\!\left[e^{\,tX - \frac{t^2}{2}}\, e^{\,u(\rho X + \sqrt{1-\rho^2}\, Y) - \frac{u^2}{2}}\right]$$

$$= \mathbb{E}\!\left[e^{\,(t+\rho u)X - \frac{t^2}{2}}\right]\mathbb{E}\!\left[e^{\,u\sqrt{1-\rho^2}\, Y - \frac{u^2}{2}}\right].$$

For $G \sim \mathcal{N}(0,1)$ we have $\mathbb{E}[e^{aG}] = e^{a^2/2}$, hence $\mathbb{E}\!\left[e^{\,aG - \frac{a^2}{2}}\right] = 1$ due to Lemma 4. Applying this twice,

$$\mathbb{E}\!\left[e^{\,(t+\rho u)X - \frac{t^2}{2}}\right] = \exp\!\left(\frac{(t+\rho u)^2}{2} - \frac{t^2}{2}\right) = \exp\!\left(\rho tu + \frac{\rho^2 u^2}{2}\right),$$

$$\mathbb{E}\!\left[e^{\,u\sqrt{1-\rho^2}\, Y - \frac{u^2}{2}}\right] = \exp\!\left(\frac{u^2(1-\rho^2)}{2} - \frac{u^2}{2}\right) = \exp\!\left(-\frac{\rho^2 u^2}{2}\right).$$

Multiplying the two factors yields

$$\exp\!\left(\rho tu + \frac{\rho^2 u^2}{2}\right)\exp\!\left(-\frac{\rho^2 u^2}{2}\right) = \exp(\rho tu),$$

as claimed.

$\square$

**Lemma 4** (Moment identity). *For $X \sim \mathcal{N}(0,1)$, $\mathbb{E}[e^{tX}] = \exp(t^2/2)$. Equivalently, $\mathbb{E}[e^{tX - t^2/2}] = 1$.*

*Proof.* Complete the square:

$$\mathbb{E}[e^{tX}] = \frac{1}{\sqrt{2\pi}}\int_{\mathbb{R}} e^{tx} e^{-x^2/2}\, dx = \frac{1}{\sqrt{2\pi}}\int e^{-(x-t)^2/2}\, e^{t^2/2}\, dx = \exp\!\left(\frac{t^2}{2}\right).$$

$\square$

**Lemma 1** (Structure of backpropagated gradient $G_F$). *Assume that (1) entries of $W$ follow standard normal distribution $N(0,1)$, (2) $\|\mathbf{x}_i\|_2 = const$, (3) $\|\mathbf{x}_i^\top \mathbf{x}_{i'} - \rho\|_2 \le \epsilon$ for all $i \ne i'$ and (4) large width $K$, then both $\tilde{F}^\top \tilde{F}$ and $\tilde{F}\tilde{F}^\top$ becomes a multiple of identity and Eqn. 4 becomes:*

$$G_F(+\infty) = \frac{\eta}{(Kc_1 + \eta)(nc_2 + \eta)}\tilde{Y}\tilde{Y}^\top \tilde{F} + O(K^{-1}\epsilon) \tag{5}$$

*where $c_1, c_2 > 0$ are constants related to nonlinearity. When $\eta$ is small, we have $G_F \propto \eta \tilde{Y}\tilde{Y}^\top \tilde{F}$. Note that the input features and/or weights can be scaled and what changes is $c_1$ and $c_2$.*

---

[2]https://en.wikipedia.org/wiki/Hermite_polynomials

*Proof.* In the following, we will prove that (1) $\tilde{F}^\top \tilde{F}$ is a multiple of identity and (2) $FF^\top \propto \alpha I + \beta \mathbf{1}\mathbf{1}^\top$. Without loss of generality, we assume that entry of $W$ follows standard normal distribution $\mathcal{N}(0, 1)$.

$\tilde{F}^\top \tilde{F}$ **is a multiple of identity**. Since each column of $\tilde{F}$ is $P_1^\perp \sigma(X\mathbf{w}_j)$ a zero-mean $n$-dimensional random vector and columns are i.i.d. due to the independence of columns of $W$. With large width $K$, $\tilde{F}^\top \tilde{F}$ becomes a multiple of identity.

$FF^\top$ **is a diagonal plus an all-constant matrix.** Note that the $i$-th row of $F$ is $[\sigma(\mathbf{w}_1^\top \mathbf{x}_i), \sigma(\mathbf{w}_2^\top \mathbf{x}_i), \ldots, \sigma(\mathbf{w}_K^\top \mathbf{x}_i)]$, with large width $K$, the inner product between the $i$-th row and $j$-th row of $F$ approximates to $K\mathcal{K}(i, j)$ where $\mathcal{K}(i, j)$ is defined as follows:

$$\mathcal{K}(i, j) = \mathbb{E}_{\mathbf{w}}[\sigma(\mathbf{w}^\top \mathbf{x}_i)\sigma(\mathbf{w}^\top \mathbf{x}_j)] \tag{12}$$

To estimate the entry $\mathcal{K}(i, j)$, we first do standardization by setting $Z_1 := \mathbf{w}^\top \mathbf{x}_i/s_i$ and $Z_2 := \mathbf{w}^\top \mathbf{x}_j/s_j$ where $s_i = \|\mathbf{x}_i\|_2$ and $s_j = \|\mathbf{x}_j\|_2$. Then $(Z_1, Z_2)$ are standard normals with $\mathrm{Corr}(Z_1, Z_2) = \rho_{ij}$, and $\mathcal{K}(i, j) = \mathbb{E}[\sigma(s_i Z_1)\sigma(s_j Z_2)]$.

Let $\phi_l(z) := \mathrm{He}_l(z)/\sqrt{l!}$ be the orthonormal Hermite system on $L^2(\gamma)$, where $\gamma$ is the standard Gaussian measure and $\mathrm{He}_l$ are the Hermite polynomials. For $s \geq 0$ define $f_s(z) := \sigma(sz)$. By the $L^2(\gamma)$ assumption, $f_s = \sum_{n=0}^\infty a_l(s)\,\phi_l$ with

$$a_l(s) = \langle f_s, \phi_l \rangle_{L^2(\gamma)} = \frac{1}{\sqrt{l!}}\,\mathbb{E}[\sigma(sZ)\,\mathrm{He}_l(Z)].$$

Thus

$$\sigma(s_i Z_1) = \sum_{l \geq 0} a_l(s_i)\,\phi_l(Z_1), \qquad \sigma(s_j Z_2) = \sum_{l \geq 0} a_l(s_j)\,\phi_l(Z_2).$$

By bilinearity and Lemma 3,

$$\mathcal{K}(i, j) = \mathbb{E}\left[\sum_{l \geq 0} a_l(s_i)\phi_l(Z_1) \sum_{m \geq 0} a_m(s_j)\phi_m(Z_2)\right] = \sum_{l,m \geq 0} a_l(s_i)a_m(s_j)\,\mathbb{E}[\phi_l(Z_1)\phi_m(Z_2)]$$

$$= \sum_{l \geq 0} a_l(s_i)a_l(s_j)\,\rho_{ij}^l.$$

If $s_i \equiv s$ and $\|\rho_{ij} - \rho\|_2 \leq \epsilon$ for $i \neq j$, then

$$\mathcal{K}(i, i) = \sum_{l \geq 0} a_l^2(s) =: a$$

Let $c := \sum_{l \geq 1} l a_l^2(s) < +\infty$ (it is convergent due to the big factor $l!$ in the denominator). Let $b := \sum_{l \geq 0} a_l^2(s)\,\rho^l$ and we have for all $i \neq j$:

$$\|\mathcal{K}(i, j) - b\|_2 \leq \sum_{l \geq 0} a_l^2(s)\,\|\rho_{ij}^l - \rho^l\|_2 \leq \sum_{l \geq 1} l a_l^2(s)\,\epsilon = c\epsilon$$

due to the fact that $\|\rho_{ij}^l - \rho^l\|_2 \leq l\xi^{l-1}\epsilon$ for all $l \geq 1$ and some $\xi$ in between $\rho_{ij}$ and $\rho$. hence $\mathcal{K}(i, j) = (a - b)\delta_{ij} + b + O(\epsilon)$ and thus $FF^\top = K(a - b)I + Kb\mathbf{1}\mathbf{1}^\top + O(K\epsilon)\mathbf{1}\mathbf{1}^\top$. Note that by Parseval's identity, $a = \mathbb{E}_{Z \sim \mathcal{N}(0,1)}[\sigma^2(sZ)]$.

Therefore, $\tilde{F}\tilde{F}^\top = K(a - b + O(\epsilon))P_1^\perp = K(a - b + O(\epsilon))(I - \mathbf{1}\mathbf{1}^\top/n) + O(K\epsilon)\mathbf{1}\mathbf{1}^\top$ and $P_\eta \tilde{Y} = \frac{\eta}{K(a-b)+\eta}\tilde{Y}$. Since $\tilde{F}^\top \tilde{F}$ is proportional to identity matrix, $(\tilde{F}^\top \tilde{F} + \eta I)^{-1}$ is also proportional to identity matrix and the conclusion follows. $\square$

## F.1 THE ENERGY FUNCTION $\mathcal{E}$ (SEC. 4.3)

**Theorem 1** (The energy function $\mathcal{E}$ for independent feature learning). *The dynamics (Eqn. 6) of independent feature learning is exactly the gradient ascent dynamics of the energy function $\mathcal{E}$ w.r.t. $\mathbf{w}_j$, a nonlinear canonical-correlation analysis (CCA) between the input $X$ and target $\tilde{Y}$:*

$$\mathcal{E}(\mathbf{w}_j) = \frac{1}{2}\|\tilde{Y}^\top \sigma(X\mathbf{w}_j)\|_2^2 \tag{7}$$

*Proof.* Taking gradient of $\mathcal{E}$ w.r.t. $\mathbf{w}_j$, and we have $\cdot \mathbf{w}_j = X^\top D_j \tilde{Y} \tilde{Y}^\top \sigma(X\mathbf{w}_j)$, which proves the theorem. $\qquad \square$

**Theorem 2** (Local maxima of $\mathcal{E}$ for group input). *For group arithmetics tasks with $\sigma(x) = x^2$, $\mathcal{E}$ has multiple local maxima $\mathbf{w}^* = [\mathbf{u}; \pm P\mathbf{u}]$. Either it is in a real irrep of dimension $d_k$ (with $\mathcal{E}^* = M/8d_k$ and $\mathbf{u} \in \mathcal{H}_k$), or in a pair of complex irrep of dimension $d_k$ (with $\mathcal{E}^* = M/16d_k$ and $\mathbf{u} \in \mathcal{H}_k \oplus \mathcal{H}_{\bar{k}}$). These local maxima are not connected. No other local maxima exist.*

*Proof.* Following this setting, if ordered by target values, we can write down the data matrix $X = [X_{h_1}; X_{h_2}; \ldots X_{h_M}]$ (i.e., each $X_h$ occupies $M$ rows of $X$) in which each $X_h = [R_h^\top, P] \in \mathbb{R}^{M \times 2M}$. Here $R_h$ is the *regular representation* (a special case of permutation representation) of group element $h$ so that $\mathbf{e}_{h_1 h_2} = R_{h_1} \mathbf{e}_{h_2}$ for all $h_1, h_2 \in H$, and $P$ is the group inverse operator so that $P\mathbf{e}_h = \mathbf{e}_{h^{-1}}$. This is because each row of $X$ that corresponds to the target $h$ can be written as $[\mathbf{e}_{hh_1}^\top, \mathbf{e}_{h_1}^\top P] = [\mathbf{e}_{h_1}^\top R_h^\top, \mathbf{e}_{h_1}^\top P]$. Stacking the rows that lead to target $h$ together, and order them by $h_1$, we get $X_h = [R_h^\top, P]$.

Let $\mathbf{w} = [\mathbf{u}; P\mathbf{v}]$. Let matrix $S_{ij} := \sigma(u_i + v_j)$, since $R_h$ is a permutation matrix, then $\sigma(X_h\mathbf{w}) = \sigma(R_h^\top \mathbf{u} + \mathbf{v})$ is a row shuffling of $S$. Therefore, $\sigma(X_h\mathbf{w}) = \mathrm{diag}(R_h^\top S)\mathbf{1}_M$, where $\mathrm{diag}(\cdot)$ is the diagonal of a matrix. Note that in this target label ordering, we have $Y = I_M \otimes \mathbf{1}_M$. So for each column $h$ of $Y$, we have $\mathbf{y}_h = \mathbf{e}_h \otimes \mathbf{1}_M$. So

$$z_h := \mathbf{y}_h^\top \sigma(X\mathbf{w}) = \mathbf{1}_M^\top \sigma(X_h\mathbf{w}) = \mathbf{1}_M^\top \mathrm{diag}(R_h^\top S)\mathbf{1}_M = \mathrm{tr}(R_h^\top S) = \langle R_h, S \rangle_F \qquad (13)$$

where $\langle A, B \rangle_F := \mathrm{tr}(A^\top B)$ is the Frobenius inner product. And the energy $\mathcal{E}$ can be written as:

$$\mathcal{E}(\mathbf{w}) = \frac{1}{2} \sum_h (z_h - \bar{z})^2 \qquad (14)$$

where $\bar{z} := \frac{1}{M} \sum_h z_h = \frac{1}{M} \sum_h \langle R_h, S \rangle_F = \langle \frac{1}{M} \sum_h R_h, S \rangle_F = \frac{1}{M} \langle \mathbf{1}_M \mathbf{1}_M^\top, S \rangle_F$. Therefore, using $R_h \mathbf{1}_M = \mathbf{1}_M$, $\mathcal{E}(\mathbf{w})$ can be written as:

$$\mathcal{E}(\mathbf{w}) = \frac{1}{2} \sum_h \langle \tilde{R}_h, S \rangle_F^2 \qquad (15)$$

where $\tilde{R}_h = R_h P_1^\perp$. Now we study its property. We decompose $\{\tilde{R}_h\}$ into complex irreducible representations:

$$\tilde{R}_h = Q \left( \bigoplus_{k \neq 0} \bigoplus_{r=1}^{m_k} C_k(h) \right) Q^* \qquad (16)$$

where $C_k(h)$ is the $k$-th irreducible representation block of $R_h$, $Q$ is the unitary matrix (and $Q^*$ is the conjugate transpose of $Q$) and $m_k$ is the multiplicity of the $k$-th irreducible representation. Since $\tilde{R}_h$ is a zero-meaned representation, we remove the trivial representation $C_0(h)$ and thus $Q^*\mathbf{1} = 0$. Let $\hat{S} = Q^\top S Q$. Then

$$\sum_h \langle \tilde{R}_h, S \rangle_F^2 = \sum_h \langle Q \left( \bigoplus_{k \neq 0} \bigoplus_{r=1}^{m_k} C_k(h) \right) Q^*, S \rangle_F = \langle \bigoplus_{k \neq 0} \bigoplus_{r=1}^{m_k} C_k(h), \hat{S} \rangle_F = \sum_{k \neq 0} \sum_{r=1}^{m_k} \mathrm{tr}(C_k^*(h) \hat{S}_{k,r}) \qquad (17)$$

$$= \sum_{(k,r),(k',r')} \mathrm{vec}^*(\hat{S}_{k,r}) \left[ \sum_h \mathrm{vec}(C_k(h)) \mathrm{vec}(C_{k'}^*(h)) \right] \mathrm{vec}(\hat{S}_{k',r'}) \qquad (18)$$

**Case 1.** If $k \neq k'$ are inequivalent irreducible representations of dimension $d_k$ and $d_{k'}$, then we can prove that $\sum_h \mathrm{vec}(C_k(h)) \mathrm{vec}(C_{k'}^*(h)) = 0$. To see this, let $\mathsf{A}_{k,k'}(Z) = \sum_h C_k(h) Z C_{k'}^{-1}(h)$, then $\mathsf{A}_{k,k'}(Z)$ is a $H$-invariant linear mapping from $d_k$ to $d_{k'}$ dimensional space. Thus by Schur's lemma, $\mathsf{A}_{k,k'}(Z) = 0$ for any $Z$. But since $\mathrm{vec}(\mathsf{A}_{k,k'}(Z)) = \left( \sum_h \bar{C}_{k'}(h) \otimes C_k(h) \right) \mathrm{vec}(Z)$, we have $\sum_h \bar{C}_{k'}(h) \otimes C_k(h) = 0$. Expanding each component, we have $\sum_h \mathrm{vec}(C_k(h)) \mathrm{vec}(C_{k'}^*(h)) = 0$.

**Case 2.** If $k = k'$ are equivalent irreducible representations (and both have dimension $d_k$), then we can prove that $\sum_h \mathrm{vec}(C_k(h)) \mathrm{vec}(C_k^*(h)) = \frac{M}{d_k} \mathrm{vec}(I_{d_k}) \mathrm{vec}^*(I_{d_k})$. Then with Schur's average

lemma, we have $\mathsf{A}_{kk}(Z) = \frac{M}{d_k}\operatorname{tr}(Z)I_{d_k}$. A vectorization leads to $\left(\sum_h \bar{C}_k(h)\otimes C_k(h)\right)\operatorname{vec}(Z) = \frac{M}{d_k}\operatorname{tr}(Z)\operatorname{vec}(I_{d_k})$. Notice that $\operatorname{vec}^*(I_{d_k})\operatorname{vec}(Z) = \operatorname{tr}(Z)$ and we arrive at the conclusion.

Therefore, for the objective function we have:

$$\mathcal{E}(\mathbf{w}) = \frac{1}{2}\sum_h \langle \tilde{R}_h, S\rangle_F^2 = \frac{M}{2}\sum_{k\neq 0}\frac{1}{d_k}\left|\sum_r \operatorname{tr}(\hat{S}_{k,r})\right|^2 \tag{19}$$

**Special case of quadratic activation**. If $\sigma(x) = x^2$, then we have $S = (\mathbf{u}\circ\mathbf{u})\mathbf{1}^\top + \mathbf{1}(\mathbf{v}\circ\mathbf{v})^\top + \mathbf{u}\mathbf{v}^\top$ and thus $\hat{S} = \hat{\mathbf{u}}\hat{\mathbf{v}}^*$, where $\hat{\mathbf{u}} = Q^*\mathbf{u}$ and $\hat{\mathbf{v}} = Q^*\mathbf{v}$. Therefore, since $Q^*\mathbf{1} = 0$, $\hat{S}_{k,r} = \hat{\mathbf{u}}_{k,r}\hat{\mathbf{v}}_{k,r}^*$ and $\operatorname{tr}(\hat{S}_{k,r}) = \hat{\mathbf{u}}_{k,r}^*\hat{\mathbf{v}}_{k,r}$. Therefore, with Cauchy-Schwarz inequality, we have

$$\mathcal{E} = \frac{1}{2}\sum_h \langle \tilde{R}_h, S\rangle_F^2 = \frac{M}{2}\sum_{k\neq 0}\frac{1}{d_k}\left|\sum_r \hat{\mathbf{u}}_{k,r}^*\hat{\mathbf{v}}_{k,r}\right|^2 \leq \frac{M}{2}\sum_{k\neq 0}\frac{1}{d_k}\left(\sum_r|\hat{\mathbf{u}}_{k,r}|^2\right)\left(\sum_r|\hat{\mathbf{v}}_{k,r}|^2\right) \tag{20}$$

Let $a_k = \sum_r|\hat{\mathbf{u}}_{k,r}|^2$, $b_k = \sum_r|\hat{\mathbf{v}}_{k,r}|^2$, and $c_k = a_k + b_k \geq 0$. Then we have:

$$\mathcal{E} = \frac{1}{2}\sum_h \langle \tilde{R}_h, S\rangle_F^2 \leq \frac{M}{2}\sum_{k\neq 0}\frac{a_k b_k}{d_k} \leq \frac{M}{8}\sum_{k\neq 0}\frac{c_k^2}{d_k}, \qquad \text{subject to}\ \sum_{k\neq 0}c_k = 1 \tag{21}$$

which has one global maxima (i.e., $c_{k_0} = 1$ for $k_0 = \arg\min_k d_k$) and multiple local maxima. The maximum is achieved if and only if $\hat{\mathbf{u}}_{k_0,r} = \pm\hat{\mathbf{v}}_{k_0,r}$ for all $r$ and $\sum_r|\hat{\mathbf{u}}_{k_0,r}|^2 = \sum_r|\hat{\mathbf{v}}_{k_0,r}|^2 = 1/2$.

**Local maxima**. For each irreducible representation $k_0$, $c_{k_0} = 1$ is a local maxima. This is because for small perturbation $\epsilon$ that moves the solution from $c_k = \mathbb{I}(k = k_0)$ to $c_k' = \begin{cases} 1-\epsilon & \text{if } k = k_0 \\ \epsilon_k & \text{if } k \neq k_0 \end{cases}$ with $\epsilon_k \geq 0$ and $\sum_{k\neq k_0}\epsilon_k = \epsilon$, for $\mathcal{E} = \mathcal{E}(\{c_k\})$ and $\mathcal{E}' = \mathcal{E}(\{c_k'\})$ we have:

$$\mathcal{E}' = \frac{M}{8}\sum_{k\neq 0}\frac{(c_k')^2}{d_k} = \frac{M}{8}\left(\frac{(c_{k_0}-\epsilon)^2}{d_{k_0}} + \sum_{k\neq k_0,0}\frac{\epsilon_k^2}{d_k}\right) \tag{22}$$

$$\leq \frac{M}{8}\left(\frac{c_{k_0}^2}{d_{k_0}} - \frac{2\epsilon}{d_{k_0}}\right) + O(\epsilon^2) < \frac{M}{8}\frac{c_{k_0}^2}{d_{k_0}} = \frac{M}{8}\sum_{k\neq 0}\frac{c_k^2}{d_k} = \mathcal{E} \tag{23}$$

All local maxima are flat, since we can always move around within $\hat{\mathbf{u}}_{k,r}$ and $\hat{\mathbf{v}}_{k,r}$, while the objective function remains the same. $\qquad\square$

**Optimizing in Real domain**. The above analysis uses complex irreducible representations. For real $\mathbf{w}$, $\hat{S}_{k,r}$ will be a complex conjugate of $\hat{S}_{-k,r}$ for conjugate irreducible representations $k$ and $-k$. This means that we can partition the sum in Eqn. 19 into real and complex parts:

$$\mathcal{E}(\mathbf{w}) = \frac{M}{2}\sum_{k\neq 0,k\ \text{real}}\frac{1}{d_k}\left|\sum_r \operatorname{tr}(\hat{S}_{k,r})\right|^2 + M\sum_{k\neq 0,k\ \text{complex, take one}}\frac{1}{d_k}\left|\sum_r \operatorname{tr}(\hat{S}_{k,r})\right|^2 \tag{24}$$

The above equation holds since $R_g$ is real, and for any complex irreducible representation $k$, its conjugate representation $-k$ is also included. Therefore, to optimize $\mathcal{E}$ in the real domain $\mathbb{R}$, we can just optimize only on the real part plus the complex part taken one of the conjugate pair in the complex domain $\mathbb{C}$.

**Zero-meaned one hot representation**. Note that if we use zero-meaned one hot representation $\tilde{\mathbf{e}}_h = P_1^\perp\mathbf{e}_h$, then $R_{h_1}\tilde{\mathbf{e}}_{h_2} = \tilde{\mathbf{e}}_{h_1h_2}$ and $P\tilde{\mathbf{e}}_h = \tilde{\mathbf{e}}_{h^{-1}}$ still hold, and $\tilde{X}_h = P_1^\perp X_h = P_1^\perp[R_h^\top, P] = [R_h^\top, P][P_1^\perp; P_1^\perp]$. This means that we can still use $X_h$ but enforce zero-meaned constraints on $\mathbf{u}$ and $\mathbf{v}$, which is already included since $Q^*\mathbf{1} = 0$.

**Corollary 1** (Flatness of local maxima of $\mathcal{E}$ for group input). *Local maxima of $\mathcal{E}$ for group arithmetics tasks with $|H| = M > 2$ are flat, i.e., at least one eigenvalue of its Hessian is zero.*

*Proof.* For Abelian group $H$ with $|H| = M > 2$, all irreducible representations are 1-dimensional, and at least one of it is complex. Since $\mathbb{C}$ is treated as 2D space in optimization, it has at least 1 degree of freedom to change without changing its function value (Eqn. 24). So the Hessian has at least 1 zero eigenvalue. For non-Abelian group, there is at least one irreducible representation $k$ with dimension greater than 1, which means it has at least 1 degrees of freedom to change $\hat{S}_{k,r}$ without changing $|\sum_r \text{tr}(\hat{S}_{k,r})|^2$ and thus its function value (Eqn. 24). So the Hessian has at least 1 zero eigenvalue. $\qquad\square$

### F.2 RECONSTRUCTION POWER OF LEARNED FEATURES (SEC. 4.4)

**Theorem 3** (Target Reconstruction). *Assume (1) $\mathcal{E}$ is optimized in complex domain $\mathbb{C}$, (2) for each irrep $k$, there are $m_k^2 d_k^2$ pairs of learned weights $\mathbf{w} = [\mathbf{u}; \pm P\mathbf{u}]$ whose associated rank-1 matrices $\{\mathbf{u}\mathbf{u}^*\}$ form a complete bases for $\mathcal{H}_k$ and (3) the top layer $V$ also learns with $\eta = 0$, then $\hat{Y} = \tilde{Y}$.*

*Proof.* For each nontrivial irrep $k$, let $\Pi_k$ be the central idempotent projector onto the isotypic subspace $\mathcal{H}_k = I_{m_k} \otimes \mathbb{C}^{d_k}$ (for the regular rep, $m_k = d_k$). Let $\text{End}(\mathcal{H}_k)$ be the space of all linear operators that map $\mathcal{H}_k$ to itself. Note that the dimensionality of $\mathcal{H}_k$ is $D_k := m_k d_k$.

Let $\mathbf{w} = [\mathbf{u}, P\mathbf{v}]$ be the weights learned by optimizing the energy function $\mathcal{E}$ with quadratic activation $\sigma(x) = x^2$. From Thm. 2, we know that at local optima, $\mathbf{u}_j = \pm\mathbf{v}_j$ and $\mathbf{1}^\top \mathbf{u}_j = 0$. Therefore, the feature $\tilde{\mathbf{f}}_{j,h} \in \mathbb{R}^M$ is given by ($\circ$ denotes the Hadamard product)

$$\tilde{\mathbf{f}}_{j,h} = \pm 2\,(R_h^\top \mathbf{u}_j) \circ \mathbf{u}_j + (R_h^\top \mathbf{u}_j)^{\circ 2} - \frac{1}{M}\sum_h (R_h^\top \mathbf{u}_j)^{\circ 2}$$

The third term $\mathbf{u}^{\circ 2}$ is a constant across all $h$ and was removed in the zero-meaned projection. By our assumption we have node $j$ and $j'$ with both positive and negative signs. So $\frac{1}{2}\left(\tilde{\mathbf{f}}_{j,h} - \tilde{\mathbf{f}}_{j',h}\right) = 2\,(R_h^\top \mathbf{u}_j) \circ \mathbf{u}_j$. If a linear representation of $\{\tilde{\mathbf{f}}_j\}$ can perfectly reconstruct the target $\tilde{Y}$, so does the original representation. So for now we just let feature $\tilde{\mathbf{f}}_{j,h} = 2\,(R_h^\top \mathbf{u}_j) \circ \mathbf{u}_j = 2\text{diag}(R_h^\top \mathbf{u}_j \mathbf{u}_j^*)$. Let $U_j := \mathbf{u}_j \mathbf{u}_j^*$, which is Hermitian in $\text{End}(\mathcal{H}_k)$, then $\tilde{\mathbf{f}}_{j,h} = 2\text{diag}(R_h^\top U_j)$.

**Gram block diagonalization.** For each irrep $k$, let $J_k$ be the set of all node $j$ that converges to the $k$-th irrep. For any Hermitian operator $U$ supported in $\mathcal{H}_k$ (i.e. $U = \Pi_k U \Pi_k$), define the centered quadratic cross-feature

$$\mathbf{c}_U(h) := 2\,\text{diag}(R_h^\top U) \in \mathbb{C}^M,$$

and write $\mathbf{c}_{U_j} = [\mathbf{c}_{U_j}(h)]_{h\in H} \in \mathbb{C}^{M^2}$ as a concatenated vector.

For $U, V \in \text{End}(\mathcal{H}_k)$, define $\mathcal{G}(U,V) := \sum_{h\in H}\langle \mathbf{c}_U(h), \mathbf{c}_V(h)\rangle$. On $\mathcal{H}_k$, $R_h = I_{m_k} \otimes C_k(h)$, so the map $U \mapsto \mathbf{c}_U(h)$ is linear and the bilinear form $\mathcal{G}$ is invariant under $U \mapsto (I \otimes C_k(g))U(I \otimes C_k(g))^*$. By Schur's lemma, $\mathcal{G}(U,V) = \alpha_k\langle U,V\rangle = \alpha_k \text{tr}(UV^*)$ for some scalar $\alpha_k$. Evaluating on rank-one $U = V$ (or by a direct calculation) gives $\alpha_k = 4$, hence

$$\sum_h \langle \mathbf{c}_U(h), \mathbf{c}_V(h)\rangle = 4\,\text{tr}(UV^*).$$

For $U_j = \mathbf{u}_j \mathbf{u}_j^*$ and $U_\ell = \mathbf{u}_\ell \mathbf{u}_\ell^*$ from $\mathcal{H}_k$ and $\mathcal{H}_\ell$ with $k \neq \ell$, we have

$$\sum_h \langle \mathbf{c}_{U_j}(h), \mathbf{c}_{U_\ell}(h)\rangle = 4\mathbf{1}^\top \sum_h \text{diag}(R_h^\top \mathbf{u}_j \mathbf{u}_j^*) \circ \text{diag}(R_h^\top \bar{\mathbf{u}}_\ell \bar{\mathbf{u}}_\ell^*)$$

$$= 4\mathbf{1}^\top \sum_h (R_h^\top \mathbf{u}_j) \circ \bar{\mathbf{u}}_j \circ R_h^\top \bar{\mathbf{u}}_\ell \circ \mathbf{u}_\ell = 4\mathbf{1}^\top \left[\left(\sum_h R_h\right)(\mathbf{u}_j \circ \bar{\mathbf{u}}_\ell)\right] \circ \bar{\mathbf{u}}_j \circ \mathbf{u}_\ell$$

$$= 4|\mathbf{u}_j^* \mathbf{u}_\ell|^2$$

This means that $\langle \tilde{\mathbf{f}}_j, \tilde{\mathbf{f}}_\ell\rangle = \langle \mathbf{c}_{U_j}, \mathbf{c}_{U_\ell}\rangle = 0$. And thus the Gram matrix $G := \tilde{F}^\top \tilde{F}$ is block diagonal with each block $G_k$ corresponding to an irrep subspace $k$. Here $G_k \in \mathbb{C}^{N_k \times N_k}$. Note that since we sample $D_k^2 = m_k^2 d_k^2$ weights, then $\{U_j\}_{j\in J_k}$ becomes a complete set of bases (not necessarily orthogonal bases) and thus $G_k$ is invertible.

**Right-hand side.** For any $U \in \text{End}(\mathcal{H}_k)$,

$$r_U(h') = \sum_x \mathbf{c}_U(h')_x = 2\,\text{tr}\left((\Pi_k R_{h'} \Pi_k)U\right) = 2\,\text{tr}\left((I_{m_k} \otimes C_k(h'))U\right).$$

and we have $[\tilde{\mathbf{f}}_j^\top Y]_{h'} = [\tilde{\mathbf{f}}_j^\top \tilde{Y}]_{h'} = r_{U_j}(h')$.

**Solve LS.** Now we try to solve the LS problem $GV = \tilde{F}^\top \tilde{Y}$. Due to the block diagonal nature, this can be solved independently for each $G_k$. Consider $G_k V_k = \tilde{F}_k^\top \tilde{Y}$. Here $\tilde{F}_k = [\tilde{\mathbf{f}}_j]_{j \in J_k}$ collects the subset column $J_k$ from $\tilde{F}$.

Therefore, $V_k = G_k^{-1} \tilde{F}_k^\top \tilde{Y}$ and $v_j(h')$ as the $(j, h')$ entry of $V_k$, has $v_j(h') = \sum_l [G_k^{-1}]_{jl} r_{U_l}(h') = 2\sum_l [G_k^{-1}]_{jl}\,\text{tr}\left((I_{m_k} \otimes C_k(h'))U_l\right)$. Then we have $\hat{Y}^{(k)} = \tilde{F}_k V_k$:

$$\hat{Y}^{(k)}_{(\cdot, h),\, h'} = \sum_{j \in J_k} v_j(h')\,\mathbf{c}_{U_j}(h) = 4\sum_{j \in J_k}\sum_l [G_k^{-1}]_{jl}\,\text{tr}\left((I \otimes C_k(h'))U_l\right) \cdot \text{diag}(R_h^\top U_j).$$

By linearity in $U$ and completeness of $\{U_j\}$ (the Hermitian bases span all operators in $\mathcal{H}_k$), we have for any $A \in \text{End}(\mathcal{H}_k)$:

$$4\sum_{jl}[G_k^{-1}]_{jl}\,\text{tr}(AU_l)\,\text{diag}(R_h^\top U_j) = \text{diag}(R_h^\top A)$$

The last equality holds by noticing that $\langle A, U_l \rangle = \text{vec}^*(U_l)\,\text{vec}(A)$ and thus $4\sum_{jl}[G_k^{-1}]_{jl}\langle A, U_l \rangle U_j = A$. Take $A = I \otimes C_k(h') = \Pi_k R_{h'} \Pi_k \in \text{End}(\mathcal{H}_k)$, and we have:

$$\boxed{\hat{Y}^{(k)}_{(\cdot, h),\, h'} = \text{diag}\left(R_h^\top \Pi_k R_{h'} \Pi_k\right) \qquad (h, h' \in H).}$$

To see why $\hat{Y} = \tilde{Y}$, we have:

$$\hat{Y}^{(k)}_{(\cdot, h), h'} = \text{diag}\left(R_h^\top(\Pi_k R_{h'} \Pi_k)\right) \;\Rightarrow\; \sum_{k \neq 0} \hat{Y}^{(k)}_{(\cdot, h), h'} = \text{diag}\left(R_h^\top\left(\sum_{k \neq 0}\Pi_k R_{h'}\Pi_k\right)\right).$$

Since $\sum_k \Pi_k = I$ and $\Pi_k R_{h'} = R_{h'}\Pi_k$,

$$\sum_{k \neq 0} \Pi_k R_{h'} \Pi_k = R_{h'} - \Pi_0.$$

where $\Pi_0 = \frac{1}{M}\mathbf{1}_M\mathbf{1}_M^\top$ is the central idempotent projector onto the trivial irrep. Thus

$$\sum_{k \neq 0} \hat{Y}^{(k)}_{(\cdot, h), h'} = \text{diag}(R_h^\top R_{h'}) - \text{diag}(R_h^\top \Pi_0) = \begin{cases} (1 - \frac{1}{M})\mathbf{1}_M, & h = h', \\ -\frac{1}{M}\mathbf{1}_M, & h \neq h', \end{cases}$$

because $\text{diag}(R_h^\top R_{h'}) = \mathbf{1}_M$ iff $h = h'$ and 0 otherwise, while $\text{diag}(R_h^\top \Pi_0) = \frac{1}{M}\mathbf{1}_M$ for all $h$. Hence $\sum_{k \neq 0} \hat{Y}^{(k)} = P_1^\perp Y = \tilde{Y}$. $\qquad\square$

**Remark.** The above proof also works for real $\mathbf{w}$ since we can always take a real decomposition of $R_h$ and all the above steps follow.

**Property of the square term.** With quadratic features the class-centered column for node $j$ and block $h$ decomposes as $\tilde{F} = [A, B]$, where for $B$ each column $j$ (and block $h$) is $\mathbf{b}_{j,h} := R_h^\top(\mathbf{u}_j^{\circ 2}) - \frac{\|\mathbf{u}_j\|_2^2}{M}\mathbf{1}_M$ (the "square" part) and for $A$ each column $j$ (and block $h$) is $\mathbf{a}_{j,h} := 2\left(R_h^\top \mathbf{u}_j\right) \circ \mathbf{u}_j$ (the "cross" part we discussed above). The vector $\mathbf{b}_j$ is entrywise mean-zero, i.e. $\sum_x \mathbf{b}_j(x) = 0$ for all $h$, hence it has zero correlation with any class-centered target column $\tilde{Y}_{(\cdot, h')} \propto \mathbf{1}$: $(\mathbf{b}_{j,h}^\top \tilde{Y})_{h'} =$

$\sum_x \mathbf{b}_{j,h'}(x) = 0$. Moreover, under $\mathbf{1}^\top \mathbf{u}_j = \mathbf{1}^\top \mathbf{u}_\ell = 0$ one has $\sum_h \langle \mathbf{b}_{j,h}, \mathbf{a}_{\ell,h} \rangle = 0$. So the normal equation becomes

$$\tilde{F}^\top \tilde{F} V = \begin{bmatrix} A^\top A & A^\top B \\ B^\top A & B^\top B \end{bmatrix} V = \begin{bmatrix} A^\top \tilde{Y} \\ B^\top \tilde{Y} \end{bmatrix}$$

which gives

$$\begin{bmatrix} A^\top A & 0 \\ 0 & B^\top B \end{bmatrix} V = \begin{bmatrix} A^\top \tilde{Y} \\ 0 \end{bmatrix}$$

So even with the square term $B$ in $\tilde{F}$, $V$ will still have zero coefficient on them.

### F.3 SCALING LAWS OF MEMORIZATION AND GENERALIZATION (SEC. 4.5)

**Theorem 4** (Amount of samples to maintain local optima)**.** *If we select $n \gtrsim d_k^2 M \log(M/\delta)$ data sample from $H \times H$ uniformly at random, then with probability at least $1 - \delta$, the empirical energy function $\hat{\mathcal{E}}$ keeps local maxima for $d_k$-dimensional irreps (Thm. 2).*

*Proof.* **Overview**. We keep the setting and notation of the theorem in the prompt (group $H$, $|H| = M$, quadratic activation, $S$ as defined there, $z_h = \langle R_h, S \rangle = \mathrm{tr}(R_h^\top S)$, zero-mean removal already folded into $\tilde{R}_h$). We analyze random row subsampling and show that the empirical objective keeps the same local-maxima structure with $n \gtrsim M \log(M/\delta)$ retained rows.

**Setup**. There are $M^2$ rows indexed by pairs $(h_1, h_2) \in H \times H$, with target $h = h_1 h_2$. For each $h \in H$, exactly $M$ rows map to $h$; we index them by $j \in [M]$ after ordering by $h_1$ as in the proof, and write

$$s_{h,j} := \left(R_h^\top S\right)_{jj}. \qquad \text{so that} \qquad z_h = \sum_{j=1}^M s_{h,j} = \langle R_h, S \rangle.$$

We subsample *rows* independently with keep-probability $p \in (0, 1]$. Let $\xi_{h,j} \in \{0, 1\}$ be the keep indicator for the row $(h, j)$:

$$\Pr(\xi_{h,j} = 1) = p, \quad \text{i.i.d. over } (h, j).$$

The number of kept rows for target $h$ is

$$\hat{m}_h := \sum_{j=1}^M \xi_{h,j} \sim \mathrm{Bin}(M, p), \qquad \mathbb{E}[\hat{m}_h] = pM, \quad \mathrm{Var}(\hat{m}_h) = Mp(1-p).$$

**Estimator for $z_h$.** We use the *linear/unbiased* (Horvitz–Thompson) target-wise estimator

$$\hat{z}_h := \frac{1}{p} \sum_{j=1}^M \xi_{h,j} \, s_{h,j}. \qquad \Rightarrow \qquad \mathbb{E}[\hat{z}_h \,|\, S] = z_h.$$

Define the diagonal sampling matrix

$$W_h^{\mathrm{HT}} := \mathrm{diag}\left(\frac{\xi_{h,1}}{p}, \ldots, \frac{\xi_{h,M}}{p}\right), \quad \text{so} \quad \hat{z}_h = \mathrm{tr}\left(R_h^\top S \, W_h^{\mathrm{HT}}\right) = \langle R_h W_h^{\mathrm{HT}}, S \rangle.$$

**The empirical Gram operator.** Set the normalized per-target weight

$$w_h := \frac{\hat{m}_h}{pM}, \qquad \mathbb{E}[w_h] = 1, \qquad \mathrm{Var}(w_h) = \frac{1-p}{pM} \leq \frac{1}{pM}.$$

Decompose $W_h^{\mathrm{HT}}$ into its mean and zero-mean parts:

$$W_h^{\mathrm{HT}} = w_h I + \Delta_h, \qquad \mathrm{tr}(\Delta_h) = 0, \qquad \mathbb{E}[\Delta_h \,|\, \hat{m}_h] = 0.$$

Therefore

$$\hat{z}_h = \langle R_h(w_h I + \Delta_h), S \rangle = w_h z_h + \varepsilon_h, \qquad \varepsilon_h := \langle R_h \Delta_h, S \rangle, \qquad \mathbb{E}[\varepsilon_h \,|\, S, \hat{m}_h] = 0. \tag{25}$$

Using the decomposition

$$z_h = \sum_{k \neq 0} \sum_{r=1}^{m_k} \operatorname{tr}(C_{k,h}^* \widehat{S}_{k,r}) = \sum_{k \neq 0} \sum_{r=1}^{m_k} \operatorname{vec}(\widehat{S}_{k,r})^* \operatorname{vec}(C_{k,h}),$$

we obtain

$$\sum_h \widehat{z}_h^2 = \sum_h \left(w_h z_h + \varepsilon_h\right)^2 = \underbrace{\sum_h w_h^2 z_h^2}_{\text{signal}} + 2\underbrace{\sum_h w_h z_h \varepsilon_h}_{\text{mixed}} + \underbrace{\sum_h \varepsilon_h^2}_{\text{noise}}. \tag{26}$$

The signal term can be written as a quadratic form over irrep blocks:

$$\sum_h w_h^2 z_h^2 = \sum_{(k,r),(k',r')} \operatorname{vec}(\widehat{S}_{k,r})^* \Big[ \sum_h w_h^2 \operatorname{vec}(C_{k,h}) \operatorname{vec}(C_{k',h})^* \Big] \operatorname{vec}(\widehat{S}_{k',r'}). \tag{27}$$

Recall that the full-data operator is

$$\mathsf{A}_{k,k'} := \frac{1}{M} \sum_h \overline{C}_{k',h} \otimes C_{k,h}.$$

and $\operatorname{vec}(C_{k,h}) \operatorname{vec}(C_{k',h})^*$ is just a column and row reshuffling of $\overline{C}_{k',h} \otimes C_{k,h}$. In the following we will study approximation errors of $\mathsf{A}_{k,k'}$ instead. Let

$$\widehat{\mathsf{A}}_{k,k'}^{(2)} := \frac{1}{M} \sum_h w_h^2 \overline{C}_{k',h} \otimes C_{k,h} \qquad \text{and} \qquad \widehat{\mathsf{A}}_{k,k'} := \frac{1}{M} \sum_h w_h \overline{C}_{k',h} \otimes C_{k,h}$$

the *second-* and *first-weighted* empirical Gram operators, respectively. By construction, $\mathbb{E}[\widehat{\mathsf{A}}_{k,k'}] = \mathsf{A}_{k,k'}$ and $\mathbb{E}[\widehat{\mathsf{A}}_{k,k'}^{(2)}] = \mathsf{A}_{k,k'} + \frac{1-p}{pM} \mathsf{A}_{k,k'}$ (a tiny bias of order $1/(pM)$).

**Error bounds for each $(k,k')$ block.** We will control three deviations, uniformly over all $(k,k')$:

$$\textbf{E1}: \quad \left\| \widehat{\mathsf{A}}_{k,k'} - \mathsf{A}_{k,k'} \right\|_{\text{op}} \leq c_1 \sqrt{\frac{\log(M/\delta)}{Mp}}, \tag{28}$$

$$\textbf{E2}: \quad \left\| \widehat{\mathsf{A}}_{k,k'}^{(2)} - \widehat{\mathsf{A}}_{k,k'} \right\|_{\text{op}} \leq c_2 \sqrt{\frac{\log(M/\delta)}{Mp}} + \frac{c_2'}{Mp}, \tag{29}$$

$$\textbf{E3}: \quad \left| \sum_h w_h z_h \varepsilon_h \right| \leq c_3 \|z\|_2 \sqrt{\frac{M \log(M/\delta)}{p}}, \qquad \sum_h \varepsilon_h^2 \leq c_4 \frac{M \log(M/\delta)}{p}, \tag{30}$$

for numerical constants $c_i, c_i'$, with probability at least $1 - \delta/3$.

**Tool: Matrix Bernstein (self-adjoint dilation form) (Tropp, 2012).** Let $\{X_i\}$ be independent, mean-zero random $d \times d$ matrices with $\|X_i\| \leq L$ and $\|\sum_i \mathbb{E}[X_i X_i^*]\| \leq v$. Then for all $t > 0$,

$$\Pr\left( \left\| \sum_i X_i \right\| \geq t \right) \leq 2d \, \exp\left( -\frac{t^2/2}{v + Lt/3} \right),$$

**Proof of** (28). Fix $(k,k')$ and define $B_h := \overline{C}_{k',h} \otimes C_{k,h}$ (unitary, so $\|B_h\| = 1$). Consider

$$X_h := \frac{1}{M} (w_h - 1) B_h, \qquad \mathbb{E}[X_h] = 0, \qquad \|X_h\| \leq \frac{|w_h - 1|}{M} \leq \frac{1}{M}.$$

We have

$$\mathbb{E}[X_h X_h^*] = \frac{\mathbb{E}[(w_h - 1)^2]}{M^2} B_h B_h^* = \frac{\operatorname{Var}(w_h)}{M^2} I \preceq \frac{1}{pM^3} I.$$

Summing over $h$ gives variance proxy $v \leq M \cdot \frac{1}{pM^3} = \frac{1}{pM^2}$. Since $d \leq M$, with probability at least $1 - \delta/3$, Matrix Bernstein yields

$$\left\| \widehat{\mathsf{A}}_{k,k'} - \mathsf{A}_{k,k'} \right\|_{\text{op}} \lesssim \sqrt{\frac{\log(M/\delta)}{Mp}},$$

which is (28).

**Proof of** (29). Write

$$\widehat{\mathsf{A}}_{k,k'}^{(2)} - \widehat{\mathsf{A}}_{k,k'} = \frac{1}{M} \sum_h (w_h^2 - w_h) B_h = \underbrace{\frac{1}{M} \sum_h \left( (w_h - 1)^2 + (w_h - 1) \right) B_h}_{:=\Sigma_1 + \Sigma_2}.$$

For $\Sigma_2$ we reuse the argument of (28). For $\Sigma_1$, note that $\mathbb{E}[(w_h - 1)^2] = \mathrm{Var}(w_h) \leq 1/(pM)$, and $(w_h - 1)^2$ is sub-exponential with scale $\mathcal{O}(1/(pM))$, so matrix Bernstein again gives that with probability at least $1 - \delta/3$,

$$\|\Sigma_1\|_{\mathrm{op}} \lesssim \sqrt{\frac{\log(M/\delta)}{Mp}} + \frac{1}{Mp}.$$

Combining yields (29).

**Bounds for the mixed and noise terms in** (30). Conditional on $S$ and $\{w_h\}$, the $\{\varepsilon_h\}$ are independent, mean-zero, and

$$|\varepsilon_h| = \left| \langle R_h \Delta_h, S \rangle \right| \leq \|R_h \Delta_h\|_F \|S\|_F \leq \|\Delta_h\|_F \|S\|_F, \quad \mathbb{E}[\varepsilon_h^2 \mid S, w_h] \lesssim \frac{\|S\|_F^2}{p}.$$

Hence by scalar Bernstein (and Cauchy–Schwarz for the mixed sum),

$$\left| \sum_h w_h z_h \varepsilon_h \right| \lesssim \|z\|_2 \sqrt{\frac{M \log(M/\delta)}{p}}, \qquad \sum_h \varepsilon_h^2 \lesssim \frac{M \log(M/\delta)}{p},$$

with probability at least $1 - \delta/3$, which is (30).

Combine the above three bounds, we know that with probability at least $1 - \delta$, (28)–(30) hold at the same time.

**Stability of local maxima.** For the quadratic case (after mean removal), with the collinear and equal length $\mathbf{u}$ and $\mathbf{v}$ required by local maxima, $\mathcal{E}$ can be written as a positive semidefinite quadratic in the block masses $c_k$ (Eqn. 21):

$$\mathcal{E}(c) = \frac{M}{8} \sum_{k \neq 0} \frac{c_k^2}{d_k}, \qquad \sum_{k \neq 0} c_k = 1, \ c_k \geq 0.$$

The empirical energy has the form

$$\widehat{\mathcal{E}}(c) = \frac{M}{8} c^\top (D + E) c + \text{(terms independent of } c\text{)},$$

where $D = \mathrm{diag}(1/d_k)$ and $E$ is the symmetric perturbation induced by replacing $\mathsf{A}_{k,k'}$ with $\widehat{\mathsf{A}}_{k,k'}^{(2)}$ and by the mixed/noise terms. By (28)–(30),

$$\|E\|_{\mathrm{op}} \lesssim \sqrt{\frac{\log(M/\delta)}{Mp}} + \frac{1}{Mp} \tag{31}$$

with probability at least $1 - \delta$.

**Directional slope at a vertex (no gap needed).** Consider a pure-irrep vertex $c = \mathbf{e}_a$ and leak $\varepsilon$ mass to any other coordinate $b \neq a$: $c_a' = 1 - \varepsilon$, $c_b' = \varepsilon$, others 0. Population change:

$$\Delta \mathcal{E} = \frac{M}{8} \left( \frac{(1 - \varepsilon)^2 - 1}{d_a} + \frac{\varepsilon^2}{d_b} \right) = -\frac{M}{4d_a} \varepsilon + \mathcal{O}(\varepsilon^2).$$

Hence every leakage direction is strictly downhill at rate $\frac{M}{4d_a}$, even if multiple $d_k$ tie. Therefore, a first-order approximation of $\Delta \widehat{\mathcal{E}}$ is

$$\Delta \widehat{\mathcal{E}} = \Delta \mathcal{E} + \frac{M}{8} \Delta \left( c^\top E c \right) = -\frac{M}{4d_a} \varepsilon + \mathcal{O}(\varepsilon^2) + \frac{M}{4} \mathcal{O}\left( \|E\|_{\mathrm{op}} \varepsilon \right).$$

Therefore $\Delta\widehat{\mathcal{E}} < 0$ for all sufficiently small $\varepsilon > 0$ provided

$$\frac{M}{4}\|E\|_{\mathrm{op}} \; < \; \frac{M}{4d_a} \qquad \Longleftrightarrow \qquad \|E\|_{\mathrm{op}} \; < \; \frac{1}{d_a}.$$

Combining with (31), a sufficient sampling condition is

$$\sqrt{\frac{\log(M/\delta)}{Mp}} \; + \; \frac{1}{Mp} \; < \; \frac{1}{C\,d_a} \quad \Rightarrow \quad Mp \; \gtrsim \; d_a^2 \log\frac{M}{\delta},$$

for a universal numerical constant $C$. Since the total number of kept rows is $n = pM^2$, this is exactly

$$\boxed{\; n \; \gtrsim \; M\,d_a^2 \, \log\frac{M}{\delta} \;}$$

(up to universal constants). Under this condition, with probability at least $1 - \delta$, every pure-irrep vertex remains a strict local maximum of the empirical objective (energies shift by $\mathcal{O}(\sqrt{\log(M/\delta)/(Mp)})$). When several irreps have the same $d_k$ (tied energies), *which* one is the global maximizer may swap, but the local-maxima set is preserved. □

### F.4 MEMORIZATION

**Setting**. Fix a group element $h$. The admissible training pairs are $(g, g^{-1}h)$ for $g \in H$ with probabilities $p_g := p_{g,\,g^{-1}h}$ and a unique maximum at $g^*$, i.e., $p_{g^*} > p_g$ for all $g \neq g^*$. Let $w = [u;v] \in \mathbb{R}^{2M}$ with budget $\|u\|_2^2 + \|v\|_2^2 = 1$. Define the pair-sums $s_g := u_g + v_{g^{-1}h} \geq 0$. Then $\sum_g s_g^2 \leq 2$ and the (single-target) objective reduces to

$$F(s) \; := \; \sum_g p_g\,\sigma(s_g) \qquad \text{subject to} \qquad s_g \geq 0, \; \sum_g s_g^2 \leq 2,$$

where $\sigma \in C^1([0,\infty))$ is strictly increasing on $(0,\infty)$. Maximizing the energy $\mathcal{E}$ is equivalent (up to a fixed positive factor) to maximizing $F$.

**Lemma 5** (KKT characterization via $\phi = \sigma'/x$). *Assume $\sigma'(x) > 0$ for $x > 0$, and define $\phi(x) := \sigma'(x)/x$ for $x > 0$. Let $s^\star$ be an optimal solution. Then there exists $\lambda \geq 0$ such that for each $g$:*

$$p_g\,\phi\big(s_g^\star\big) \; = \; 2\lambda, \quad \text{if } s_g^\star > 0, \tag{32}$$

*Moreover, the budget is tight: $\sum_g (s_g^\star)^2 = 2$ (hence $\lambda > 0$). If $\phi$ is strictly monotone on $(0,\infty)$, then for every active coordinate $s_g^\star > 0$,*

$$s_g^\star \; = \; \phi^{-1}\!\left(\frac{2\lambda}{p_g}\right). \tag{33}$$

*Proof.* Consider the Lagrangian $L(s,\lambda,\mu) = \sum_g p_g\,\sigma(s_g) - \lambda(\sum_g s_g^2 - 2) - \sum_g \mu_g s_g$, with $\lambda \geq 0$, $\mu_g \geq 0$. Stationarity gives $p_g\,\sigma'(s_g) - 2\lambda s_g - \mu_g = 0$. If $s_g > 0$, then $\mu_g = 0$ and $p_g\,\sigma'(s_g) = 2\lambda s_g$, i.e., $p_g\,\phi(s_g) = 2\lambda$. If $s_g = 0$, complementary slackness allows $\mu_g \geq 0$ and the stationarity reads $p_g\,\sigma'(0) - \mu_g = 0$. Interpreting $\phi(0^+) := \lim_{x\downarrow 0}\sigma'(x)/x$ (possibly $+\infty$), the inequality $p_g\,\phi(0^+) \leq 2\lambda$ encodes the fact that activating $s_g > 0$ would violate the KKT balance. Since $\sigma' > 0$ and the objective is increasing in each $s_g$, the budget must be tight at optimum, hence $\sum_g s_g^2 = 2$ and $\lambda > 0$. If $\phi$ is strictly monotone, (32) uniquely determines $s_g$ as in (33). □

**Lemma 6** (Memorization vs. spreading by $\phi$-monotonicity). *Under the setup above and assuming $\phi(x) = \sigma'(x)/x$ is continuous on $(0,\infty)$:*

(A) *If $\phi$ is nondecreasing on $(0,\sqrt{2}]$, then the unique maximizer is the* memorization *(peaked) solution*

$$s_{g^*}^\star = \sqrt{2}, \qquad s_{g\neq g^*}^\star = 0,$$

*realized by $u = \frac{1}{\sqrt{2}}e_{g^*}$, $v = \frac{1}{\sqrt{2}}e_{(g^*)^{-1}h}$.*

*(B) If $\phi$ is strictly decreasing on $(0, \infty)$, then the unique maximizer* spreads *and is given by*

$$s_g^\star = \phi^{-1}\Big(\frac{2\lambda}{p_g}\Big) \quad (\text{for all } g \text{ with } 2\lambda/p_g < \phi(0^+)),$$

*and $s_g^\star = 0$ for any $g$ with $2\lambda/p_g \geq \phi(0^+)$ (if $\phi(0^+) < \infty$). The multiplier $\lambda > 0$ is uniquely determined by the budget $\sum_g (s_g^\star)^2 = 2$. In particular, if $\phi(0^+) = \infty$ (e.g., ReLU on $[0, \infty)$: $\phi(x) = 1/x$; SiLU: $\phi(x) = \frac{\text{sigmoid}(x)}{x} + \text{sigmoid}(x)(1 - \text{sigmoid}(x))$), then all coordinates are strictly positive and*

$$p_i > p_j \implies s_i^\star > s_j^\star > 0.$$

*Proof.* (A) *Peaking when $\phi$ is nondecreasing.* Take any feasible $s$ with two positive coordinates $s_i \geq s_j > 0$ and $p_i > p_j$. Define a squared-mass transfer preserving $\sum s_g^2$: $s_i(t) := \sqrt{s_i^2 + t}$, $s_j(t) := \sqrt{s_j^2 - t}$, and $\Psi(t) := p_i \sigma(s_i(t)) + p_j \sigma(s_j(t))$. Then

$$\Psi'(t) = \tfrac{1}{2}\big[p_i \phi(s_i(t)) - p_j \phi(s_j(t))\big] \geq \tfrac{1}{2}\big[(p_i - p_j)\phi(s_j(t))\big] > 0,$$

because $s_i(t) \geq s_j(t)$ and $\phi$ is nondecreasing. Hence $\Psi$ increases with $t$, so any two-support point can be strictly improved by pushing mass to the larger $p$. Iterating this collapse yields the single-support boundary $s_{g^*} = \sqrt{2}$, others $0$. Uniqueness follows from strict inequality and the uniqueness of $p_{g^*}$.

(B) *Spreading when $\phi$ is strictly decreasing.* By Lemma 5, the optimal active coordinates satisfy $p_g \phi(s_g^\star) = 2\lambda$. Since $\phi$ is strictly decreasing, $\phi^{-1}$ exists and is strictly decreasing, yielding $s_g^\star = \phi^{-1}(2\lambda/p_g)$ on the active set; complementary slackness gives the thresholding when $\phi(0^+) < \infty$. The budget $\sum_g (s_g^\star)^2 = 2$ fixes $\lambda$, and strict monotonicity implies the profile is strictly ordered by $p_g$. $\qquad\square$

**Theorem 5** (Memorization solution). *Let $\phi(x) := \sigma'(x)/x$ and assume $\sigma'(x) > 0$ for $x > 0$. For group arithmetic tasks, suppose we only collect sample $(g, g^{-1}h)$ for one target $h$ with probability $p_g$. Then the global optimal of $\mathcal{E}$ is a memorization solution, either (1) a* focused memorization $\mathbf{w} = \frac{1}{\sqrt{2}}(\mathbf{e}_{g^*}, \mathbf{e}_{g^{*-1}h})$ for $g^* = \arg\max p_g$ *if $\phi$ is nondecreasing, or (2) a* spreading memorization *with $\mathbf{w} = \frac{1}{2}\sum_g s_g[\mathbf{e}_g, \mathbf{e}_{g^{-1}h}]$, if $\phi$ is strictly decreasing. Here $s_g = \phi^{-1}(2\lambda/p_g)$ and $\lambda$ is determined by $\sum_g s_g^2 = 2$. No other local optima exist.*

*Proof.* The conclusion follows directly from Thm. 6. $\qquad\square$

**Some discussions**. We know that

- For power activations $\sigma(x) = x^q$ ($q \geq 2$) have $\phi(x) = q\, x^{q-2}$ nondecreasing; Thm. 6(A) gives memorization. In all these cases, the peaked solution is realized by even split $u = \frac{1}{\sqrt{2}} e_{g^*}$, $v = \frac{1}{\sqrt{2}} e_{(g^*)^{-1}h}$; any profile $s^\star$ can be realized with, e.g., $u_g = v_{g^{-1}h} = s_g^\star/2$.
- ReLU on $[0, \infty)$: $\sigma(x) = x$, $\phi(x) = 1/x$ strictly decreasing; Thm. 6(B) yields $s^\star \propto p$.
- SiLU/Swish/Tanh/Sigmoid: $\phi$ strictly decreasing with $\phi(0^+) = \infty$; Thm. 6(B) gives a strictly ordered spread $s_g^\star = \phi^{-1}(2\lambda/p_g)$.

## G INTERACTIVE FEATURE LEARNING (SEC. 5.1)

### G.1 FEATURE REPULSION (SEC. 5.1)

**Theorem 6** (Repulsion of similar features). *The $j$-th column of $\tilde{F}B$ is given by:*

$$[\tilde{F}B]_j = b_{jj}\tilde{\mathbf{f}}_j + \sum_{l=1}^{K} b_{jl}\tilde{\mathbf{f}}_l \tag{9}$$

*where $\text{sign}(b_{jl}) = -\text{sign}(\tilde{\mathbf{f}}_j^\top P_{\eta,-jl}\tilde{\mathbf{f}}_l)$. Here $P_{\eta,-jl} := I - \tilde{F}_{-jl}(\tilde{F}_{-jl}^\top \tilde{F}_{-jl} + \eta I)^{-1}\tilde{F}_{-jl}^\top$ is a projection matrix constructed from $\tilde{F}_{-jl}$, which is $\tilde{F}$ excluding the $l$-th and $j$-th columns.*

*Proof.* Let $Q := (\tilde{F}^\top \tilde{F} + \eta I)^{-1}$. Without loss of generality (by a column permutation similarity that preserves signs of the corresponding inverse entries), reorder columns so that the pair $(j, \ell)$ becomes $(1, 2)$. Write the partition

$$\tilde{F} = \begin{bmatrix} \tilde{\mathbf{f}}_1 & \tilde{\mathbf{f}}_2 & \tilde{F}_r \end{bmatrix}, \quad \tilde{F}_r := \tilde{F}_{-(1,2)} \in \mathbb{R}^{n \times (K-2)}.$$

Then the ridge Gram matrix $G = \tilde{F}^\top \tilde{F} + \eta I_K$ acquires the $2 \times 2$ / remainder block form

$$G = \begin{bmatrix} a & b & \mathbf{u}^\top \\ b & c & \mathbf{v}^\top \\ \mathbf{u} & \mathbf{v} & H \end{bmatrix}, \quad \text{where} \quad \begin{matrix} a := \tilde{\mathbf{f}}_1^\top \tilde{\mathbf{f}}_1 + \eta, & b := \tilde{\mathbf{f}}_1^\top \tilde{\mathbf{f}}_2, & \mathbf{u} := \tilde{F}_r^\top \tilde{\mathbf{f}}_1, \\ c := \tilde{\mathbf{f}}_2^\top \tilde{\mathbf{f}}_2 + \eta, & \mathbf{v} := \tilde{F}_r^\top \tilde{\mathbf{f}}_2, & H := \tilde{F}_r^\top \tilde{F}_r + \eta I. \end{matrix}$$

Because $\eta > 0$, $H$ is positive definite and hence invertible. The inverse of a block matrix is governed by the Schur complement. Define the $2 \times 2$ Schur complement

$$S := \begin{bmatrix} a & b \\ b & c \end{bmatrix} - \begin{bmatrix} \mathbf{u}^\top \\ \mathbf{v}^\top \end{bmatrix} H^{-1} \begin{bmatrix} \mathbf{u} & \mathbf{v} \end{bmatrix} = \begin{bmatrix} \alpha & \beta \\ \beta & \gamma \end{bmatrix},$$

where the entries are

$$\alpha = a - \mathbf{u}^\top H^{-1} \mathbf{u}, \qquad \beta = b - \mathbf{u}^\top H^{-1} \mathbf{v}, \qquad \gamma = c - \mathbf{v}^\top H^{-1} \mathbf{v}.$$

A standard block inversion formula (e.g., via Schur complements) yields that the top-left $2 \times 2$ block of $G^{-1}$ equals $S^{-1}$. In particular, the off–diagonal entry of $Q = G^{-1}$ for indices $(1, 2)$ is the off–diagonal entry of $S^{-1}$. Since

$$S^{-1} = \frac{1}{\alpha\gamma - \beta^2} \begin{bmatrix} \gamma & -\beta \\ -\beta & \alpha \end{bmatrix} \quad \text{with} \quad \alpha\gamma - \beta^2 > 0$$

(because $G \succ 0$ implies $S \succ 0$), we obtain

$$q_{12} = (S^{-1})_{12} = -\frac{\beta}{\alpha\gamma - \beta^2}.$$

It remains to identify $\alpha, \beta, \gamma$ in terms of ridge residuals with respect to $\tilde{F}_r$. Note that

$$H = \tilde{F}_r^\top \tilde{F}_r + \eta I \implies \tilde{F}_r H^{-1} \tilde{F}_r^\top = I_n - P_{\eta,r},$$

by the definition $P_{\eta,r} := I - \tilde{F}_r H^{-1} \tilde{F}_r^\top$. Therefore

$$\alpha = \tilde{\mathbf{f}}_1^\top \tilde{\mathbf{f}}_1 + \eta - \tilde{\mathbf{f}}_1^\top \tilde{F}_r H^{-1} \tilde{F}_r^\top \tilde{\mathbf{f}}_1 = \eta + \tilde{\mathbf{f}}_1^\top \left( I - \tilde{F}_r H^{-1} \tilde{F}_r^\top \right) \tilde{\mathbf{f}}_1 = \eta + \tilde{\mathbf{f}}_1^\top P_{\eta,r} \tilde{\mathbf{f}}_1,$$

$$\beta = \tilde{\mathbf{f}}_1^\top \tilde{\mathbf{f}}_2 - \tilde{\mathbf{f}}_1^\top \tilde{F}_r H^{-1} \tilde{F}_r^\top \tilde{\mathbf{f}}_2 = \tilde{\mathbf{f}}_1^\top \left( I - \tilde{F}_r H^{-1} \tilde{F}_r^\top \right) \tilde{\mathbf{f}}_2 = \tilde{\mathbf{f}}_1^\top P_{\eta,r} \tilde{\mathbf{f}}_2,$$

$$\gamma = \eta + \tilde{\mathbf{f}}_2^\top P_{\eta,r} \tilde{\mathbf{f}}_2.$$

Substituting these identities into the expression for $q_{12}$ gives

$$q_{12} = -\frac{\tilde{\mathbf{f}}_1^\top P_{\eta,r} \tilde{\mathbf{f}}_2}{\left( \eta + \tilde{\mathbf{f}}_1^\top P_{\eta,r} \tilde{\mathbf{f}}_1 \right) \left( \eta + \tilde{\mathbf{f}}_2^\top P_{\eta,r} \tilde{\mathbf{f}}_2 \right) - \left( \tilde{\mathbf{f}}_1^\top P_{\eta,r} \tilde{\mathbf{f}}_2 \right)^2}.$$

The denominator is strictly positive (it is the determinant of the positive definite $2 \times 2$ matrix $S$), hence

$$\text{sign}(q_{12}) = -\text{sign}(\tilde{\mathbf{f}}_1^\top P_{\eta,r} \tilde{\mathbf{f}}_2).$$

Undoing the preliminary permutation shows the same formula for the original indices $(j, \ell)$, which proves the sign claim.

Finally, since $Q$ is the inverse Gram with ridge, the $j$-th column of $\tilde{F}Q$ is

$$(\tilde{F}Q)_{\bullet j} = \sum_{m=1}^{K} q_{mj} \tilde{\mathbf{f}}_m = q_{jj} \tilde{\mathbf{f}}_j + \sum_{m \neq j} q_{mj} \tilde{\mathbf{f}}_m.$$

Because $q_{mj}$ has sign opposite to the ridge-residual similarity $\tilde{\mathbf{f}}_m^\top P_{\eta,-mj} \tilde{\mathbf{f}}_j$, features that are (residually) similar to $\tilde{\mathbf{f}}_j$ enter with negative coefficients and hence subtract from $(\tilde{F}Q)_{\bullet j}$ along those directions—"repelling" similar features and promoting specialization. This completes the proof. $\square$

## G.2 TOP-DOWN MODULATION (SEC. 5.2)

**Theorem 7** (Top-down Modulation). *For group arithmetic tasks with $\sigma(x) = x^2$, if the hidden layer learns only a subset $\mathcal{S}$ of irreps, then the backpropagated gradient $G_F \propto (\Phi_{\mathcal{S}} \otimes \mathbf{1}_M)(\Phi_{\mathcal{S}} \otimes \mathbf{1}_M)^* F$ (see proof for the definition of $\Phi_{\mathcal{S}}$), which yields a modified $\mathcal{E}_{\mathcal{S}}$ that only has local maxima on the missing irreps $k \notin \mathcal{S}$.*

*Proof.* Fix a nontrivial isotype (irrep) $k$ and we have

$$\hat{Y}^{(k)}_{(\cdot, h), h'} = \text{diag}\Big(R_h^\top \left(\Pi_k R_{h'} \Pi_k\right)\Big).$$

Since $\Pi_k$ is central and idempotent, it commutes with $R_{h'}$ and $\Pi_k^2 = \Pi_k$, hence

$$\Pi_k R_{h'} \Pi_k = \Pi_k R_{h'} = R_{h'} \Pi_k.$$

Expand the central idempotent in the group algebra using unitary irreps $\{C_k\}$ and characters $\chi_k$:

$$\Pi_k = \frac{d_k}{M} \sum_{g \in H} \overline{\chi_k(g)} R_g = \frac{d_k}{M} \sum_{g \in H} \chi_k(g^{-1}) R_g. \tag{34}$$

Therefore

$$\Pi_k R_{h'} = \frac{d_k}{M} \sum_{g \in H} \overline{\chi_k(g)} R_g R_{h'} = \frac{d_k}{M} \sum_{g \in H} \overline{\chi_k(g)} R_{gh'}.$$

Taking the diagonal after the left shift by $R_h^\top$ gives

$$\text{diag}\big(R_h^\top (\Pi_k R_{h'})\big) = \frac{d_k}{M} \sum_{g \in H} \overline{\chi_k(g)} \, \text{diag}\big(R_h^\top R_{gh'}\big).$$

Since $R_h^\top R_{gh'} = R_{h^{-1}gh'}$, we have

$$\text{diag}(R_h^\top R_{gh'}) = \begin{cases} \mathbf{1}_M, & h^{-1}gh' = e, \\ \mathbf{0}, & \text{otherwise.} \end{cases}$$

Only the unique term $g = hh'^{-1}$ survives, so

$$\text{diag}\big(R_h^\top (\Pi_k R_{h'})\big) = \frac{d_k}{M} \overline{\chi_k(hh'^{-1})} \mathbf{1}_M = \frac{d_k}{M} \chi_k(h'^{-1}h) \mathbf{1}_M,$$

where we used $\overline{\chi_k(a)} = \chi_k(a^{-1})$ for unitary irreps. Consequently,

$$\boxed{\hat{Y}^{(k)}_{(\text{rows for block } h), h'} = \frac{d_k}{M} \chi_k(h'^{-1}h) \mathbf{1}_M.}$$

Summing over a subset $\mathcal{S}$ of isotypes yields

$$\hat{Y}_{(\text{rows for block } h), h'} = \sum_{k \in \mathcal{S}} \hat{Y}^{(k)}_{(\text{rows for block } h), h'} = \frac{1}{M} \sum_{k \in \mathcal{S}} d_k \, \chi_k(h) \overline{\chi_k(h')} \mathbf{1}_M.$$

Since summing over all $k \neq 0$ leads to $\hat{Y} = \tilde{Y}$ (Thm. 3), for the residual $\hat{Y} - \tilde{Y}$ we have

$$[\hat{Y} - \tilde{Y}]_{(\text{rows for block } h), h'} = \frac{1}{M} \sum_{k \neq 0, k \notin \mathcal{S}} d_k \, \chi_k(h) \overline{\chi_k(h')} \mathbf{1}_M.$$

which means that $\hat{Y} - \tilde{Y} = \Phi_{\mathcal{S}} \Phi_{\mathcal{S}}^* \otimes \mathbf{1}_M$, where $\Phi_{\mathcal{S}} := \left[\sqrt{\frac{d_k}{M}} \chi_k(\cdot)\right]_{k \neq 0, k \notin \mathcal{S}} \in \mathbb{C}^{M \times (\kappa(H) - |\mathcal{S}| - 1)}$.

Since $\tilde{Y} = P_1^\perp \otimes \mathbf{1}_M$, we have:

$$G_F \propto (\hat{Y} - \tilde{Y})\tilde{Y}^\top F = \left(\Phi_{\mathcal{S}} \Phi_{\mathcal{S}}^* \otimes \mathbf{1}_M \mathbf{1}_M^\top\right) F = \left(\Phi_{\mathcal{S}} \otimes \mathbf{1}_M\right) \left(\Phi_{\mathcal{S}} \otimes \mathbf{1}_M\right)^* F$$

Therefore, the energy function $\mathcal{E}$ now becomes

$$\mathcal{E}_{\mathcal{S}} = \frac{1}{2}\|(\Phi_{\mathcal{S}} \otimes \mathbf{1}_M)^* F\|_2^2 = \frac{1}{2}\|\Phi_{\mathcal{S}}^* \mathbf{z}\|_2^2$$

where $\mathbf{z} = [z_h] = [\langle R_h, S\rangle_F] \in \mathbb{C}^M$ defined in Eqn. 13. Computing each row $k$ in $\Phi_{\mathcal{S}}^* \mathbf{z}$ and use the property of projection matrix $\Pi_k$ (Eqn. 34), we have:

$$[\Phi_{\mathcal{S}}^* \mathbf{z}]_k = \langle \sum_{h \in H} \sqrt{\frac{d_k}{M}} \overline{\chi_k(h)} R_h, S\rangle = \sqrt{\frac{M}{d_k}} \langle \Pi_k, S\rangle$$

In the $Q$ space, we have $\langle \Pi_k, S\rangle = \sum_{r=1}^{m_k} \operatorname{tr}(\hat{S}_{k,r})$ and therefore

$$\mathcal{E}_{\mathcal{S}} = \frac{1}{2} \sum_{k \neq 0, k \notin \mathcal{S}} \frac{M}{d_k} |\langle \Pi_k, S\rangle|^2 = \frac{M}{2} \sum_{k \neq 0, k \notin \mathcal{S}} \frac{1}{d_k} \Big| \sum_r \operatorname{tr}(\hat{S}_{k,r})\Big|^2$$

which is exactly the same form as the decomposition (Eqn. 19) in Thm. 2 (but a much cleaner derivation). Therefore, all the local maxima of $\mathcal{E}_{\mathcal{S}}$ are still in the same form as Thm. 2, but we just remove those local maxima that are in isotype/irreps $k \in \mathcal{S}$, and focus on missing ones. $\qquad\square$

### G.3 DIVERSITY ENHANCEMENT WITH MUON (SEC. 5.3)

**Lemma 2** (Muon optimizes the same as gradient flow). *Muon finds ascending direction to maximize the joint energy function $\mathcal{E}_{\text{joint}}(W) = \sum_j \mathcal{E}(\mathbf{w}_j)$ and has stationary points iff the original gradient $G_W$ vanishes.*

*Proof.* Let $G = [\nabla_{\mathbf{w}_1}\mathcal{E}, \nabla_{\mathbf{w}_2}\mathcal{E}, \ldots, \nabla_{\mathbf{w}_K}\mathcal{E}]$ be the gradient matrix. Let $G = UDV^\top$ be the singular value decomposition. Then Muon direction is $\hat{G} = UV^\top$ and thus the inner product between $\hat{G}$ and $G$ is

$$\langle \hat{G}, G\rangle_F = \operatorname{tr}(\hat{G}^\top G) = \operatorname{tr}(VU^\top UDV^\top) = \operatorname{tr}(D) \geq 0 \tag{35}$$

So Muon always follows the gradient direction and improve the objective. Furthermore, $\langle \hat{G}, G\rangle_F = 0$ iff $D = 0$, which means that $G = 0$. So the stationary points of the Muon dynamics and the original gradient dynamics are identical. $\qquad\square$

**Lemma 7** (Proposition of Fréchet / log-Gumbel selection). *Let $x_1, \ldots, x_n$ be i.i.d. positive random variables with Fréchet($\alpha$) CDF*

$$F(x) = \exp\big(-x^{-\alpha}\big), \qquad x > 0, \ \alpha > 0,$$

*and let $w_1, \ldots, w_n > 0$ be fixed weights. Define*

$$i^* = \arg\max_{1 \leq j \leq n} w_j x_j.$$

*Then*

$$\boxed{\Pr\big(i^* = i\big) = \frac{w_i^\alpha}{\sum_{j=1}^n w_j^\alpha}, \qquad i = 1, \ldots, n.}$$

*In particular, when $\alpha = 1$,*

$$\boxed{\Pr\big(i^* = i\big) = \frac{w_i}{\sum_{j=1}^n w_j}.}$$

*Proof.* Set $Y_j := w_j x_j$. For $t > 0$,

$$\Pr\big(\max_j Y_j \leq t\big) = \prod_{j=1}^n F\Big(\frac{t}{w_j}\Big) = \exp\Big(-\sum_{j=1}^n (w_j/t)^\alpha\Big).$$

Differentiating gives the density of the maximum:

$$f_{\max}(t) = \frac{d}{dt}\Pr\big(\max_j Y_j \leq t\big) = \Big(\sum_{j=1}^n \alpha\, w_j^\alpha\, t^{-\alpha-1}\Big) \exp\Big(-\sum_{j=1}^n (w_j/t)^\alpha\Big).$$

The density that "$i$ achieves the maximum at level $t$" is

$$f_{Y_i}(t) \prod_{j \neq i} F\left(\frac{t}{w_j}\right) = \alpha \, w_i^\alpha \, t^{-\alpha-1} \, \exp\left(-\sum_{j=1}^n (w_j/t)^\alpha\right).$$

Hence the conditional probability that $i$ is the argmax given $\max_j Y_j = t$ is

$$\Pr\left(i^* = i \mid \max_j Y_j = t\right) = \frac{\alpha \, w_i^\alpha \, t^{-\alpha-1}}{\sum_{j=1}^n \alpha \, w_j^\alpha \, t^{-\alpha-1}} = \frac{w_i^\alpha}{\sum_{j=1}^n w_j^\alpha},$$

which is independent of $t$. Averaging over $t$ yields the stated result. $\qquad\square$

**Lemma 8** (The properties of the dynamics in Eqn. 10). *The dynamics always converges to $\zeta_{l^*}$ for $l^* = \arg\max_l \mu_l \alpha_l(0)$. That is, the initial leader always win.*

*Proof.* Note that due to orthogonality of $\{\zeta_l\}$, the dynamics can be written as

$$\dot{\alpha}_j = \mu_j \alpha_j^2, \qquad \mu_j > 0,$$

with the constraint $\sum_{j=1}^L \alpha_j^2 \leq 1$. Define

$$r_j := \mu_j \alpha_j.$$

**Interior**. In the interior, we have

$$\dot{r}_j = \mu_j \dot{\alpha}_j = \mu_j(\mu_j \alpha_j^2) = r_j^2.$$

For any pair $i, k$ define the ratio

$$\rho_{ik} := \frac{r_i}{r_k}.$$

Its derivative is

$$\dot{\rho}_{ik} = \frac{\dot{r}_i}{r_k} - \frac{r_i}{r_k^2}\dot{r}_k = \frac{r_i^2}{r_k} - \frac{r_i}{r_k^2}r_k^2 = \rho_{ik}(r_i - r_k).$$

Equivalently,

$$\frac{d}{dt}\log\frac{r_i}{r_k} = r_i - r_k. \tag{1}$$

Thus if $r_\ell(0) > r_j(0)$, then $\frac{d}{dt}\log(r_\ell/r_j) > 0$ and $\rho_{\ell j}(t)$ is strictly increasing. Hence a strict leader in $r$ cannot be overtaken in the interior.

**Boundary region** ($\sum_j \alpha_j^2 = 1$). On the unit sphere, the projected dynamics is

$$\dot{\alpha}_j = \mu_j \alpha_j^2 - \lambda \alpha_j, \qquad \lambda = \sum_{k=1}^L \mu_k \alpha_k^3.$$

In terms of $r_j$,

$$\dot{r}_j = r_j(r_j - \nu), \qquad \nu = \sum_{k=1}^L \alpha_k^2 r_k = \sum_{k=1}^L \frac{r_k^2}{\mu_k^2} r_k.$$

For the ratio $\rho_{ik} = r_i/r_k$ we again obtain

$$\dot{\rho}_{ik} = \rho_{ik}(r_i - r_k) \implies \frac{d}{dt}\log\frac{r_i}{r_k} = r_i - r_k. \tag{2}$$

**Monotonicity of ratios**. From (1)–(2), if $r_\ell(0) > r_j(0)$ then

$$\frac{d}{dt}\log\frac{r_\ell}{r_j} > 0 \quad \forall t,$$

so $\rho_{\ell j}(t) = r_\ell(t)/r_j(t)$ is strictly increasing for every $j \neq \ell$. Thus a strict leader $\ell$ remains the unique leader for all time.

**Convergence to the vertex**. Define weights

$$w_j := \alpha_j^2 = \frac{r_j^2}{\mu_j^2}, \qquad \sum_j w_j = 1.$$

Their dynamics is

$$\dot{w}_j = 2w_j(r_j - \nu).$$

Taking ratios,

$$\frac{d}{dt} \log \frac{w_i}{w_k} = 2(r_i - r_k).$$

In particular, $\frac{w_\ell}{w_j}$ is strictly increasing for every $j \neq \ell$. Therefore

$$\frac{w_j(t)}{w_\ell(t)} \to 0 \quad (j \neq \ell),$$

implying $w_\ell(t) \to 1$ and $w_j(t) \to 0$. Hence

$$\boldsymbol{\alpha}(t) \to \mathbf{e}_\ell \qquad \text{as } t \to \infty.$$

$\square$

**Lemma 9** (Muon projection). *For the matrix $A = [Q, \mathbf{v}]$ where $Q$ is a column orthonormal matrix and $\mathbf{v}$ is a vector with small magnitude, its Muon regulated version $\hat{A} = [\hat{A}_1, \hat{\mathbf{v}}]$ takes the following form:*

$$\hat{\mathbf{v}} = \left( \frac{\mathbf{v}_\perp}{\|\mathbf{v}_\perp\|} + \frac{\mathbf{v}_\|}{1 + \|\mathbf{v}_\perp\|} \right) + O(\|\mathbf{v}_\perp\|^2) \tag{36}$$

*where $\mathbf{v}_\| = QQ^\top \mathbf{v}$ and $\mathbf{v}_\perp = I - QQ^\top \mathbf{v}$.*

*Proof.* Given $A = [Q, B]$ with $Q^\top Q = I_k$, write $B = QC + B_\perp$ where $C := Q^\top B \in \mathbb{R}^{k \times m}$ and $B_\perp := (I - QQ^\top)B$.

Let $T := B_\perp^\top B_\perp \succ 0$. For $c > 0$ define

$$\hat{A}^{(c)} = A(A^\top A)^{-1/c}, \qquad \hat{A}^{(c)} = [\hat{A}_1^{(c)}, \hat{A}_2^{(c)}].$$

We derive a first-order (in $C$) formula for the last block $\hat{A}_2^{(c)}$.

The exact Gram matrix is

$$G := A^\top A = \begin{bmatrix} I_k & C \\ C^\top & C^\top C + T \end{bmatrix} = G_0 + H, \qquad G_0 := \text{diag}(I_k, T), \quad H := \begin{bmatrix} 0 & C \\ C^\top & C^\top C \end{bmatrix}.$$

Treat $C$ as small. To first order in $C$ we may drop the quadratic block:

$$H = \begin{bmatrix} 0 & C \\ C^\top & 0 \end{bmatrix} + O(\|C\|^2).$$

**Diagonalizing** $T$. Let $T = U\Lambda U^\top$ with $\Lambda = \text{diag}(\lambda_1, \ldots, \lambda_m)$, $\lambda_j > 0$. Define the block orthogonal change of basis

$$P := \text{diag}(I_k, U) \quad \Rightarrow \quad \tilde{G} := P^\top G P, \quad \tilde{G}_0 := P^\top G_0 P = \text{diag}(I_k, \Lambda), \quad \tilde{H} := P^\top H P = \begin{bmatrix} 0 & \tilde{C} \\ \tilde{C}^\top & 0 \end{bmatrix},$$

where $\tilde{C} := CU$. All first-order statements can be done in this basis and then mapped back by $P$.

**First-order Taylor Expansion**. Now let's do the Taylor expansion. Write

$$\tilde{G} = \tilde{G}_0 + \tilde{H} = \tilde{G}_0^{1/2} \Big( I + \underbrace{\tilde{G}_0^{-1/2} \tilde{H} \tilde{G}_0^{-1/2}}_{=:E} \Big) \tilde{G}_0^{1/2}.$$

Since $\widetilde{G}_0 = \mathrm{diag}(I_k, \Lambda)$,

$$E = \begin{bmatrix} 0 & \widetilde{C}\,\Lambda^{-1/2} \\ \Lambda^{-1/2}\widetilde{C}^\top & 0 \end{bmatrix} \quad \text{is } O(\|C\|).$$

For the scalar function $f(x) = x^{-1/c}$,

$$(I + E)^{-1/c} = I - \frac{1}{c}E + O(\|E\|^2).$$

Therefore

$$\widetilde{G}^{-1/c} = \widetilde{G}_0^{-1/2}\,(I+E)^{-1/c}\,\widetilde{G}_0^{-1/2} = \widetilde{G}_0^{-1/c} - \frac{1}{c}\,\widetilde{G}_0^{-1/2}E\,\widetilde{G}_0^{-1/2} + O(\|C\|^2).$$

Compute the blocks using $\widetilde{G}_0^{-1/2} = \mathrm{diag}(I_k, \Lambda^{-1/2})$:

$$\widetilde{G}_0^{-1/2}E\,\widetilde{G}_0^{-1/2} = \begin{bmatrix} 0 & \widetilde{C}\,\Lambda^{-1} \\ \Lambda^{-1}\widetilde{C}^\top & 0 \end{bmatrix}.$$

Hence, to first order,

$$\widetilde{G}^{-1/c} = \begin{bmatrix} I_k & 0 \\ 0 & \Lambda^{-1/c} \end{bmatrix} - \frac{1}{c}\begin{bmatrix} 0 & \widetilde{C}\,\Lambda^{-1} \\ \Lambda^{-1}\widetilde{C}^\top & 0 \end{bmatrix} + O(\|C\|^2). \tag{37}$$

**Back to the original space**. Now

$$G^{-1/c} = P\,\widetilde{G}^{-1/c}\,P^\top.$$

Using (37) and $P = \mathrm{diag}(I_k, U)$,

$$G^{-1/c} = \begin{bmatrix} I_k & 0 \\ 0 & U\,\Lambda^{-1/c}\,U^\top \end{bmatrix} - \frac{1}{c}\begin{bmatrix} 0 & C\,U\,\Lambda^{-1}U^\top \\ U\,\Lambda^{-1}U^\top C^\top & 0 \end{bmatrix} + O(\|C\|^2).$$

Since $U\,\Lambda^{-1}U^\top = T^{-1}$ and $U\,\Lambda^{-1/c}U^\top = T^{-1/c}$,

$$G^{-1/c} = \begin{bmatrix} I_k & 0 \\ 0 & T^{-1/c} \end{bmatrix} - \frac{1}{c}\begin{bmatrix} 0 & C\,T^{-1} \\ T^{-1}C^\top & 0 \end{bmatrix} + O(\|C\|^2).$$

Now multiply

$$\widehat{A}^{(c)} = [\,Q,\ QC + B_\perp\,]\,G^{-1/c}.$$

Taking the *last $m$ columns* (the 2nd block) and keeping first-order terms:

$$\widehat{A}_2^{(c)} = Q\Big(-\frac{1}{c}C\,T^{-1}\Big) + (QC + B_\perp)\,T^{-1/c} + O(\|C\|^2)$$

$$= B_\perp\,T^{-1/c} + Q\Big(C\,T^{-1/c} - \frac{1}{c}C\,T^{-1}\Big) + O(\|C\|^2).$$

Factor the $Q$-part columnwise via the spectral calculus of $T$. If $T = U\Lambda U^\top$, then on each eigenvalue $\lambda$ the scalar factor is

$$\lambda^{-1/c} - \frac{1}{c}\lambda^{-1} = \frac{1 - \lambda^{1-1/c}}{1 - \lambda}.$$

Thus, in matrix form,

$$C\,T^{-1/c} - \frac{1}{c}C\,T^{-1} = C\,(I - T^{1-1/c})\,(I - T)^{-1}.$$

and we have

$$\boxed{\widehat{A}_2^{(c)} = B_\perp\,T^{-1/c} + B_\parallel\,(I - T^{1-1/c})\,(I - T)^{-1} + O(\|C\|^2).} \tag{38}$$

where $B_\parallel = QQ^\top B$.

For polar case $c = 2$, the operator becomes $(I - T^{1/2})(I - T)^{-1}$. For $B = \mathbf{v}$, we have $T = B_\perp^\top B_\perp = \|\mathbf{v}_\perp\|_2^2$ and the conclusion follows. $\qquad\square$

**Lemma 10** (Bound of $T_0$).

$$T_0 \geq \max\left(\min_{l=1}^{L} 1/p_l, L \sum_{l=1}^{L} 1/l\right). \tag{39}$$

*Proof.* $T_0 \geq \min_l 1/p_l$ since the expected time to collect all the coupons is always larger than collecting the rarest coupon alone.

To prove $T_0 \geq L \sum_{l=1}^{L} 1/l$, fix $t > 0$ and consider the function

$$h(p) = \log(1 - e^{-pt}), \qquad p > 0.$$

A direct computation shows

$$h''(p) = -\frac{t^2}{4\sinh^2(pt/2)} < 0,$$

so $h$ is concave. By Jensen's inequality and $\sum_i p_i = 1$,

$$\sum_{i=1}^{L} \log(1 - e^{-p_i t}) \leq L \log\left(1 - e^{-t/L}\right).$$

Exponentiating gives the pointwise bound

$$\prod_{i=1}^{L} (1 - e^{-p_i t}) \leq (1 - e^{-t/L})^L.$$

Therefore

$$\mathbb{E}[T_0] \geq \int_0^\infty \left(1 - (1 - e^{-t/L})^L\right) dt.$$

To evaluate the integral, set $u = e^{-t/L}$, so $dt = -L\, du/u$ and $t : 0 \to \infty$ maps to $u : 1 \to 0$:

$$\int_0^\infty \left(1 - (1 - e^{-t/L})^L\right) dt = L \int_0^1 \frac{1 - (1-u)^L}{u}\, du = L \int_0^1 \sum_{l=0}^{L-1} (1-u)^l du = L \sum_{l=0}^{L-1} \frac{1}{l+1}$$

Thus the conclusion holds. Equality holds if and only if $p_1 = \cdots = p_L = 1/L$, since that is the case of equality in Jensen. $\qquad\square$

**Theorem 8** (Muon rebalances gradient updates). *Consider the following dynamics (Tian, 2023):*

$$\dot{\mathbf{w}} = A(\mathbf{w})\mathbf{w}, \qquad \|\mathbf{w}\|_2 \leq 1 \tag{10}$$

*where $A(\mathbf{w}) := \sum_l \lambda_l(\mathbf{w})\boldsymbol{\zeta}_l\boldsymbol{\zeta}_l^\top$. Assume that (1) $\{\boldsymbol{\zeta}_l\}$ form orthonormal bases, (2) for $\mathbf{w} = \sum_l \alpha_l \boldsymbol{\zeta}_l$, we have $\lambda_l(\mathbf{w}) = \mu_l \alpha_l$ with $\mu_l \leq 1$, and (3) $\{\alpha_l\}$ is initialized from inverse-exponential distribution with $\mathrm{CDF}(x) = \exp(-x^{-a})$ with $a > 1$. Then*

- ***Independent feature learning.*** *$\Pr[\mathbf{w} \to \boldsymbol{\zeta}_l] = p_l := \mu_l^a / \sum_l \mu_l^a$. Then the expected #nodes to get all local maxima is $T_0 \geq \max\left(1/\min_l p_l, \sum_{l=1}^{L} 1/l\right)$.*

- ***Muon guiding.*** *If we use Muon optimizer to optimize $K$ nodes sequentially, then the expected #nodes to get all local maxima is $T_a = 2^{-a}T_0 + (1 - 2^{-a})L$. For large $a$, $T_a \sim L$.*

*Proof.* From Lemma 8, we know that the final mode $\boldsymbol{\zeta}_l$ that the nodes converge into is the one with largest initial $\alpha_l$:

$$\Pr[\mathbf{w} \to \boldsymbol{\zeta}_l] = \Pr[l = \arg\max_{l'} \mu_{l'}\alpha_{l'}(0)] \tag{40}$$

By Lemma 7, we have $\Pr[\mathbf{w} \to \boldsymbol{\zeta}_l] = p_l := \mu_l^a / \sum_l \mu_l^a$.

**Independent feature learning**. In this case, getting all local modes $\{\boldsymbol{\zeta}_l\}$ is identical to the coupon collector problem with $L$ coupons. With the property of the distribution (Lemma 7), we know that the probability of getting $l$-th local maxima is $p_l := \mu_l^a / \sum_l \mu_l^a$.

Therefore, the expected number of trials to collect all local maxima is (Flajolet et al., 1992):

$$T_0 = \int_0^{+\infty} \left( 1 - \prod_{l=1}^{L} (1 - e^{-p_l t}) \right) dt \tag{41}$$

Note that $T_0 \geq \max\left( 1/\min_l p_l, L \sum_{l=1}^{L} 1/l \right)$ (Lemma 39). Since each node is independently optimized, we need $K \sim T_0$ to collect all local maxima in $K$ hidden nodes with high probability.

**Muon guiding**. Consider the following setting that we optimize the hidden nodes "incrementally". When learning the weights of node $j$, we assume all the previous nodes (node 1 to node $j-1$) have been learned, i.e., they have converged to one of the ground truth bases $\{\zeta_l\}$, but still keep the gradients of them (after deduplication) in the Muon update. Let $S_{j-1} \subseteq [L] = \{1, \ldots, L\}$ be the subset of local maxima that have been collected.

By Lemma 9, we know that

$$\hat{\mathbf{g}}_j = \frac{1}{\|\mathbf{g}_{j,\perp}\|} \left( \mathbf{g}_{j,\perp} + \frac{\|\mathbf{g}_{j,\perp}\|}{1 + \|\mathbf{g}_{j,\perp}\|} \mathbf{g}_{j,\parallel} \right) + O(\|\mathbf{g}_{j,\perp}\|^2) \tag{42}$$

where $\mathbf{g}_{j,\parallel} = P_{j-1} P_{j-1}^\top \mathbf{g}_j$ and $\mathbf{g}_{j,\perp} = \mathbf{g}_j - \mathbf{g}_{j,\parallel}$. Here $P_{j-1} = [\zeta_s]_{s \in S_{j-1}}$ is the projection matrix formed by the previous $j-1$ nodes. Since

$$\|\mathbf{g}_{j,\perp}\| \leq \|\mathbf{g}_j\| = \|\sum_l \lambda_l(\alpha_l) \alpha_l \zeta_l\| = |\sum_l (\lambda_l(\alpha_l)\alpha_l)^2| \leq |\sum_l \alpha_l^2| \leq 1 \tag{43}$$

We have $\frac{\|\mathbf{g}_{j,\perp}\|}{1+\|\mathbf{g}_{j,\perp}\|} \leq 1/2$. Therefore, this means that the parallel components, i.e., the components that are duplicated with the previous $j-1$ nodes in the gradient was suppressed by at least $1/2$, compared to the orthogonal components (i.e., the directions towards new local maxima). This is equivalent to dividing $\mu_l$ for all $l$s that appear in $P_{j-1}$ by (at least) 2. By Lemma 7, for the node $j$, the probability of converging to a new local maximum other than $S_{j-1}$ is

$$p_{new, S_{j-1}} \geq \frac{\sum_{l \notin S_{j-1}} p_l}{2^{-a} \sum_{l \in S_{j-1}} p_l + \sum_{l \notin S_{j-1}} p_l} \tag{44}$$

We do this sequentially starting from node $j$, then node $j+1$, etc. Let $m = |S_{j-1}|$ be the number of discovered local maxima. Then the expected time that we find a new local maxima is:

$$\mathbb{E}[\tilde{T}_{m \to m+1}] = \frac{1}{p_{new, S_{j-1}}} \leq 2^{-a} \mathbb{E}[T_{m \to m+1}] + 1 - 2^{-a} \tag{45}$$

where $\mathbb{E}[T_{m \to m+1}] = 1/\sum_{l \notin S_{j-1}} p_l$ is the expected time for the original coupon collector problem to pick a new local maximum, given $S_{j-1}$ known ones. Adding the expected time together, we have

$$T_a = \sum_{m=0}^{L-1} \mathbb{E}[\tilde{T}_{m \to m+1}] \leq 2^{-a} T_0 + (1 - 2^{-a}) L \tag{46}$$

Note that all the expected time are conditioned on the sequence of known local maxima. But since these values are independent of the specific sequence, they are also the expected time overall. $\quad\square$

# H   MORE DETAILED ANALYSIS ON STAGE I (LAZY LEARNING)

To analyze the Stage I more thoroughly, we consider the gradient-flow dynamics of the output layer weights $V$.

Let $\tilde{F} \in \mathbb{R}^{n \times K}$ be a fixed feature matrix and $\tilde{Y} \in \mathbb{R}^{n \times M}$.

We assume throughout that

(A1) $\tilde{F}$ has full column rank $K$, and
(A2) $\operatorname{col}(\tilde{Y}) \subseteq \operatorname{col}(\tilde{F})$, i.e. there exists $V^\star \in \mathbb{R}^{K \times M}$ such that $\tilde{Y} = \tilde{F} V^\star$.

(A3) Small and independent random initialization on entries of $V(0)$, with mean zero and variance $\alpha^2$, where $0 < \alpha \ll 1$, and thus $\|V(0)\|_F = O(\alpha\sqrt{KM})$ with high probability.

(A4) Zero-mean centering: $\mathbf{1}^\top \tilde{F} = 0$ and $\mathbf{1}^\top \tilde{Y} = 0$.

Note that (A4) is optional. It simplifies some interpretations but is not needed for the main analysis.

We train a linear readout $V \in \mathbb{R}^{K \times M}$ by minimizing

$$J(V) \;=\; \frac{1}{2}\big(\|\tilde{Y} - \tilde{F}V\|_F^2 + \eta\|V\|_F^2\big), \qquad \eta \geq 0. \tag{47}$$

We define the (matrix) prediction error and the backpropagated gradient $G_{\tilde{F}}$ as

$$E(t) := \tilde{Y} - \tilde{F}V(t) \in \mathbb{R}^{n \times M}, \qquad G_{\tilde{F}}(t) := E(t)V(t)^\top \in \mathbb{R}^{n \times K}. \tag{48}$$

Note that in the main text, we use $G_F$ to denote the backpropagated gradient on the uncentered feature matrix $F$, i.e., $G_F = P_1^\perp G_{\tilde{F}}$, where $P_1^\perp := I - \mathbf{1}\mathbf{1}^\top/n$ is the zero-mean projection matrix along the sample dimension. As we will see below, the leading term of $G_{\tilde{F}}$ is $\tilde{Y}\tilde{Y}^\top\tilde{F}$ and thus

$$G_F = P_1^\perp G_{\tilde{F}} \propto P_1^\perp \tilde{Y}\tilde{Y}^\top\tilde{F} = \tilde{Y}\tilde{Y}^\top\tilde{F} = \tilde{Y}\tilde{Y}^\top F. \tag{49}$$

because $\mathbf{1}^\top \tilde{F} = 0$ and $\mathbf{1}^\top \tilde{Y} = 0$.

We consider continuous–time gradient flow for $V$:

$$\frac{dV(t)}{dt} = -\nabla_V J(V(t)). \tag{50}$$

The gradient of $J$ with respect to $V$ is

$$\nabla_V J(V) = \tilde{F}^\top(\tilde{F}V - \tilde{Y}) + \eta V = AV - B, \quad A := \tilde{F}^\top\tilde{F} + \eta I_K, \quad B := \tilde{F}^\top\tilde{Y}. \tag{51}$$

We study the *gradient flow* dynamics

$$\frac{dV}{dt} = -\nabla_V J(V) = -AV + B. \tag{52}$$

Define the error matrix and the backpropagated gradient on $\tilde{F}$ by

$$E(t) := \tilde{Y} - \tilde{F}V(t) \in \mathbb{R}^{n \times M}, \qquad G_{\tilde{F}}(t) := E(t)V(t)^\top \in \mathbb{R}^{n \times K}.$$

Our goal is to understand:

1. the *small–time expansion* of $G_{\tilde{F}}(t)$ and show that the leading term is $\tilde{Y}\tilde{Y}^\top\tilde{F}$; and
2. the *long–time decay* behavior of $G_{\tilde{F}}(t)$, for both $\eta = 0$ and $\eta > 0$.

## H.1 The dynamics of $G_{\tilde{F}}$ at initial time stamps

### H.1.1 Small–time expansion and leading term

Write the Taylor expansions at $t = 0$ as

$$V(t) = V_0 + tV_1 + O(t^2), \qquad E(t) = E_0 + tE_1 + O(t^2),$$

where $V_0 := V(0)$ and $E_0 := \tilde{Y} - \tilde{F}V_0$. From (52),

$$V_1 = \frac{dV}{dt}\Big|_{t=0} = -AV_0 + B = -(\tilde{F}^\top\tilde{F} + \eta I_K)V_0 + \tilde{F}^\top\tilde{Y}. \tag{53}$$

Differentiating $E(t) = \tilde{Y} - \tilde{F}V(t)$ gives

$$E_1 = \frac{dE}{dt}\Big|_{t=0} = -\tilde{F}V_1 = \tilde{F}(\tilde{F}^\top\tilde{F} + \eta I_K)V_0 - \tilde{F}\tilde{F}^\top\tilde{Y}. \tag{54}$$

Now expand $G_{\tilde{F}}(t)$:

$$G_{\tilde{F}}(t) = E(t)V(t)^\top = (E_0 + tE_1)(V_0 + tV_1)^\top + O(t^2) = E_0V_0^\top + t(E_0V_1^\top + E_1V_0^\top) + O(t^2).$$

Using $E_0 = \tilde{Y} - \tilde{F}V_0$ and $V_1$ from (53),

$$\begin{aligned}
E_0V_1^\top &= (\tilde{Y} - \tilde{F}V_0)\big(-V_0^\top(\tilde{F}^\top\tilde{F} + \eta I_K) + \tilde{Y}^\top\tilde{F}\big) \\
&= \tilde{Y}\tilde{Y}^\top\tilde{F} - \tilde{F}V_0\tilde{Y}^\top\tilde{F} - \tilde{Y}V_0^\top(\tilde{F}^\top\tilde{F} + \eta I_K) + \tilde{F}V_0V_0^\top(\tilde{F}^\top\tilde{F} + \eta I_K).
\end{aligned}$$

Every term except $\tilde{Y}\tilde{Y}^\top\tilde{F}$ contains (at least one factor of) $V_0$, hence is $O(\alpha)$ in Frobenius norm. Moreover, $E_1V_0^\top$ also contains $V_0$:

$$E_1V_0^\top = \tilde{F}(\tilde{F}^\top\tilde{F} + \eta I_K)V_0V_0^\top - \tilde{F}\tilde{F}^\top\tilde{Y}V_0^\top,$$

so $\|E_1V_0^\top\|_F = O(\alpha)$ as well.

We therefore obtain the small–time expansion

$$G_{\tilde{F}}(t) = \underbrace{\tilde{Y}V_0^\top}_{O(\alpha)} + t\,\tilde{Y}\tilde{Y}^\top\tilde{F} + t\,R_1(V_0) + O(t^2), \tag{55}$$

where $R_1(V_0)$ collects all order–$t$ terms that contain $V_0$ and thus satisfy $\|R_1(V_0)\|_F = O(\alpha)$.

### H.1.2   WHY $\tilde{Y}\tilde{Y}^\top\tilde{F}$ IS THE LEADING TERM

We now compare the deterministic term $\tilde{Y}\tilde{Y}^\top\tilde{F}$ to the $V_0$–dependent terms using norm inequalities.

**Lemma 11** (Lower bound on $\|\tilde{Y}\tilde{Y}^\top\tilde{F}\|_F$). *Let $\tilde{F}$ have full column rank and $\tilde{Y}$ be nonzero. Then*

$$\|\tilde{Y}\tilde{Y}^\top\tilde{F}\|_F \geq \sigma_{\min}(\tilde{F})\,\|\tilde{Y}\tilde{Y}^\top\|_F > 0,$$

*where $\sigma_{\min}(\tilde{F})$ is the smallest singular value of $\tilde{F}$.*

*Proof.* For any matrices $A, B$, $\|AB\|_F^2 = \operatorname{tr}(BB^\top A^\top A)$. Take $A = \tilde{Y}\tilde{Y}^\top$, $B = \tilde{F}$. Since $BB^\top$ is PSD with eigenvalues bounded below by $\sigma_{\min}(\tilde{F})^2$,

$$\|AB\|_F^2 = \operatorname{tr}(BB^\top A^\top A) \geq \sigma_{\min}(\tilde{F})^2\|A\|_F^2 = \sigma_{\min}(\tilde{F})^2\|A\|_F^2.$$

Taking square roots gives the result. $\qquad\square$

Next, bound the $V_0$–dependent part. For concreteness, consider the term $\tilde{F}\tilde{F}^\top\tilde{Y}V_0^\top$ (other mixed terms are bounded similarly). Using $\|AB\|_F \leq \|A\|_F\|B\|_F$,

$$\|\tilde{F}\tilde{F}^\top\tilde{Y}V_0^\top\|_F \leq \|\tilde{F}\tilde{F}^\top\tilde{Y}\|_F\|V_0\|_F.$$

Under the iid initialization with variance $\alpha^2$, $\|V_0\|_F = O(\alpha\sqrt{KM})$, hence

$$\|\tilde{F}\tilde{F}^\top\tilde{Y}V_0^\top\|_F = O(\alpha).$$

The same argument applies to all other $V_0$–dependent order–$t$ terms in $R_1(V_0)$.

Combining Lemma 11 with these upper bounds yields

$$\frac{\|R_1(V_0)\|_F}{\|\tilde{Y}\tilde{Y}^\top\tilde{F}\|_F} \leq C(\tilde{F}, \tilde{Y}, K, M)\,\alpha$$

for some constant $C$ independent of $\alpha$. Thus, in the limit $\alpha \to 0$ (small random initialization), the term $\tilde{Y}\tilde{Y}^\top\tilde{F}$ is the unique leading contribution at order $t$.

**Proposition 2** (Small–time leading term of $G_{\tilde{F}}$). *Under assumptions (A1)–(A2) and small random initialization with scale $\alpha \ll 1$,*

$$G_{\tilde{F}}(t) = \tilde{Y}V_0^\top + t\,\tilde{Y}\tilde{Y}^\top\tilde{F} + O(t\alpha + t^2)$$

*in Frobenius norm. In particular, as $\alpha \to 0$,*

$$\frac{G_{\tilde{F}}(t) - \tilde{Y}V_0^\top}{t} \to \tilde{Y}\tilde{Y}^\top\tilde{F}, \qquad \text{and} \qquad \|G_{\tilde{F}}(t)\|_F \sim t\,\|\tilde{Y}\tilde{Y}^\top\tilde{F}\|_F$$

*for fixed small $t$, independently of whether $\eta = 0$ or $\eta > 0$.*

**Remark on the role of $\eta$.** The weight decay parameter $\eta$ only appears in products involving $V_0$, and hence all $\eta$–dependent order–$t$ contributions are also $O(\alpha)$ in norm. Therefore the leading deterministic term $\tilde{Y}\tilde{Y}^\top \tilde{F}$ is the same for both $\eta = 0$ and $\eta > 0$.

## H.2 Long–time decay of $G_{\tilde{F}}$

We now analyze the behavior of $G_{\tilde{F}}(t)$ as $t \to \infty$, again for both $\eta = 0$ and $\eta > 0$.

### H.2.1 General solution of the gradient flow

From (52), the gradient flow is a linear ODE with constant coefficients. The unique fixed point $V^\star$ satisfies
$$AV^\star = B \quad \Rightarrow \quad V^\star = A^{-1}B.$$
Define $\Delta V(t) := V(t) - V^\star$. Then
$$\frac{d}{dt}\Delta V(t) = -A\Delta V(t), \qquad \Delta V(t) = e^{-At}\Delta V(0),$$
and hence
$$V(t) = e^{-At}(V(0) - V^\star) + V^\star. \tag{56}$$

Let $\lambda_{\min}(A)$ denote the smallest eigenvalue of $A$. Since $A \succeq \tilde{F}^\top \tilde{F}$ and $\tilde{F}$ has full column rank, $\lambda_{\min}(A) \geq \sigma_{\min}(\tilde{F})^2$ for $\eta = 0$ and $\lambda_{\min}(A) \geq \sigma_{\min}(\tilde{F})^2 + \eta$ for $\eta > 0$. Standard bounds on matrix exponentials give
$$\|\Delta V(t)\|_F \leq e^{-\lambda_{\min}(A)t}\|\Delta V(0)\|_F. \tag{57}$$

The error satisfies
$$E(t) = \tilde{Y} - \tilde{F}V(t) = \tilde{Y} - \tilde{F}V^\star - \tilde{F}\Delta V(t) =: E^\star - \tilde{F}\Delta V(t),$$
where $E^\star := \tilde{Y} - \tilde{F}V^\star$ is the residual at the minimizer. Using $\|\tilde{F}\Delta V(t)\|_F \leq \|\tilde{F}\|_2\|\Delta V(t)\|_F$ and (57),
$$\|E(t) - E^\star\|_F \leq \|\tilde{F}\|_2\, e^{-\lambda_{\min}(A)t}\,\|\Delta V(0)\|_F. \tag{58}$$

### H.2.2 Case $\eta = 0$

When $\eta = 0$, we have $A = \tilde{F}^\top \tilde{F}$. By assumption (A2), $\tilde{Y} = \tilde{F}V^\star$ is exactly realized by the model, so $E^\star = 0$, i.e.
$$\lim_{t\to\infty} E(t) = 0.$$
Equations (57) and (58) imply exponential decay:
$$\|V(t) - V^\star\|_F \leq e^{-\sigma_{\min}(\tilde{F})^2 t}\|V(0) - V^\star\|_F, \qquad \|E(t)\|_F \leq \|\tilde{F}\|_2 e^{-\sigma_{\min}(\tilde{F})^2 t}\|V(0) - V^\star\|_F.$$

We can now bound $G_{\tilde{F}}(t)$:
$$G_{\tilde{F}}(t) = E(t)V(t)^\top,$$
so
$$\|G_{\tilde{F}}(t)\|_F \leq \|E(t)\|_F\|V(t)\|_2 \leq \|E(t)\|_F\left(\|V^\star\|_2 + \|V(t) - V^\star\|_2\right). \tag{59}$$
Using the exponential bounds above and the fact that $\|V(t) - V^\star\|_2 \leq \|V(t) - V^\star\|_F$, we obtain
$$\|G_{\tilde{F}}(t)\|_F \leq C_0\, e^{-\sigma_{\min}(\tilde{F})^2 t}$$
for some constant $C_0$ depending on $\tilde{F}$, $V^\star$ and $V(0)$ but not on $t$.

Thus in the realizable, unregularized case, the backpropagated gradient decays exponentially to zero.

**Proposition 3** (Exponential decay of $G_{\tilde{F}}$ for $\eta = 0$). *Assume (A1)–(A2) and $\eta = 0$. Then*
$$\lim_{t\to\infty} G_{\tilde{F}}(t) = 0,$$
*and there exists $C_0 > 0$ such that*
$$\|G_{\tilde{F}}(t)\|_F \leq C_0\, e^{-\sigma_{\min}(\tilde{F})^2 t} \qquad \text{for all } t \geq 0.$$

A more refined analysis using the SVD $\tilde{F} = U\Sigma W^\top$ shows that every singular direction of $G_{\tilde{F}}(t)$ is a finite linear combination of exponentials $e^{-(\sigma_i^2 + \sigma_{i'}^2)t}$ and $e^{-\sigma_i^2 t}$, so the slowest rate in the Frobenius norm is indeed $e^{-\sigma_{\min}(\tilde{F})^2 t}$.

### H.2.3 CASE $\eta > 0$

When $\eta > 0$, the minimizer $V^\star = A^{-1}\tilde{F}^\top \tilde{Y}$ is the ridge solution. In general it does *not* exactly interpolate $\tilde{Y}$, and the residual

$$E^\star := \tilde{Y} - \tilde{F}V^\star$$

is nonzero. Consequently the limiting backpropagated gradient

$$G_{\tilde{F}}^\star := \lim_{t\to\infty} G_{\tilde{F}}(t) = E^\star V^{\star\top}$$

is also nonzero in general.

To study the convergence, write

$$G_{\tilde{F}}(t) - G_{\tilde{F}}^\star = E(t)V(t)^\top - E^\star V^{\star\top} = (E(t) - E^\star)V(t)^\top + E^\star(V(t) - V^\star)^\top.$$

Using (57)–(58) and $\|AB\|_F \le \|A\|_F\|B\|_2$, we obtain

$$\|G_{\tilde{F}}(t) - G_{\tilde{F}}^\star\|_F \le \|E(t) - E^\star\|_F \|V(t)\|_2 + \|E^\star\|_F \|V(t) - V^\star\|_2$$

$$\le \Big(\|\tilde{F}\|_2\|V(0) - V^\star\|_F \|V(t)\|_2 + \|E^\star\|_F\|V(0) - V^\star\|_F\Big)e^{-\lambda_{\min}(A)t}.$$

Since $\|V(t)\|_2$ is bounded (it converges to $\|V^\star\|_2$), this shows exponential convergence of $G_{\tilde{F}}(t)$ to $G_{\tilde{F}}^\star$. Therefore, we have the following proposition:

**Proposition 4** (Exponential convergence of $G_{\tilde{F}}$ for $\eta > 0$). *Assume (A1)–(A2) and $\eta > 0$. Then*

$$\lim_{t\to\infty} G_{\tilde{F}}(t) = G_{\tilde{F}}^\star := E^\star V^{\star\top} \ne 0 \text{ in general},$$

*and there exists $C_1 > 0$ such that*

$$\|G_{\tilde{F}}(t) - G_{\tilde{F}}^\star\|_F \le C_1\, e^{-\lambda_{\min}(A)t}, \qquad \lambda_{\min}(A) \ \ge \ \sigma_{\min}(\tilde{F})^2 + \eta.$$

*Finally, note that*

$$G_{\tilde{F}}^\star = E^\star V^{\star\top} = P_\eta \tilde{Y}V^{\star\top} = \eta(\tilde{F}\tilde{F}^\top + \eta I)^{-1}\tilde{Y}\tilde{Y}^\top \tilde{F}(\tilde{F}^\top \tilde{F} + \eta I)^{-1} \tag{60}$$

*where $P_\eta := I - \tilde{F}(\tilde{F}^\top \tilde{F} + \eta I)^{-1}\tilde{F}^\top = \eta(\tilde{F}\tilde{F}^\top + \eta I)^{-1}$, by Woodbury matrix formula.*

**Summary.**

- For small $t$, the leading term in $G_{\tilde{F}}(t)$ is $t\tilde{Y}\tilde{Y}^\top \tilde{F}$, independent of $\eta$. All other terms (including those involving $V(0)$ and $\eta$) are lower order in the initialization scale $\alpha$.

- For $\eta = 0$ and realizable $\tilde{Y} \in \text{col}(\tilde{F})$, both the error $E(t)$ and $G_{\tilde{F}}(t)$ decay exponentially to zero at rate at least $\sigma_{\min}(\tilde{F})^2$.

- For $\eta > 0$, $E(t)$ and $V(t)$ converge exponentially to $(E^\star, V^\star)$, and $G_{\tilde{F}}(t)$ converges exponentially to a nonzero limit $G_{\tilde{F}}^\star = E^\star V^{\star\top}$.

### H.3 SCALING THE NETWORK OUTPUT BY $\beta$

We now modify the objective by inserting a scalar factor $\beta > 0$ on the network output:

$$J(V) \ = \ \frac{1}{2}\big\|\tilde{Y} - \beta\tilde{F}V\big\|_F^2, \qquad \tilde{F} \in \mathbb{R}^{n\times K}, \ \tilde{Y} \in \mathbb{R}^{n\times M}, \ V \in \mathbb{R}^{K\times M}. \tag{61}$$

We study the gradient flow dynamics for $V$ and, in particular, the initial behavior of the backpropagated gradient on $\tilde{F}$,

$$G_{\tilde{F}}(t) := E(t)V(t)^\top, \qquad E(t) := \tilde{Y} - \beta\tilde{F}V(t).$$

As before, we assume a small random initialization $V(0) = V_0$ whose entries are iid, mean zero, with variance $\alpha^2$, so that $\|V_0\|_F = O(\alpha\sqrt{KM})$ for $\alpha \ll 1$.

**Gradient flow with output scaling**. The gradient of $J$ with respect to $V$ is

$$\nabla_V J(V) = \beta \tilde{F}^\top (\beta \tilde{F} V - \tilde{Y}) = \beta^2 \tilde{F}^\top \tilde{F} V - \beta \tilde{F}^\top \tilde{Y}. \tag{62}$$

The gradient flow is

$$\frac{dV}{dt} = -\nabla_V J(V) = -\beta^2 \tilde{F}^\top \tilde{F} V + \beta \tilde{F}^\top \tilde{Y}. \tag{63}$$

We define

$$E(t) := \tilde{Y} - \beta \tilde{F} V(t), \qquad G_{\tilde{F}}(t) := E(t) V(t)^\top.$$

At $t = 0$,

$$V_0 := V(0), \qquad E_0 := E(0) = \tilde{Y} - \beta \tilde{F} V_0.$$

**Small–time expansion of $G_{\tilde{F}}(t)$** We expand

$$V(t) = V_0 + t V_1 + O(t^2), \qquad E(t) = E_0 + t E_1 + O(t^2),$$

where $V_1 = \frac{dV}{dt}\big|_{t=0}$ and $E_1 = \frac{dE}{dt}\big|_{t=0}$. From (63),

$$V_1 = -\beta^2 \tilde{F}^\top \tilde{F} V_0 + \beta \tilde{F}^\top \tilde{Y}. \tag{64}$$

Differentiating $E(t) = \tilde{Y} - \beta \tilde{F} V(t)$ gives

$$E_1 = \frac{dE}{dt}\Big|_{t=0} = -\beta \tilde{F} V_1 = \beta^3 \tilde{F} \tilde{F}^\top \tilde{F} V_0 - \beta^2 \tilde{F} \tilde{F}^\top \tilde{Y}. \tag{65}$$

Now expand $G_{\tilde{F}}(t)$:

$$\begin{aligned}
G_{\tilde{F}}(t) &= E(t) V(t)^\top \\
&= (E_0 + t E_1)(V_0 + t V_1)^\top + O(t^2) \\
&= E_0 V_0^\top + t\big(E_0 V_1^\top + E_1 V_0^\top\big) + O(t^2).
\end{aligned}$$

*Zeroth order in $t$.* Using $E_0 = \tilde{Y} - \beta \tilde{F} V_0$,

$$G_{\tilde{F}}(0) = E_0 V_0^\top = \tilde{Y} V_0^\top - \beta \tilde{F} V_0 V_0^\top. \tag{66}$$

Thus at $t = 0$ we have an $O(\beta^0)$ term and an $O(\beta^1)$ term, both proportional to the random initialization.

*First order in $t$.* We next compute $E_0 V_1^\top$ and $E_1 V_0^\top$ and keep track of all powers of $\beta$.

From (64), $V_1^\top = -\beta^2 V_0^\top \tilde{F}^\top \tilde{F} + \beta \tilde{Y}^\top \tilde{F}$, hence

$$\begin{aligned}
E_0 V_1^\top &= (\tilde{Y} - \beta \tilde{F} V_0)\big(-\beta^2 V_0^\top \tilde{F}^\top \tilde{F} + \beta \tilde{Y}^\top \tilde{F}\big) \\
&= \tilde{Y}(-\beta^2 V_0^\top \tilde{F}^\top \tilde{F}) + \tilde{Y}(\beta \tilde{Y}^\top \tilde{F}) - \beta \tilde{F} V_0(-\beta^2 V_0^\top \tilde{F}^\top \tilde{F}) - \beta \tilde{F} V_0(\beta \tilde{Y}^\top \tilde{F}) \\
&= \underbrace{\beta \tilde{Y} \tilde{Y}^\top \tilde{F}}_{\text{no } V_0, \text{ order } \beta} - \beta^2 \tilde{Y} V_0^\top \tilde{F}^\top \tilde{F} + \beta^3 \tilde{F} V_0 V_0^\top \tilde{F}^\top \tilde{F} - \beta^2 \tilde{F} V_0 \tilde{Y}^\top \tilde{F}.
\end{aligned}$$

From (65),

$$E_1 V_0^\top = \beta^3 \tilde{F} \tilde{F}^\top \tilde{F} V_0 V_0^\top - \beta^2 \tilde{F} \tilde{F}^\top \tilde{Y} V_0^\top.$$

Adding these,

$$\begin{aligned}
E_0 V_1^\top + E_1 V_0^\top &= \beta \tilde{Y} \tilde{Y}^\top \tilde{F} \\
&\quad - \beta^2 \big(\tilde{Y} V_0^\top \tilde{F}^\top \tilde{F} + \tilde{F} V_0 \tilde{Y}^\top \tilde{F} + \tilde{F} \tilde{F}^\top \tilde{Y} V_0^\top\big) \\
&\quad + \beta^3 \big(\tilde{F} V_0 V_0^\top \tilde{F}^\top \tilde{F} + \tilde{F} \tilde{F}^\top \tilde{F} V_0 V_0^\top\big).
\end{aligned} \tag{67}$$

*Collecting terms.* It is convenient to separate the $V_0$–independent and $V_0$–dependent contributions. Define

$$\begin{aligned}
R_2(V_0) &:= -\big(\tilde{Y} V_0^\top \tilde{F}^\top \tilde{F} + \tilde{F} V_0 \tilde{Y}^\top \tilde{F} + \tilde{F} \tilde{F}^\top \tilde{Y} V_0^\top\big), \\
R_3(V_0) &:= \tilde{F} V_0 V_0^\top \tilde{F}^\top \tilde{F} + \tilde{F} \tilde{F}^\top \tilde{F} V_0 V_0^\top.
\end{aligned}$$

Then (67) can be written as

$$E_0 V_1^\top + E_1 V_0^\top = \beta \tilde{Y} \tilde{Y}^\top \tilde{F} + \beta^2 R_2(V_0) + \beta^3 R_3(V_0).$$

Thus the small–time expansion of $G_{\tilde{F}}(t)$ is

$$G_{\tilde{F},\beta}(t) = \underbrace{\tilde{Y} V_0^\top - \beta \tilde{F} V_0 V_0^\top}_{G_{\tilde{F},\beta}(0)} + t \left[ \beta \tilde{Y} \tilde{Y}^\top \tilde{F} + \beta^2 R_2(V_0) + \beta^3 R_3(V_0) \right] + O(t^2). \tag{68}$$

**Orders in $\beta$ and interaction with small init.** Under the small–init assumption $\|V_0\|_F = O(\alpha \sqrt{KM})$ and bounded $\tilde{F}, \tilde{Y}$, we have

$$\|R_2(V_0)\|_F = O(\|V_0\|_F) = O(\alpha), \qquad \|R_3(V_0)\|_F = O(\|V_0\|_F^2) = O(\alpha^2).$$

Therefore, for fixed $\beta$,

$$\beta^2 \|R_2(V_0)\|_F = O(\beta^2 \alpha), \qquad \beta^3 \|R_3(V_0)\|_F = O(\beta^3 \alpha^2).$$

In contrast, the $V_0$–independent term $\beta \tilde{Y} \tilde{Y}^\top \tilde{F}$ has norm

$$\|\beta \tilde{Y} \tilde{Y}^\top \tilde{F}\|_F = \beta \|\tilde{Y} \tilde{Y}^\top \tilde{F}\|_F,$$

with a strictly positive factor $\|\tilde{Y} \tilde{Y}^\top \tilde{F}\|_F$ whenever $\tilde{Y} \neq 0$ and $\tilde{F}$ has full column rank.

Hence, in the regime of small random initialization ($\alpha \ll 1$), the leading $O(t)$ contribution to $G_{\tilde{F},\beta}(t)$ is

$$t \, \beta \tilde{Y} \tilde{Y}^\top \tilde{F},$$

while the $V_0$–dependent corrections are suppressed by factors $\beta^2 \alpha$ and $\beta^3 \alpha^2$.

Similarly the final fixed point of $G_{\tilde{F},\beta}(+\infty)$ can be computed by replacing $\tilde{Y}$ with $\tilde{Y}/\beta$ in Eqn. 60:

$$G_{\tilde{F},\beta}^\star = \frac{1}{\beta^2} G_{\tilde{F}}^\star \tag{69}$$

Therefore, if $\beta > 1$ is large, then the final fixed point of $G_{\tilde{F},\beta}(+\infty)$ leads to a smaller backpropagated gradient $G_F$ and may delay grokking.

## I   DISCUSSION

**Relationship between grokking and regularization**. A key insight of the Li$_2$ framework is that a non-zero backpropagated gradient $G_F$ initiates the feature learning process, setting the stage for optimizing the energy function $\mathcal{E}$ that determines which features are learned and how. Intuitively, as long as the lazy learning phase produces a sufficiently large nonzero $G_F$ correlated with the target label $Y$, feature learning will be triggered.

Several ways exist to ensure this nonzero gradient. Regularization mechanisms, such as weight decay, help guarantee the presence of a nonzero $G_F$, and conveniently allow us to demonstrate the feature learning process in a mathematically rigorous way. However, we want to emphasize that regularization is not the only option. For example, $G_F$ might become large during lazy learning and later vanish due to perfect overfitting (see Eqn. 4). Notably, (Prieto et al., 2024) replaces the standard softmax (exponential term) with a stable softmax (linear term), which from the perspective of the Li$_2$ framework, slows down overfitting to the labels. This maintains a nonzero backpropagated gradient, thereby preserving the potential for subsequent feature learning.

In summary, our framework shows that weight decay can be a *sufficient* condition for feature learning (and grokking), but not a necessary one.

**Two different kinds of memorization**. From the analysis, it is clear that memorization of grokking stems from overfitting on random features, which is different from landing on memorization solutions following the feature learning dynamics due to limited/noisy data (Thm. 5). From this view,

grokking does not switch from memorization to generalization, but switch from overfitting to generalization.

**Flat versus sharp optima**. Common knowledge often regards flat optima as generalizable solutions, while sharp optima as memorization/overfitting. From the point of view of Li$_2$, sharp optima happen when the model overfits on random features (Sec. 3), and thus small changes of the weights will change the loss a lot. On the other hand, we can prove that the local optima from $\mathcal{E}$ are flat (Corollary 1), and thus small changes of the weights at certain directions do not change $\mathcal{E}$. If the model is over-parameterized, then multiple nodes may learn the same (or similar) set of features, which create freedom for the loss function to be flat. If we learn memorizing features due to limited/noisy data (Thm. 5), then more nodes need to be involved to "explain" the target well, and the overall weights will appear to be less flat.

**Small versus large learning rates**. According to our analysis, in Stage I, we need large learning rate to quickly learn ridge solution $V$ so that the backpropagated gradient $G_F$ becomes meaningful to trigger Stage II. In Stage II, the best learning rate depends on the amount of data available. With a lot of data, we can afford larger learning rate to find the features quickly. With limited data, we may need smaller learning rate to stay in the basins of generalizable features (Fig. 7), which may be contradictory to common belief.

## J    MORE EXPERIMENTS

### J.1    MORE EXPERIMENTS ON PHASE TRANSITION

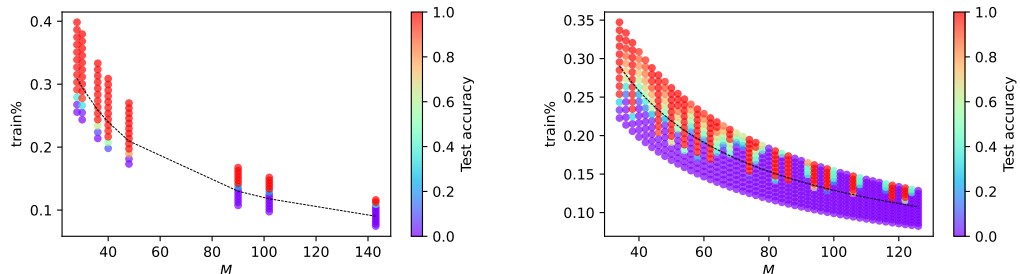

Figure 8: Generalization/memorization phase transition in modular addition tasks (cont.). **Left:** Product group $\mathbb{Z}_4 \otimes \mathbb{Z}_7$, $\mathbb{Z}_5 \otimes \mathbb{Z}_6$, $\mathbb{Z}_2 \otimes \mathbb{Z}_2 \otimes \mathbb{Z}_9$, $\mathbb{Z}_{13} \otimes \mathbb{Z}_{11}$, $\mathbb{Z}_5 \otimes \mathbb{Z}_2 \otimes \mathbb{Z}_2 \otimes \mathbb{Z}_2$, $\mathbb{Z}_6 \otimes \mathbb{Z}_4 \otimes \mathbb{Z}2$, $\mathbb{Z}_3 \otimes \mathbb{Z}_2 \otimes \mathbb{Z}_{17}$, $\mathbb{Z}_2 \otimes \mathbb{Z}_3 \otimes \mathbb{Z}_3 \otimes \mathbb{Z}_5$. **Right:** Non-Abelian groups with $\max_k d_k = 2$ (maximal irreducible dimension 2). These non-Abelian groups are generated from GAP programs (See Appendix Sec. J.2).

### J.2    USE GROUPS ALGORITHMS PROGRAMMING (GAP) TO GET NON-ABELIAN GROUPS

GAP (https://www.gap-system.org/) is a programming language with a library of thousands of functions to create and manipulate group. Using GAP, one can easily enumerate all non-abelian group of size $M \leq 127$ and create their multiplication tables, which is what we have done here. From these non-Abelian groups, for each group size $M$, we pick one for our scaling law experiments (Fig. 5 bottom right) with $\max_k d_k = 2$.

### J.3    ZERO-INIT ACCELERATES THE FEATURE LEARNING PROCESS

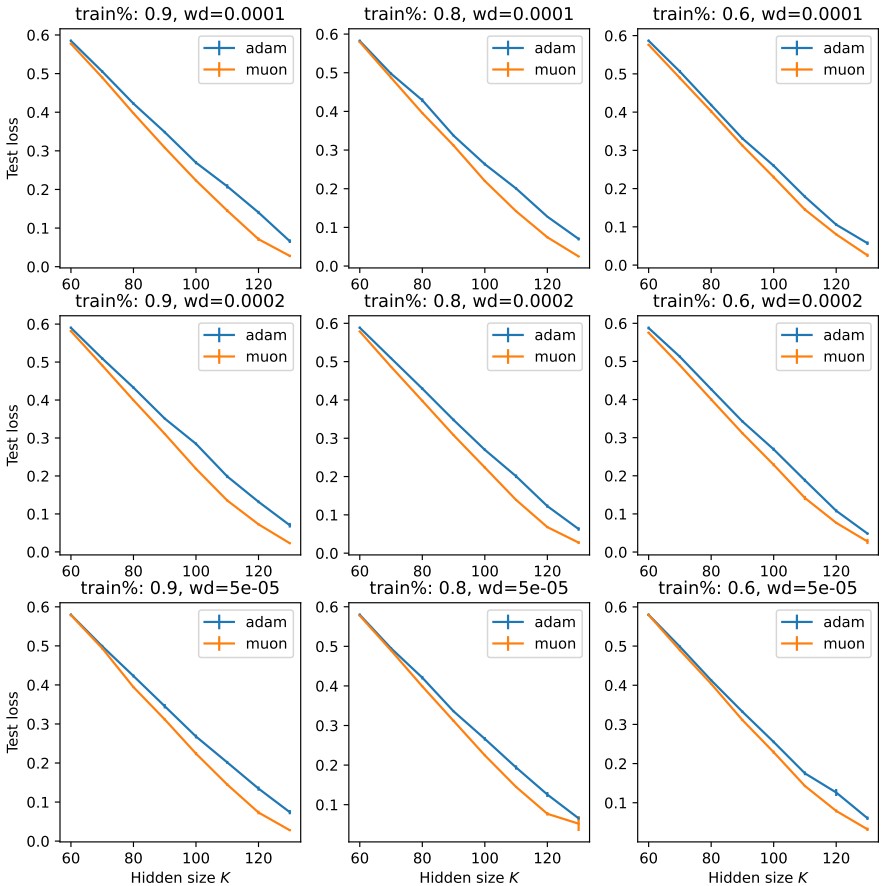

Figure 9: Adam versus Muon optimizers in modular addition tasks with $M = 71$, when the number of hidden nodes $K$ is relatively small compared to $M$. Muon optimizer achieves lower test loss compared to Adam.

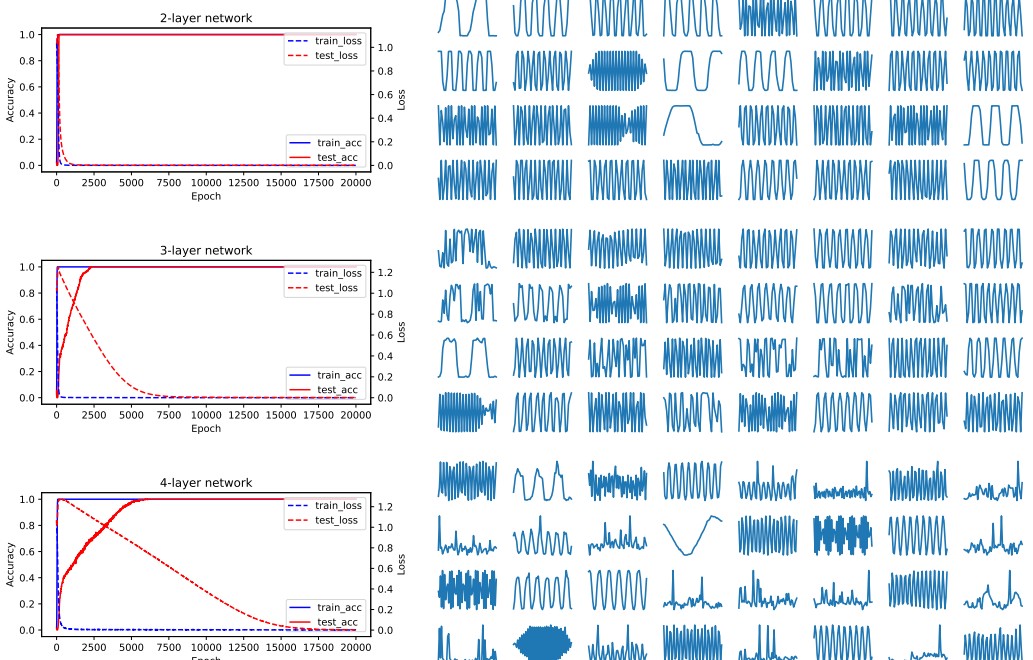

Figure 10: Training modular addition tasks with 2, 3 and 4 layer network with ReLU activations. **Left:** Training accuracy and losses. **Right:** Learned features at the lowest layer. With more layers, the training takes longer and grokking (delayed generalization) becomes more prominent. However, features at the lowest layer remain (distorted version) of Fourier bases, which are consistent with the analysis in Sec. 6.

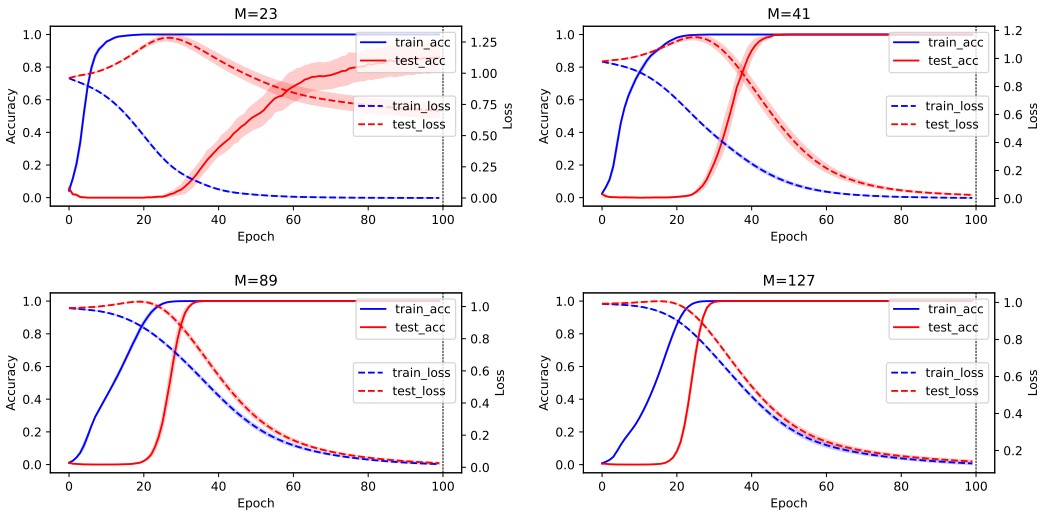

Figure 11: Training modular addition tasks with real weights ($M = 23, 41, 89, 127$). Learning rate is $0.005$, weight decay is $5e - 5$. Number of hidden nodes $K = 256$. Test sample is $20\%$ of the full set of $M^2$. Using Adam optimizer. Averaged over 5 seeds. This is a baseline.

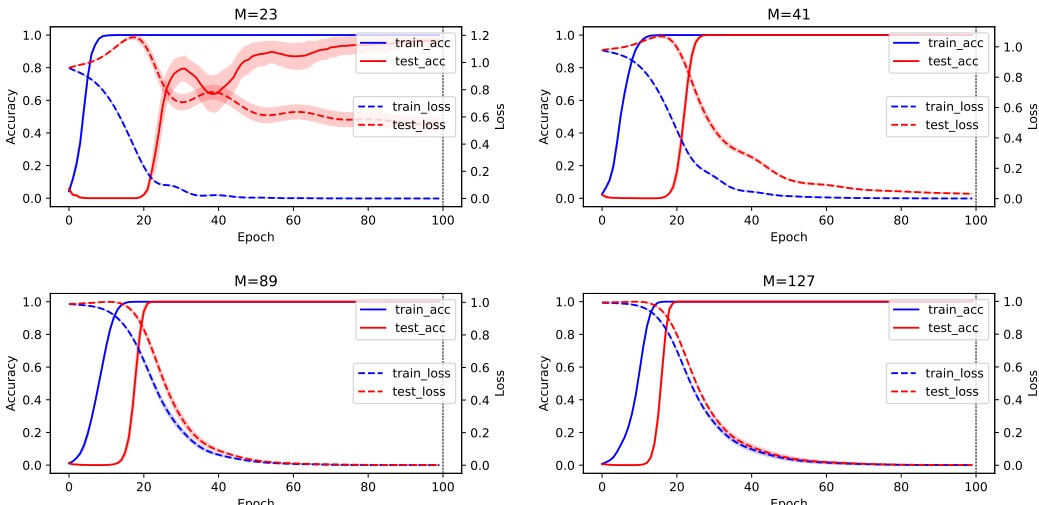

Figure 12: Training modular addition tasks with complex weights ($M = 23, 41, 89, 127$). Learning rate is 0.005, weight decay is $5e - 5$. Number of hidden nodes $K = 256$. Test sample is $20\%$ of the full set of $M^2$. Using Adam optimizer. Averaged over 5 seeds. Compared with the real case (Fig. 11), models with complex weights seem to grok faster.

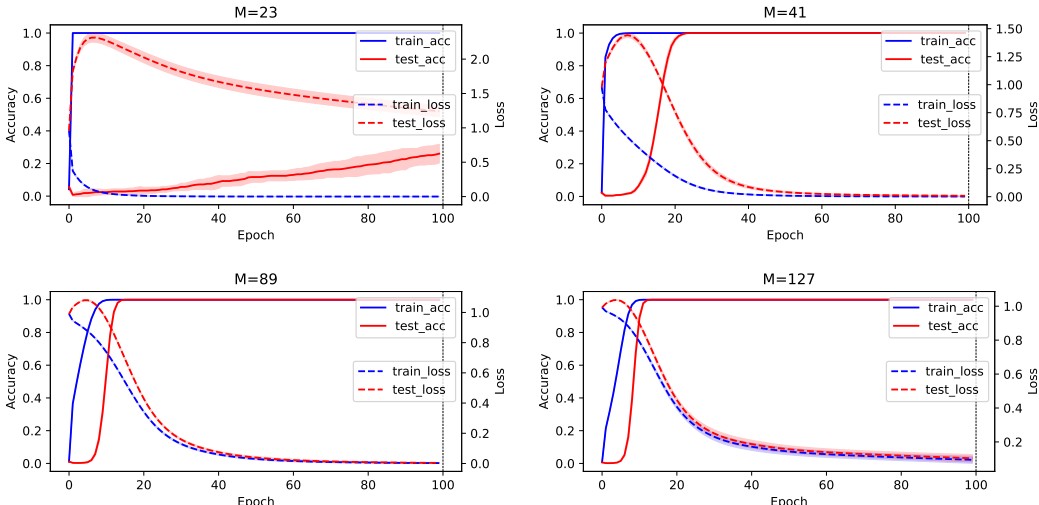

Figure 13: Training modular addition tasks with real weights ($M = 23, 41, 89, 127$). Instead of using gradient descent to update the top layer $V$, in every gradient update we use ridge regression solution $V_{\text{ridge}}$ with respect to the current $F$ (Proposition 1). Learning rate is 0.005, weight decay is $5e - 5$. Number of hidden nodes $K = 256$. Test sample is $20\%$ of the full set of $M^2$. Using Adam optimizer. Averaged over 5 seeds. The grokking still happens (for $M = 23$ check Fig. 14 for completeness). It is slower for $M = 23$ but actually faster for $M = 41, 89, 127$, compared to the baseline (Fig. 11).

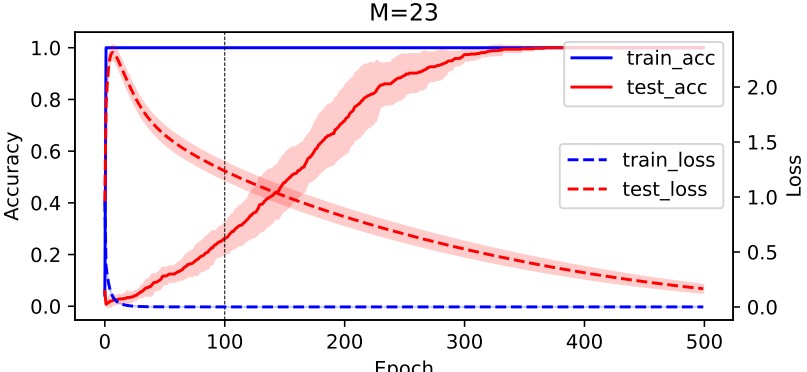

Figure 14: Training modular addition tasks with real weights $M = 23$ for 500 epochs, using $V_{\mathrm{ridge}}$ as the top layer weight. The grokking still happens but slower than the baseline (Fig. 11) for $M = 23$.

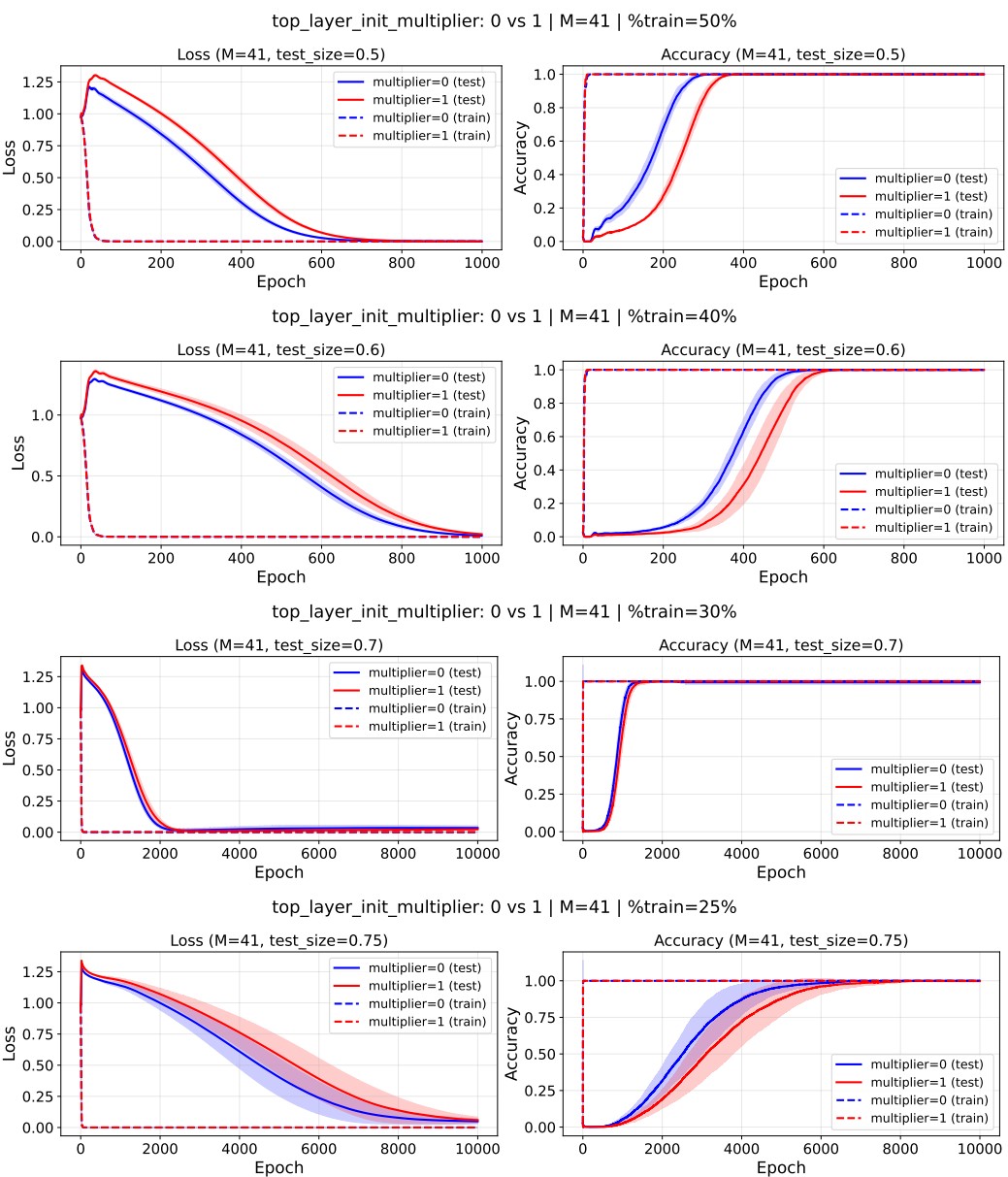

Figure 15: Zero-init accelerates the feature learning process for $M = 41$. Check Fig. 17 for the hyperparameters setting. Blue lines are zero-init for $V$ (i.e., $V(0) = 0$), red lines are regular initialization.

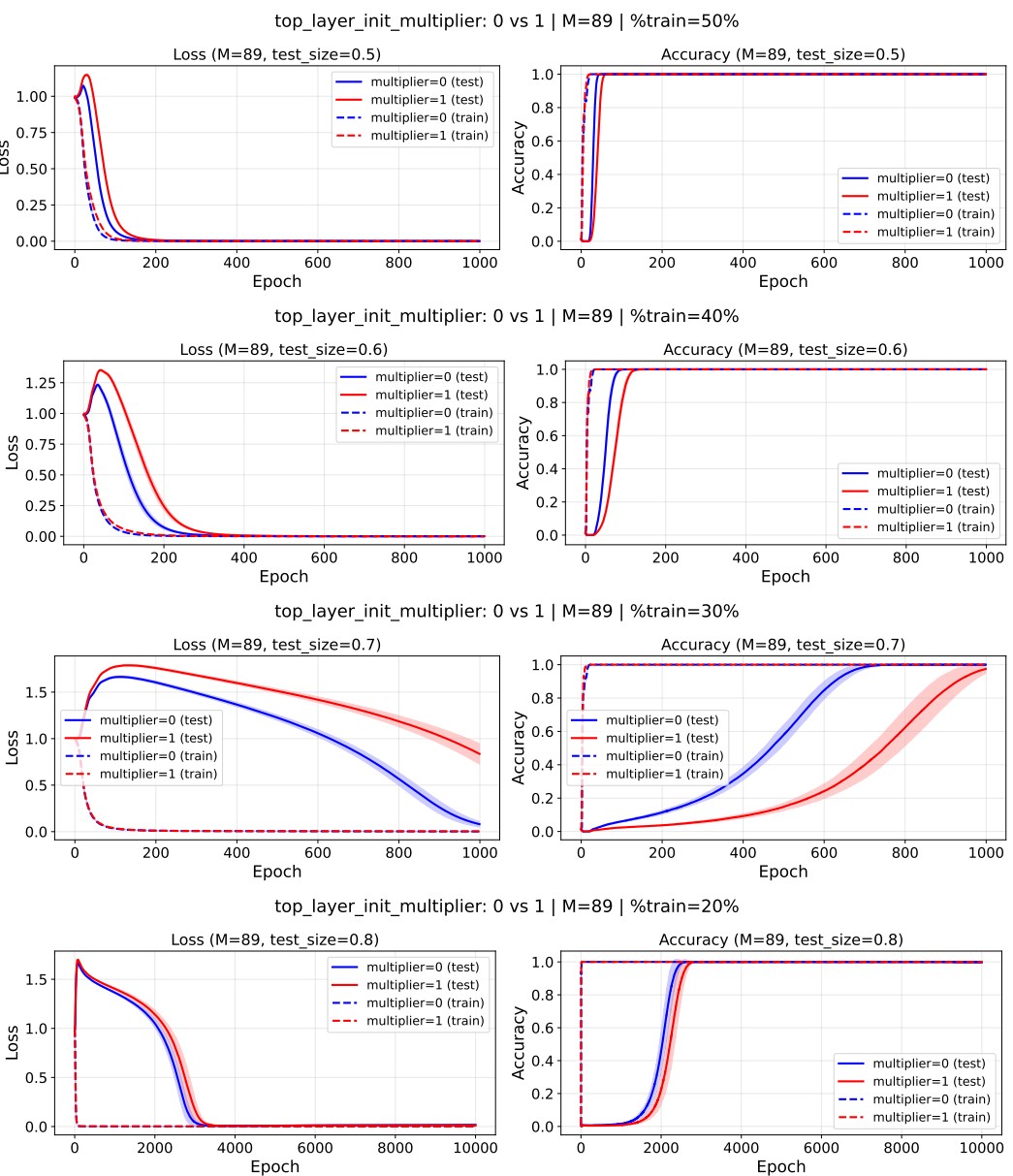

Figure 16: Zero-init accelerates the feature learning process for $M = 89$. Check Fig. 17 for the hyperparameters setting. Blue lines are zero-init for $V$ (i.e., $V(0) = 0$), red lines are regular initialization.

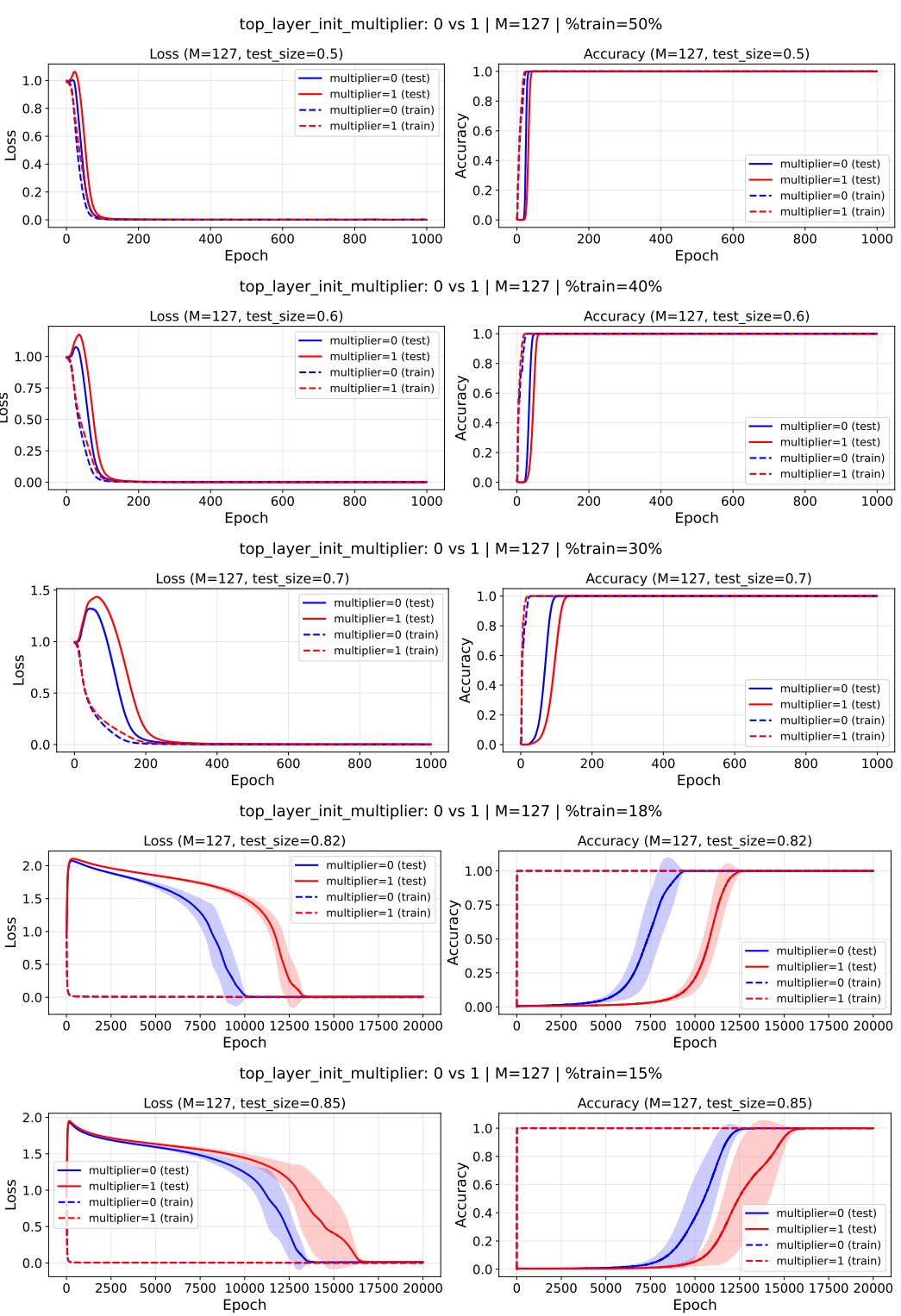

Figure 17: Zero-init accelerates the feature learning process for $M = 127$. Learning rate is $0.005$. Using Adam optimizer. $\eta = 5e - 5$. $K = 1024$. $\sigma(x) = x^2$ and MSE loss. Top_layer_init_multiplier $= 0$ means $V(0) = 0$, i.e., the top layer is initialized to be 0 (blue lines). If Top_layer_init_multiplier $= 1$, then $V$ is initialized regularly (red lines). It is clear that zero-initialization leads to faster grokking (and feature learning) than regular initialization, in particular when data are scarce.

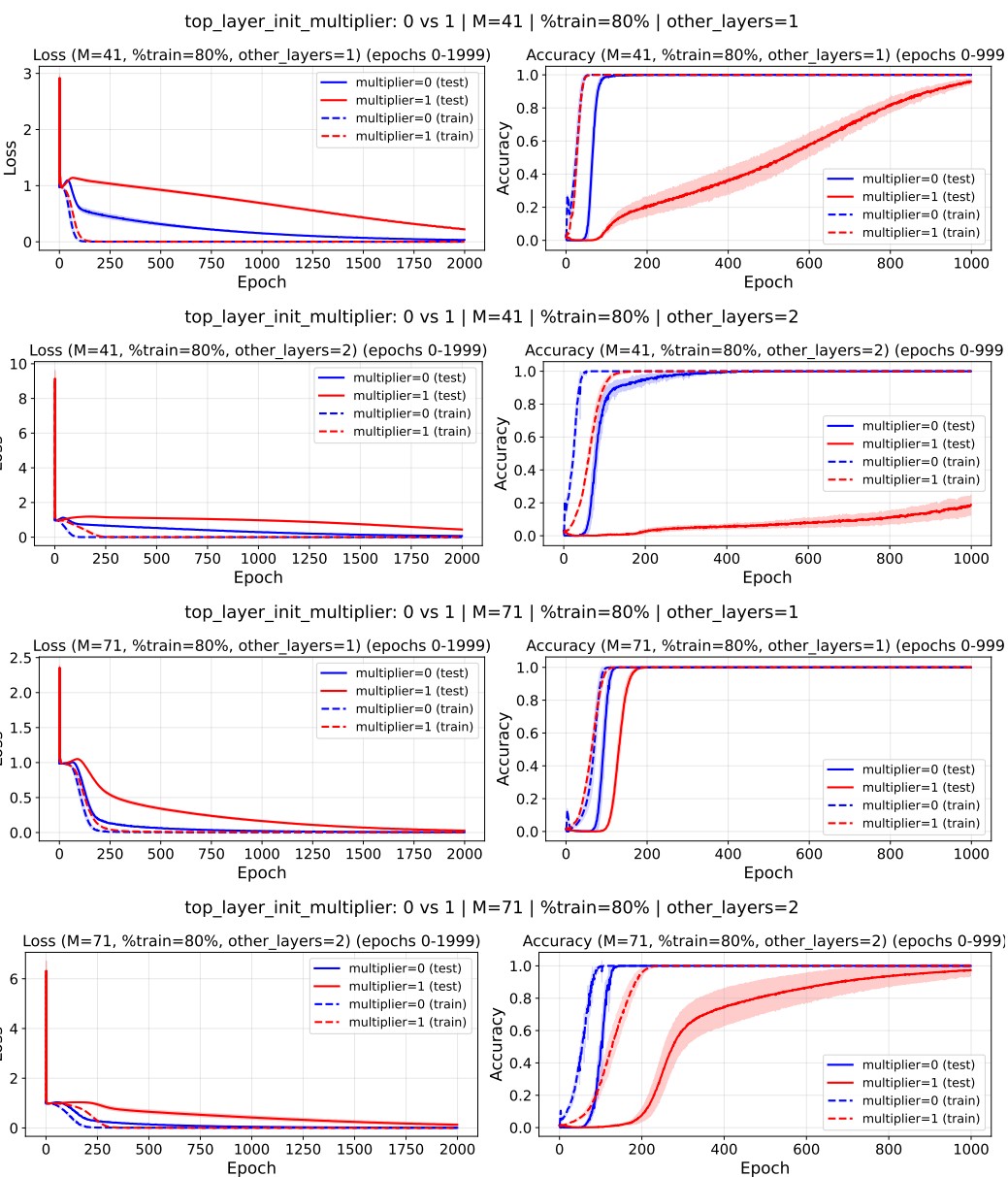

Figure 18: Zero-init accelerates the feature learning process for multi-layer setting. `other_layers = 1` means adding an additional hidden layer in between top layer $V$ and bottom layer $W$ with residual connection, etc. Learning rate is $0.005$. Using Adam optimizer. $\eta = 1e - 4$. $K = 2048$. $\sigma(x) = \text{ReLU}(x)$ and MSE loss. `Top_layer_init_multiplier = 0` means $V(0) = 0$, i.e., the top layer is initialized to be 0 (blue lines). If `Top_layer_init_multiplier = 1`, then $V$ is initialized regularly (red lines). It is clear that zero-initialization leads to much faster feature learning than regular initialization in multi-layer settings (often $10\times$).

