# OpenReview forum: "$\mathbf{Li_2}$: A Framework on Dynamics of Feature Emergence and Delayed Generalization"
_ICLR.cc/2026/Conference — ICLR 2026 Poster_

### Official Review · Reviewer_aGxx · 2025-10-23

**Soundness:** 4
**Presentation:** 3
**Contribution:** 4
**Rating:** 8
**Confidence:** 4

**Summary:**

The paper proposes a three-stage framework to theoretically study grokking using two-layer neural network with MSE loss and quadratic activation functions. It find connections between stage II and the optimization of an energy function. It rigorously analyzes the role of learning rate, weight decay, sample size, and proposes a scaling law of generalization/memorization boundary. It also theoretically analyzes the benefits of Muon optimizer in the feature learning regime.

**Strengths:**

1. The first work to rigorously analyze the role of learning rate, weight decay, and width on grokking dynamics.
2. Interesting and novel analysis of the interactive feature learning regime.
3. First theoretical analysis of Muon in the feature learning regime.
4. The theoretical results of two types of memorization is intriguing.

**Weaknesses:**

1. MSE loss is uncommon in the training of deep learning models, though it's fine for the ease of theoretical analysis.
2. The theory needs a nonzero weight decay to provably show the three phases, while in practice, weight decay is unnecessary for grokking to occur.
3. Several assumptions need to be justified.

**Questions:**

1. In stage I, why the activations F are mostly unchanged? Any empirical observations to support such assumption? I guess it needs assumptions on the learning rate, weight decay, and initialization scale of V.
2. In Lemma 1, why can we assume W always follow normal distribution at each step? Besides that, W is assumed to follow normal, but in the proof (line 714-716), you assume w_i follow N(0, I), which is stronger.
3. In Lemma 1, why can we assume $<x_i, x_{i'}> = \rho$ ? Does this assumption hold in any synthetic tasks that show grokking?
4. In Theorem 5, what are focused and spreading memorization? What's the difference between memorization and overfitting?

---

> ### Author Response · Authors · 2025-11-21
> **Rebuttal**
>
> We thank the reviewer for the encouraging comments! Here we try to address your concern and questions below:
>
> # Weakness
>
> > MSE loss is uncommon in the training of deep learning models, though it's fine for the ease of theoretical analysis.
>
> In this work, we use MSE (with zero-mean projection on the batch) as a theoretically-convenient setup. This also follows many existing grokking work (e.g., [2]). We will leave more popular loss functions such as softmax as the future work. Intuitively, if $\hat Y$ is small, compared to the target label $Y$, then with first-order approximation, the backpropagated gradient of the loss function is similar to that of MSE loss (see e.g., Lemma B.2 in https://arxiv.org/pdf/2403.03507, Eqn. 26) and similar analysis can be applied.
>
> > The theory needs a nonzero weight decay to provably show the three phases, while in practice, weight decay is unnecessary for grokking to occur.
>
> Please check the main rebuttal for the answer.
>
> One key point of the framework is that the non-zero backpropagated gradient initializes the feature learning process. This drives the optimization process of a nice energy function that governs what features will be learned and how they are learned.
>
> From the framework, intuitively, as long as the lazy learning stage generates a sufficient amount of nonzero that is related to the target label, the feature learning will happen. Adding some sorts of regularization (e.g., weight decay) makes sure that there is always a non-zero, and it is convenient to use weight decay to demonstrate the feature learning process in a mathematically rigorous way. This does not rule out other possibilities, e.g., becomes large in lazy learning and then vanishes later due to perfect overfitting (in Eqn. 5).
>
> In short, our framework shows that using weight decay is a sufficient condition to trigger feature learning (and grokking), but not a necessary condition. Following the intuition, a necessary condition is likely that the top layer does not achieve perfect overfit too quickly.
>
> We thank the reviewers for providing many references. Here we will leverage our framework to explain why they achieve grokking.
>
> + [1] uses stable softmax rather than regular softmax (exp->linear). This prevents the model from overfitting to the label too quickly, and thus maintains a nonzero backpropagated gradient that can be useful for feature learning.
>
> + [2] reports that grokking happens with full-batch GD and Adam without regularization. We have reproduced this behavior. For GD, our hypothesis is that it takes longer for GD to overfit, which gives sufficient non-zero. For Adam without regularization, we observed that grokking often happens with a large learning rate, which leads to huge loss at the beginning (“loss overshooting”) and a “shockwave” of non-zero that triggers feature learning. If there is no such “shockwave”, grokking does not happen.
>
> + Scaling up prediction output [3] or using large initialization may create similar loss overshooting, which leads to grokking behaviors later [4] (Fig. 7).
>
> In summary, we need non-zero backpropagated gradient to start the feature learning process.
>
> **Reference**
>
> [1] L. Prieto et al, Grokking at the edge of numerical stability.
>
> [2] A. Gromov, Grokking modular arithmetic
>
> [3] L. Chizat et al, On Lazy Training in Differentiable Programming
>
> [4] T. Kumar et al, Grokking as the Transition from Lazy to Rich Training Dynamics

---

> ### Author Response · Authors · 2025-11-21
> **Rebuttal #2**
>
> # Questions
>
> > Why can we assume $W$ follow normal distribution and why $F$ is mostly unchanged?
>
> According to Fig. 2, at the beginning of the training, the changes of weights in the lower layer $W$ are much smaller than that of top layer $V$ (Fig. 2 first row, last column) and thus we can assume that $W$ are still independently distributed following the initialization. Therefore, $F$ is also mostly unchanged. Previous works [e.g., (Montanari and Urbani 2025) raised by reviewer NQrJ]  also show that in the lazy stage, the weights in the lower layer do not change much.
>
> In our updated version, we show that if we replaced the top layer with a simple ridge regression on top of (changing) $F$, then grokking still happens and the dynamics remains similar. This setting is closer to Lemma 1. Please check Fig. 12.
>
> > The assumption $\mathbf{x}_i^\top \mathbf{x}_j = \rho$ in Lemma 1
>
> Note that this assumption is just for convenience in order to prove that $F F^\top = K [(a-b)I + b\mathbf{1}\mathbf{1}^\top]$, i.e., it is a weighted sum of identity and an all one matrix. If $\mathbf{x}_i^\top \mathbf{x}_j$ are not all equal but close (i.e., $\|\mathbf{x}_i^\top \mathbf{x}_j - \rho\| \le \epsilon$), then we can show that $FF^\top$ is not too far away from the form of “multiple of identity + all one matrix”, and the backpropagated gradient $G_F$ remains a similar structure. We have updated the revision to fix it.
>
> > Assumption of normal distribution versus $N(0, I)$ in the proof of Lemma 1.
>
> Thanks for pointing this out. We have updated the proof to fix it.
>
> > What's the difference between memorization and overfitting?
>
> According to our framework, overfitting is transient: first the top layer learns a model from hidden random features and cannot generalize to unseen cases, but if we continue the training process, then the hidden features transit from random to generalizable and the model starts to predict with a better representation. On the other hand, memorization happens when there is a lack of data and the model is unable to learn generalizable features (e.g., the landscape of energy function simply does not have such local maxima), no matter how long the training lasts.
>
> > In Theorem 5, what are focused and spreading memorization?
>
> Given insufficient data (e.g., the model is trained only on the data from a single class), with the same energy function $\mathcal{E}$, different nonlinearity will lead to different local maxima structure. Some nonlinearity (e.g., $\sigma(x) =x^2$) gives an optimal solution which is a sparse vector that only remembers a single pair $(h_1, h_2)$ (i.e., focused memorization), while other nonlinearity (popular ones like SiLU, sigmoid, tanh, etc) gives an optimal solution that remembers all pairs from the data (spreading memorization).

---

> > ### Comment · Reviewer_aGxx · 2025-11-22
> >
> > Thanks for the clear explanations. I'm happy to keep my score.

---

> > > ### Author Response · Authors · 2025-11-22
> > > **Acknowledgement**
> > >
> > > Great thanks for your support!

---

### Official Review · Reviewer_NQrJ · 2025-10-27

**Soundness:** 2
**Presentation:** 3
**Contribution:** 3
**Rating:** 4
**Confidence:** 2

**Summary:**

This work proposes a 3-stage framework to understand the emergence of feature learning in large-width two-layer neural networks, named **Li$_{2}$**. This consists of:

1. Lazy regime: first-layer weights are effectively random. The second layer fits the data using random features. The back-prop term to the hidden layer, $GF$, carries little usable structure and, without weight decay, can vanish at the lazy fixed point (ridge solution).
2. Independent: With weight decay \(\eta>0\), \(GF\) acquires target structure. Under some assumptions and large width \(K\), it simplifies to
$$
GF = \frac{\eta}{(Kc_1+\eta)(nc_2+\eta)}\tilde Y\tilde Y^{\top}F,
$$
so each neuron follows gradient **ascent** on the single-neuron energy
$$
E(w)=\frac{1}{2}\big\|\tilde Y^{\top}\sigma(Xw)\big\|_2^2,
$$
i.e., neurons learn useful directions independently.
3. Interactive: As several features are learned, neuron–neuron interactions reshape $GF$: similar features repel, and the signal is steered toward missing directions.

This is discussed explicitly in a modular arithmetic task and quadratic activation $\sigma(x)=x^{2}$. Using group-theoretical tools, it is shwon that local maxima of $E$ align with group irreducible representations which for this task coincide with Fourier modes, yielding closed-form descriptions of learned features and their attained energies $E^\star$.

Finally, an extension to the multi-layer case is also discussed.

**Strengths:**

The paper is clearly written and provides a fairly generic picture of the mechanism for feature learning in two-layer neural networks, which is task independent and rely only on an investigation of the gradients.

**Weaknesses:**

The results for each section rely on different assumptions, which makes **Li$_{2}$** look more like different patches of results rather than a unified picture. Some of the observations appearing in this work have also appeared in other works in the feature learning literature, and more throughout comparison is lacking. I expand on these two points below.

**Questions:**

1. Different results in this work appear to rely on different assumptions about $n,d,K,M$, and it is not immediately clear whether these are mutually consistent. For instance, Lemma 1 assumes that $K$ is sufficiently large and that $x_i^\top x_j=\rho$, which is only possible when $n<d$ unless the $x_i$ are degenerate. By contrast, Theorem 2 and Corollary 1 take $d=2M$ and $n=M^2$, which is consistent with Lemma 1 only for $M<2$. Similarly, Theorem 4 requires $n\gtrsim d_k^{,2} M \log(\delta/\delta)$ via Matrix Bernstein. Overall, I found it confusing to determine whether the regimes considered for the different phases of Li$_2$ hold simultaneously.

2. Feature learning for two-layer neural networks has been studied extensively in recent years, with several works arriving at a picture closely related to **$Li$_2$**. For example:

- One line of work systematically analyzes one-pass SGD dynamics in teacher–student settings (a 2LNN learning another, not necessarily identical, 2LNN). These studies show that the dynamics decompose into plateau / saddle-to-saddle phases: first-layer weights move within a neighborhood of zero (the *mediocrity phase*), then individual neurons correlate with target directions independently before finally coalescing (the *specialization phase*); see Saad and Solla (1995), Goldt et al. (2019), and Arnaboldi et al. (2023). While these works differ technically (finite-width networks, one-pass SGD), the overall mechanism is closely related, with a key difference being the absence of an overfitting phase under one-pass SGD.

- Another closely related line of work considers feature learning during the first few steps of GD with aggressive learning rates (Damian, 2022; Ba et al., 2022; Dandi et al., 2024). The energy in Eq. (8), often called the ``Correlation loss,’’ is a common approximation in this literature because it yields exact weak-recovery thresholds for the initial gradient steps. A notable observation is that, after a single aggressive step, the gradient can be asymptotically characterized—depending on sample complexity—by a rank-one matrix that correlates with the target, enabling the network to express nonlinear components with limited data. Is this the same mechanism as in Theorem 3?

- More recently, Montanari and Urbani (2025) studied a related teacher–student setting for full-batch GD on a wide 2LNN learning a single neuron, and identified three timescales: a lazy timescale, a generalization timescale, and an overfitting timescale (motivating early stopping). (There is no specialization here because the target has a single neuron.) In Li$_2$, overfitting is associated with the lazy phase. How can this be reconciled with Montanari and Urbani (2025)? How should we understand the benefits of early stopping within the Li$_2$ framework?

**References**

- (Saad and Solla 1995) Dynamics of On-Line Gradient Descent Learning for Multilayer Neural Networks. NeurIPS 1995.
- (Goldt et al. 2019) Dynamics of stochastic gradient descent for two-layer neural networks in the teacher-student setup. NeurIPS 2019
- (Arnaboldi et al. 2023) From high-dimensional & mean-field dynamics to dimensionless ODEs: A unifying approach to SGD in two-layers networks. COLT 2023
- (Damian 2022) Neural Networks can Learn Representations with Gradient Descent. COLT 2022.
- (Ba et al. 2022) High-dimensional Asymptotics of Feature Learning: How One Gradient Step Improves the Representation. NeurIPS 2022.
- (Dandi et al. 2024) How Two-Layer Neural Networks Learn, One (Giant) Step at a Time. JMLR 2024
- (Montanari and Urbani 2025) Dynamical Decoupling of Generalization and Overfitting in Large Two-Layer Networks. arXiv 2025

---

> ### Author Response · Authors · 2025-11-21
> **Rebuttal**
>
> We thanks the reviewer for the comments. Here we try to address your concerns.
>
> > Different results in this work appear to rely on different assumptions ...
>
> Your concern about the inconsistency of the hyperparameter ranges is totally valid and we really appreciated your insights! We have relaxed the assumption of $\mathbf{x}_i^\top \mathbf{x}_j = \rho$ into $\|\mathbf{x}_i^\top \mathbf{x}_j - \rho\| \le \epsilon$ in the revision. After that there is no inconsistency and $n$ can be larger than $d$. Overall we operate on the region of a fairly large number of hidden nodes (but not infinite), as shown in Fig. 1 (right panel).
>
> > Feature learning for two-layer neural networks has been studied extensively in recent years
>
> We acknowledge that feature learning has been studied quite thoroughly in the previous literatures. We clarify our contribution below.
> + Saad and Solla (1995), Goldt et al. (2019), and Arnaboldi et al. (2023) assumes isotropic Gaussian input, and use teacher-student setting to measure whether the model generalizes. Therefore, they only study how the teacher-defined features are learned by the student. Their frameworks also do not model the grokking behaviors (no overfitting phase). In contrast, our proposed framework focused more on feature emergence: i.e., there is no teacher network predefining the feature but the model itself figures out the structure hidden in the data. We also study more realistic inputs (the energy function $\mathcal{E}$ applies to any input and the analysis on local maxima assumes any group arithmetic tasks). Furthermore, while the previous works focus on statistical properties of the hidden nodes (modeled by macroscopic / mean-field metrics), we show that it is possible to characterize the dynamics of individual neurons and model their interactions.
>
> + Theorem 3 in $\mathbf{Li}_2$ framework actually explores a very different mechanism than previous works. Since there are many local maxima in the energy function $\mathcal{E}$, random initialization + over-parameterization will lead to learned weights that cover them and these local maxima collectively form a powerful representation to predict the target. To our knowledge, we haven’t seen any previous works that explore such a mechanism. (Damian, 2022; Ba et al., 2022; Dandi et al., 2024) used one-big gradient step for feature learning, which typically assumes that weights do not deviate too much from independent initialization $W_0$ , and focuses on hidden subspace learning. The step is also teacher-student with Gaussian inputs. In contrast, we show that the feature learning can learn highly nonlinear features, which are local maxima of energy function $\mathcal{E}$ under fairly general setup. The learned features have already reached local maxima after (many steps of) optimization procedure, rather than a single step.
>
> + When the number of training samples are in the “godilock” region (i.e., just enough), the landscape of the energy function $\mathcal{E}$ will contain both generalization and memorization local maxima, and memorization local maxima may have better energy value. With small learning rate and early stopping, the learning process will stop at the generalization local maxima, yielding good performance; if we train much longer and/or with a large learning rate, then the optimization procedure will eventually escape from generalization local maxima (whose energy is low) to memorization ones (whose energy is high) (Fig. 6), killing good representations and demonstrating "feature unlearning". Fig. 5 also demonstrates the same effects: when the size of the dataset barely meets the bar of grokking / generalization, small learning rate gives better performance than large learning rate, since small learning rate will keep the weights staying in the generalization basins. This is our explanation, which is different from Montanari and Urbani (2025) and hopefully more intuitive.

---

> ### Comment · Reviewer_NQrJ · 2025-11-25
>
> I thank you for your rebutal and for the additional analysis that relaxes the $x_{i}^{\top}x_{j}=\rho$ assumption.
>
> Regarding the relationship to the literature, I think the authors misunderstood my commnent. I do understand that the assumptions in the related papers is different from the one studied here. Nevertheless, there is substantial overlap in phenomenology, and I was expecting the authors to *put their results in perspective* with the literature, **not** to highlight the difference in assumptions.
>
> In particular, let me try to clarify some points.
>
> > *In contrast, our proposed framework focused more on feature emergence: i.e., there is no teacher network predefining the feature but the model itself figures out the structure hidden in the data.*
>
> In order to quantify the benefits of feature learning to generalization, one must assume structure in the target distribution. Teacher-student models are one way of doing that. However, note that assuming a target distribution **does not imply** that the method studied has knowledge about the underlying structure. In particular, the setting studied in these one-pass SGD papers is agnotistic to the existence of the teacher: the model also *figures out the structure hidden in the data*.
>
> I agree that grokking is not a meaningful concept there because the dynamics in these papers are one-pass (as I stressed, no overfitting, no training error). Nevertheless, there are similarities in the two mechanisms described.
>
> > *(Damian, 2022; Ba et al., 2022; Dandi et al., 2024) used one-big gradient step for feature learning, which typically assumes that weights do not deviate too much from independent initialization $W_{0}$, and focuses on hidden subspace learning.*
>
> Two points on this comment:
> - First, these works show that in this context, feature learning is manifested by the appearance of spikes correlating with the target distribution. These spikes allow the network to learn high-order frequencies in the directions learned, what is not possible at initialization. Therefore, I would say the weights and error deviate *a lot* from initialization.
> - Second, they do not assume this behavior. They **prove** it.

---

> ### Author Response · Authors · 2025-11-27
> **Answers to your questions.**
>
> Thanks for your clarification. Glad to hear that the issue around $\mathbf{x}_i^\top \mathbf{x}_j = \rho$ is now addressed.
>
> >  I was expecting the authors to put their results in perspective with the literature, not to highlight the difference in assumptions.
>
> We now have updated the draft to demonstrate that the backpropagated $G_F$ is dominated by the term $\tilde Y\tilde Y^\top F$ also at initial phase of gradient descent, in addition to the converging phase. This is similar to the idea of "one-big gradient step". The critical term $\tilde Y\tilde Y^\top F$ essentially projects the activation $F$ onto the label space, and reinforce it by backpropagated gradient. This is similar to the intuition that "align between the first-layer weights and the linear component of the teacher model f∗" (Ba et al. 2022). The paper (Dandi et al., 2024) shows that with multiple steps, the model can learn multiple target directions under certain conditions. Our framework has a similar component in Theorem 7 (Top-down modulation), which shows that the backpropagated gradient from residual target $\hat Y - \tilde Y$ will automatically focus the model on missing features.
>
> What makes us different is that the special structure of $\tilde Y\tilde Y^\top F$ paves the way of discovering the energy function $\mathcal{E}$ that governs the feature learning process, and enables us to analyze the dynamics of each neurons independently. We also focus on group arithmetic tasks rather than Gaussian inputs, etc, as mentioned in our original response.
>
> We hope this puts our work in perspective with the literature. Let us know if you have more questions / comments.
>
> > Two points on this comment:
>
> Sorry we may have miscommunicated our thoughts. What we mean here, is that initial weight $W_0$ is assumed to follow random iid distribution (or something similar) to get the first gradient step $G_0$. We agree that the actual magnitude of $G_0$ can be large. But once that gradient step is taken, similar analysis may not hold anymore for the updated weight $W_1 = W_0 + G_0$, which deviates the random iid distribution. The one-pass assumption is also based on this restriction.
>
> In our case, since we have found an energy function to govern the feature learning dynamics globally, we can reuse the same data/population distribution, and analyze sample complexity etc. That's no such restriction.
>
> We hope this clarifies any confusions. Thanks.

---

### Official Review · Reviewer_sYi6 · 2025-10-28

**Soundness:** 3
**Presentation:** 3
**Contribution:** 2
**Rating:** 4
**Confidence:** 4

**Summary:**

The paper analyzes grokking dynamics in the presence of regularization. They identify three phases in the learning process - lazy learning regime and independent and interactive feature learning regime, and prove various theorems which govern different aspects of the training dynamics.

**Strengths:**

1) The theorems proved are mathematically rigorous, with detailed proofs in the appendices.
2) The scaling law analysis in Section 5.4 will be useful since it gives a first-principles derivation of the scaling phenomenon.
3) Extensions to modern optimizers and deeper networks were adequately discussed.

**Weaknesses:**

1) My main concern is that the key observation (that there is a lazy and rich learning regime) was already reported in [1], who study this in the context of polynomial regression. While the current paper offers mathematically rigorous proofs, the distinction with [1] needs to be more elaborately discussed in the main text.
2) The mathematical framework presented displays the three stages, but fails to offer more insight on what drives these transitions or when do these transitions. For example, based on lines 471-475, it seems like it's the (inverse) learning rate which sets the scale for when these transitions occur, but it would be great if this could be explained in more detail.
3) How does one relate the top-down modulation in Sec 6 to the continuous feature learning observed in [2]?
4) Discussion on limitations not found in main text.


[1] Tanishq Kumar, Blake Bordelon, Samuel J. Gershman, and Cengiz Pehlevan. Grokking as the transition from lazy to rich training dynamics, 2024.
[2] Gromov, A. (2023). Grokking modular arithmetic. arXiv preprint arXiv:2301.02679.

**Questions:**

1) Line 105-106 : I suggest modifying "grokking mostly happens ... regularization" to "grokking is accelerated ... regularization"
2) Why are the axes cut off till epoch 300 in Fig 2 last column?
3) Since the discussion on Thm 3 involves optimization in the complex domain, were any follow-up experiments done with complex weights?
4) The existence of maxima for the ascent functions is interesting (Thms 4-5). What can be said about the network's ability to find these maxima?

---

> ### Author Response · Authors · 2025-11-21
> **Rebuttal**
>
> We thank the reviewer for the comments. We address the weakness and concerns below.
>
> # Weakness
> > My main concern is that the key observation (that there is a lazy and rich learning regime) was already reported in [1]
>
> As mentioned by sYi6, our main contribution is to build a mathematical framework to quantitatively explain what kind of features emerged and how they emerged in feature learning tasks. Since the seminar Neural Tangent Kernel (NTK) works ([A1], [A2]), people have long realized the existence of the lazy (e.g., NTK) and rich learning process, and have explored various ways to move beyond NTK regime and model the feature learning process in detail (e.g., many papers suggested by reviewer NQrJ). Our $\mathbf{Li}_2$ framework finds the energy function that governs the feature learning process, and we can precisely characterize the property of its local maxima (i.e., the emerging features), including whether they can predict the label (Theorem 3), how much data are needed to maintain these features (i.e., scaling laws, Theorem 4), characterize the transition between memorization and generalization (Sec. 5.4), and interaction in the feature learning process (Sec. 6), including how partial solutions affect continuous feature learning (Theorem 7). Such a detailed analysis on the feature learning process hasn’t been explored in the previous literature.
>
> > The mathematical framework presented displays the three stages, but fails to offer more insight on what drives these transitions or when do these transitions.
>
> Thanks for the suggestions. In this paper, we made it clear that the backpropagated gradient $G_F$ is the driving force for the three stages. A non-zero $G_F$ kick-starts the independent feature learning process, and its change (e.g., target labels have been partially learned) top-modulates the structure of local maxima (Theorem 7): some local maxima in the energy function will disappear since the corresponding features have been learned and thus the backpropagated gradient $G_F$ has been updated.
>
> As to when these transition happens, we agree that there is complicated dynamics across multiple hyperparameters and stages, and we start with a qualitative characterization for the transition of consecutive stages. A full-fledged detailed analysis will be our future work.
>
> > How does one relate the top-down modulation in Sec 6 to the continuous feature learning observed in [2]?
>
> We have talked about top-down modulation in the previous answer. We are not sure what the reviewer means by “continuous feature learning” in [2]. The closest paragraph we could find in [2] is Sec. 4.3, in which a progress metric (“inverse participation ratio”, or IPR) is defined for grokking dynamics of modular addition, by checking the localization of the frequency components for the weight of each node after Fourier transform. IPR is defined without leveraging any properties of the objective function to be optimized, and may or may not fit to the training process. On the other hand, our analysis leverages the gradient update rules, and thus starts from the first principle.
>
> > Discussion on limitations not found in main text.
>
> We have added a limitation section in our updated revision.
>
> # Questions
>
> > Line 105-106 : I suggest modifying "grokking mostly happens ... regularization" to "grokking is accelerated ... regularization"
>
> Thanks for the comments. We agree that our previous sentences are not accurate. We have updated the revision.
>
> > Why are the axes cut off till epoch 300 in Fig 2 last column?
>
> We cut it to 300 epochs to make the weight changes in the Stage I and II visually more visible. After epoch 300, the model moves to convergence and nothing interesting happens.
>
> > Since the discussion on Thm 3 involves optimization in the complex domain, were any follow-up experiments done with complex weights?
>
> We have tried optimization on complex domains and it works well (check Fig. 11 in our updated revision), which is consistent with the claim of theorem 3. We will put a systematic study later in the next revision.
>
> > The existence of maxima for the ascent functions is interesting (Thms 4-5). What can be said about the network's ability to find these maxima?
>
> Theorem 2 has characterized **all** the local maxima in the energy function for group arithmetic tasks. With SGD/Adam or other component-wise optimizers, these weights are optimized independently starting from different initialization. With sufficiently diverse weight initialization over many hidden nodes (i.e., "over-parameterization"), the network should be able to cover all of the local maxima, by coupon collector argument (Theorem 8).
>
> **Reference**
>
> [A1] A. Jacot et al, Neural Tangent Kernel: Convergence and Generalization in Neural Networks
>
> [A2] S. Du et al, Gradient Descent Provably Optimizes Over-parameterized Neural Networks

---

> > ### Comment · Reviewer_sYi6 · 2025-11-25
> >
> > I thank the authors for their responses which address most of my concerns. However, I still have a few questions/clarifications.
> >
> > (1) **What drives the transitions..:** While I agree that $G_F$ is correlated with the transitions observed in the experiment, $G_F$ is not directly set during initialization or optimization. It's been observed that high data fraction, large initialization, high learning rate, high weight decay, etc. cause/accelerate grokking. It's not clear how these quantities correspond to $G_F$ from first principles.
> >
> > (2) **Continuous feature learning...:** Gromov '23 imply that (generalizing) periodic features emerge continuously. Can this be explained through $G_F$?

---

> ### Author Response · Authors · 2025-11-27
> **Answers to your questions**
>
> Thank you for your further comments. Please check the answers below.
>
> 1. We now find that $G_F$ at initial stage of the Stage I training also delivers the right gradient signal ($\tilde Y\tilde Y^\top F$). This signal is not related to weight regularization and can still trigger feature learning, which explains many intuitions we put in the original rebuttal why grokking still happens without weight decay. Please check Appendix D in the update draft for a more detailed explanation from the first principles that connects $G_F$ with various hyperparameters you mentioned.
>
> 2. This is naturally explained by the top-down modulation (Theorem 7): initially $G_F$ contains all target signal $\hat Y$. Once some features are learned, $G_F$ changes and represent residual signal $\hat Y - \tilde Y$, which leads to a modified energy function that only contains local maxima for missing features that are useful for residual predictions. This is continuous feature learning / emergence.

---

> > ### Comment · Reviewer_sYi6 · 2025-11-27
> >
> > I thank the authors for their detailed responses and updates. I now have a positive opinion of the work and would like to increase my score.

---

> > > ### Author Response · Authors · 2025-11-27
> > > **Thanks!**
> > >
> > > We sincerely thank for your support!

---

### Official Review · Reviewer_LnxJ · 2025-10-31

**Soundness:** 2
**Presentation:** 1
**Contribution:** 2
**Rating:** 4
**Confidence:** 5

**Summary:**

The paper proposes **Li²**, a mathematical framework to explain grokking (delayed generalization) in two-layer nonlinear networks via the structure of the back-propagated gradient matrix (G_F). Training is decomposed into three phases: **(I) Lazy learning**, where (G_F) is effectively random and the top layer overfits random hidden representations; **(II) Independent feature learning**, where each hidden unit learns independently because each column of (G_F) depends only on its own activation—weight decay injects label signal, the dynamics become exact gradient ascent on an energy (E), and the local maxima of (E) coincide with the emergent features; and **(III) Interactive feature learning**, where hidden units begin to interact and (G_F) reorients toward missing features that must be acquired for generalization. On group-arithmetic tasks, the authors analyze when these energy-induced features generalize, their representational power, and how they vary with sample size. The framework yields **provable scaling laws** for memorization and generalization as functions of weight decay, learning rate, and data size, and offers a first-principles explanation for the effectiveness of optimizers such as **Muon**. The analysis is argued to extend to deeper architectures.

**Strengths:**

1. The paper is overall solid and offers a detailed study of the dynamics of two-layer neural networks.
2. It provides experiments that validate the theoretical results.

**Weaknesses:**

1. The writing feels very rushed: many symbols are undefined or unexplained, making the paper hard to follow and overly dense.
2. The theoretical setup is quite restricted; for example, a projection function is deliberately designed so that the hidden layer receives gradients that are random noise.
3. It is unclear whether the group (arithmetic) example pertains only to Stage II or also extends to Stage III.
4. The theoretical analysis of Muon appears disconnected from the main body of the paper, and the setup and assumptions of Theorem 8 are entirely unclear.
5. The relationships among the three stages are not well articulated; the analysis reads like heuristic case-by-case treatment rather than a genuinely unified three-stage dynamics analysis.
6. The abstract mentions scaling laws, but it is not evident where in the paper this is actually developed.

**Questions:**

See weaknesses.

---

> ### Author Response · Authors · 2025-11-21
> **Rebuttal**
>
> We thank the reviewer for the comments and address the concern below.
>
> > The writing feels very rushed: many symbols are undefined or unexplained, making the paper hard to follow and overly dense.
>
> May we know which symbols are not defined so that we can fix them in the next revision.
>
> > The theoretical setup is quite restricted; for example, a projection function is deliberately designed so that the hidden layer receives gradients that are random noise.
>
> The projection $P^\perp_1$ zero-means the target $Y$ and output of the hidden layer $F$, along the batch dimension. This substantially simplifies the theoretical analysis and what is lost is nothing but a difference in global mean. Previous works also have done similar things for theoretical analysis as well as for empirical experiments (e.g., [2], [1]). The random noise backpropagated gradient $G_F$ has nothing to do with the zero-mean projection, but due to the fact that the output weight $V$ is randomly initialized at the beginning of Stage I. At the end of Stage I, $G_F$ is no longer random and its expression is given by Lemma 1. It would be great for the reviewer to elaborate on why s/he got this impression.
>
> > It is unclear whether the group (arithmetic) example pertains only to Stage II or also extends to Stage III.
>
> It is applied to Stage II and Stage III. For Stage III, we have Theorem 7 (top-down modulation) that explicitly uses the group structure of the task.
>
> > The theoretical analysis of Muon appears disconnected from the main body of the paper, and the setup and assumptions of Theorem 8 are entirely unclear.
>
> The analysis on Muon optimizer is to show that such an optimizer can improve the diversity of the learned representation, preventing multiple neurons from falling into the same local maxima (i.e., same feature representation). To demonstrate that this is the case, we assume a specific structure of energy function (Eqn. 11),  and data distribution. We agree that Theorem 8 may be restrictive and will work on a more general case in the future work.
>
> > The relationships among the three stages are not well articulated; the analysis reads like heuristic case-by-case treatment rather than a genuinely unified three-stage dynamics analysis.
>
> We are not sure about the definition of  "a genuinely unified dynamics analysis". In fact, due to its challenging nature, many previous works in gradient dynamics also break it into multiple stages and analyze them separately (e.g., [3]). For grokking, previous works also characterize the feature learning behaviors into lazy and rich/feature learning stages (see [4] which is from reviewer NQrJ), and our framework makes quantitative arguments on which features are learned, as well as how they are learned.
>
> > The abstract mentions scaling laws, but it is not evident where in the paper this is actually developed.
>
> Theorem 4 clearly specifies the scaling laws to achieve generalization. In particular, it gives how many samples are needed to keep the generalizable local maxima in the energy function ($\mathcal{E}$), and how the predicted number of samples are related to the scale of the problem. Fig. 4 demonstrates that this theoretical bound is highly consistent with the experiments. The above is also mentioned by review sYi6 as one positive point. More evidence is shown in the updated version of the paper (check Fig. 9).
>
> **Reference**
>
> [1] Y. Tian, Composing Global Solutions to Reasoning Tasks via Algebraic Objects in Neural Nets.
>
> [2] J. Zbontar et al, Barlow Twins: Self-Supervised Learning via Redundancy Reduction
>
> [3] H. Yang et al, Training Dynamics of Transformers to Recognize Word Co-occurrence via Gradient Flow Analysis
>
> [4] Montanari et al. Dynamical Decoupling of Generalization and Overfitting in Large Two-Layer Networks

---

> > ### Author Response · Authors · 2025-11-26
> > **Any further questions/concerns?**
> >
> > Dear reviewer LnxJ,
> >
> > Let us know whether you have more questions/concerns. Thanks!
> >
> > Authors of paper #10292

---

### Author Response · Authors · 2025-11-21
**Overall rebuttal**

We thank all the reviewers for their constructive and insightful comments.

# Strength
Reviewers agree that the work is solid (LnxJ), novel (aGxx), mathematically rigorous with details (sYi6) and provides a “fairly generic picture of mechanism of feature learning in 2-layer network” (NQrJ). In particular, it provides scaling laws analysis from first principle (sYi6) with empirical justification and can be useful. Reviewers also appreciate that we provide experiments to validate the proposed theory (LnxJ) and connect $\mathbf{Li}^2$ with modern optimizers and deep networks (LnXJ, aGxx).

# Addressing concerns
We address reviewers' common concerns here and put separate concerns in separate comments.

**The framework only works when there is weight decay. However, Grokking may happen without regularization**

One key point of the $\mathbf{Li}_2$ framework is that the non-zero backpropagated gradient $G_F$ initializes the feature learning process. This $G_F$ drives the optimization process of a nice energy function that governs what features will be learned and how they are learned.

From the framework, intuitively, as long as the lazy learning stage generates a sufficient amount of nonzero $G_F$ that is related to the target label, the feature learning will happen. Adding some sorts of regularization (e.g., weight decay) makes sure that there is always a non-zero $G_F$, and it is convenient to use weight decay to demonstrate the feature learning process in a mathematically rigorous way. This does not rule out other possibilities, e.g., $G_F$ becomes large in lazy learning and then vanishes later due to perfect overfitting ($\eta = 0$ in Eqn. 5).

In short, our framework shows that weight decay is sufficient for feature learning (and grokking), but not a necessary condition. A necessary condition is likely that the top layer does not achieve perfect overfit too quickly, which we leave for future work.

We thank the reviewers for providing many references. Here we will leverage our framework to explain why they achieve grokking.

+ [1] uses stable softmax rather than regular softmax (exp->linear). This prevents the model from overfitting to the label too quickly, and thus maintains a nonzero backpropagated gradient that can be useful for feature learning.

+ [2] reports that grokking happens with full-batch GD and Adam without regularization. We have reproduced this behavior. For GD, our hypothesis is that it takes longer for GD to overfit, which gives sufficient non-zero $G_F$. For Adam without regularization, we observed that grokking often happens with a large learning rate, which leads to huge loss at the beginning (“loss overshooting”) and a “shockwave” of non-zero $G_F$ that triggers feature learning. If there is no such “shockwave”, grokking does not happen.

+ Scaling up prediction output [3] or using large initialization may create similar loss overshooting, which leads to grokking behaviors later [4] (Fig. 7).

In summary, according to the framework, we need non-zero backpropagated gradient $G_F$ to start the feature learning process.

**The assumption $\mathbf{x}_i^\top \mathbf{x}_j = \rho$ in Lemma 1 (aGxx and NQrJ)**

We appreciate the reviewers for finding this issue. This assumption is just for convenience in order to prove that $F F^\top = K [(a-b)I + b\mathbf{1}\mathbf{1}^\top]$, i.e., it is a weighted sum of identity and an all one matrix. If $\mathbf{x}_i^\top \mathbf{x}_j$ are not all equal but close, then we can show that $FF^\top$ is not too far away from the form of “multiple of identity + all one matrix”, and the backpropagated gradient $G_F$ remains a similar structure. We have updated the revision to fix it.

# New version uploaded
+ We relaxed the condition $x_i^\top x_j = \rho$ into $\|x_i^\top x_j - \rho\|_2 \le \epsilon$ in Lemma 1 so there is no inconsistency in the hyperparameter regions across theorems (raised by NQrJ).
+ We added limitation and future work (sYi6)
+ We performed additional experiments on the following aspects. All experiments are positive, supporting our theoretical framework.
   + complex domains (Fig.11, asked by sYi6). We find that grokking happens faster in the complex domain.
   + scaling laws on more sophisticated groups such as product group and non-Abelian group (Fig. 9). We find that the theory-backed Scaling Laws (Theorem 4) still remain valid.
   + modified training algorithm by replacing the gradient descent of the top layer V with direct ridge regression (Fig. 12). Grokking still happens.
   + comparison between Adam and Muon (Fig. 7), showing that for few hidden nodes K, Muon indeed leads to lower test loss (and works better), consistent with Theorem 8.

**Reference**

[1] L. Prieto et al, Grokking at the edge of numerical stability.

[2] A. Gromov, Grokking modular arithmetic

[3] L. Chizat et al, On Lazy Training in Differentiable Programming

[4] T. Kumar et al, Grokking as the Transition from Lazy to Rich Training Dynamics

---

### Author Response · Authors · 2025-11-27
**Updated manuscript**

We performed a detailed study of dynamics in the lazy learning regime and find that the backpropagated gradient $G_F$ is dominated by the critical term $\tilde Y\tilde Y^\top F$ not only at convergence, but also **at the initial phase** (the **new Proposition 1** in the update draft). The newly discovered initial phase $G_F$ is not related to weight decay. This further explains grokking behaviors without weight decay regularization.

Therefore, we did another update of the draft.

Here is a breakdown of our updates.
1. We rewrite "Stage I: lazy learning" and analyze $G_F$ both at initial phase and at the converging phase, showing that the critical term $\tilde Y\tilde Y^\top F$ dominates in both cases. Since the term $\tilde Y\tilde Y^\top F$ leads to our energy function $\mathcal{E}$ and kicks off the feature learning process, we could explain why grokking happens without regularization under the same mathematical framework. The detailed derivation is in Section C in the appendix.

2. With the new analysis, we can explain why $|G_F|$ first increases then decay to a constant with grounded mathematics, rather than relying on vague intuitions.

3. We could explain how learning rate, weight initialization, output scaling, etc, are related to $G_F$ from first principles. Please check Appendix D in the updated version for details.

---

### Meta-Review · Area_Chair_kFZK · 2026-01-09

**Summary:**

I believe the authors' successfully resolved critical concerns regarding mathematical consistency. Specifically the conflict between Lemma 1 and subsequent theorems regarding sample size and dimension, and the rigorous generalization of their framework to explain grokking without weight decay. While reviewers initially questioned the theoretical reliance on regularization and the specific drivers of stage transitions, the rebuttal clarified that the backpropagated gradient $G_F$ provides sufficient signal to trigger feature learning during the initial lazy learning phase even without explicit regularization. Although Reviewer NQrJ maintained that the phenomenological overlap with existing "one-big-step" literature could be better contextualized, the consensus is that the proposed Li$_2$ framework offers new, first-principles insight into feature emergence and scaling laws, warranting acceptance.

**Reviewer Concerns:**

*** Addressed

Inconsistent Assumptions: Reviewer NQrJ identified a contradiction between the assumptions in Lemma 1 and subsequent theorems regarding $n$ and $d$. The authors successfully addressed this by relaxing the strict inner product assumption to $|\mathbf{x}_i^\top \mathbf{x}_j - \rho| \le \epsilon$, which NQrJ acknowledged.

Necessity of Weight Decay: Reviewer aGxx and others questioned the reliance on weight decay. The authors provided a new analysis showing that the backpropagated gradient $G_F$ provides a sufficient signal $\tilde{Y}\tilde{Y}^\top F$ in the initial phase even without weight decay, effectively addressing this gap.

Driving Forces of Dynamics: Reviewer sYi6 questioned what drives the transitions between stages. The authors provided a detailed discussion connecting $G_F$ to hyperparameters like learning rate and initialization, leading to sYi6 explicitly stating they would raise their score.

Scaling Laws: Reviewer LnxJ missed the scaling laws in the initial read; the authors pointed to Theorem 4 and Fig. 4, clarifying the contribution.

*** Outstanding

Literature Context: Reviewer NQrJ maintained that while the assumptions differ from prior work (teacher-student, one-pass SGD), the phenomenological overlap should be better contextualized in the text. The authors argued their global energy function approach differs fundamentally from "one-big-step" analysis, but this interpretative difference remains.

Presentation: Reviewer LnxJ found the writing "rushed" with undefined symbols. While the authors asked for specifics and clarified the notation in the rebuttal, the reviewer did not engage further to confirm if the readability concerns were fully resolved.

**Reviewer Scores:**

Reviewer aGxx: maintained their score of 8 (Accept) indicating the rebuttal regarding loss functions and weight decay was satisfactory.

Reviewer sYi6: likely raised their score from 4 to 6 or 7. They explicitly commented, "I now have a positive opinion of the work and would like to increase my score," following the authors' explanation of the driving forces behind the stage transitions and continuous feature learning.

Reviewer NQrJ: likely raised their score from 4 to 5 or 6. The authors fixed the primary technical flaw (the inconsistent assumptions) which NQrJ acknowledged ("I thank you... for the additional analysis"). While they still had reservations about the literature comparison, the removal of the mathematical inconsistency might warrant a score increase.

Reviewer LnxJ: likely would have raised their score from 4 to 5 had they participated. The authors clarified the technical questions regarding the projection matrix and the application of group theory to Stage III. However, without their confirmation, the concern about "rushed writing" might prevent a higher score.

---

### Decision · Program_Chairs · 2026-01-26

Accept (Poster)